# Can LLMs Reason Over Non-Text Modalities in a Training-Free Manner? A Case Study with In-Context Representation Learning

**Tianle Zhang**[1][*], **Wanlong Fang**[2,1][*], **Jonathan Woo**[5,6][*], **Paridhi Latawa**[7],
**Deepak A. Subramanian**[7], **Alvin Chan**[1,3,4][†]

[1]College of Computing and Data Science, Nanyang Technological University
[2]AI-X, Interdisciplinary Graduate Programme, Nanyang Technological University
[3]Lee Kong Chian School of Medicine, Nanyang Technological University
[4]Centre of AI in Medicine (C-AIM), Nanyang Technological University
[5]University of Toronto, [6]Brigham and Women's Hospital, Harvard Medical School
[7]Massachusetts Institute of Technology

## Abstract

The remarkable performance of Large Language Models (LLMs) can be enhanced with test-time computation, which relies on external tools and even other deep learning models. However, existing approaches for integrating non-text modality representations into LLMs typically require additional costly supervised training, restricting on-the-fly adaptation to new domains and modalities. In this work, we explore the feasibility of integrating representations from non-text foundational models (FMs) into text-based LLMs in a training-free manner. We propose In-Context Representation Learning (ICRL) as a proof-of-concept to allow LLMs to adaptively utilize non-text modality representations with few-shot learning. Unlike traditional in-context learning, which incorporates text-label pairs, ICRL replaces text inputs with FM representations, enabling the LLM to perform multi-modal inference without fine-tuning. We evaluate ICRL on a suite of tasks in the molecular domain, investigating three core research questions: (i) how to map FM representations into LLMs in a training-free manner, (ii) what factors influence ICRL performance, and (iii) what mechanisms underlie the effectiveness of ICRL. To the best of our knowledge, ICRL is the first training-free framework for integrating non-text modality representations into text-based LLMs, presenting a promising direction for adaptable, multi-modal generalization.[3]

## 1  Introduction

Large Language Models (LLMs) have demonstrated remarkable versatility in leveraging test-time computation [57, 48], allowing them to dynamically adapt to new tasks and perform complex reasoning without additional training. A key extension of this capability is their ability to integrate external tools, including other deep learning models [4, 10], to enhance their problem-solving potential. Existing multi-agent frameworks allow LLMs to incorporate predictions from external models during inference, expanding their applicability to complex, multi-step tasks [49, 24]. Despite the advances, these methods typically use only the final outputs of external models, limiting the effective use of their internal knowledge [25, 49]. To address this, recent efforts have explored enabling LLMs to leverage intermediate representations from external models. A promising direction involves allowing

---

[*]Equal contribution.
[†]Corresponding author. Email: guoweialvin.chan@ntu.edu.sg
[3]Our code is available at https://github.com/ztlmememe/LLMxFM_ICRL.

text-based LLMs to process non-text modalities (e.g., images [34]) by incorporating representations from modality-specific foundation models (FMs) (Fig. 1(b)). However, these approaches typically require additional supervised training—either for modality-specific projection layers or even for the LLM itself—to enable a text-based LLM to incorporate a new modality. This process is typically computationally intensive and requires specialized paired dataset between text and the target new modality. This raises a fundamental question: *Can a text-based LLM leverage representations from other modality-specific foundational models during inference, without such training?*

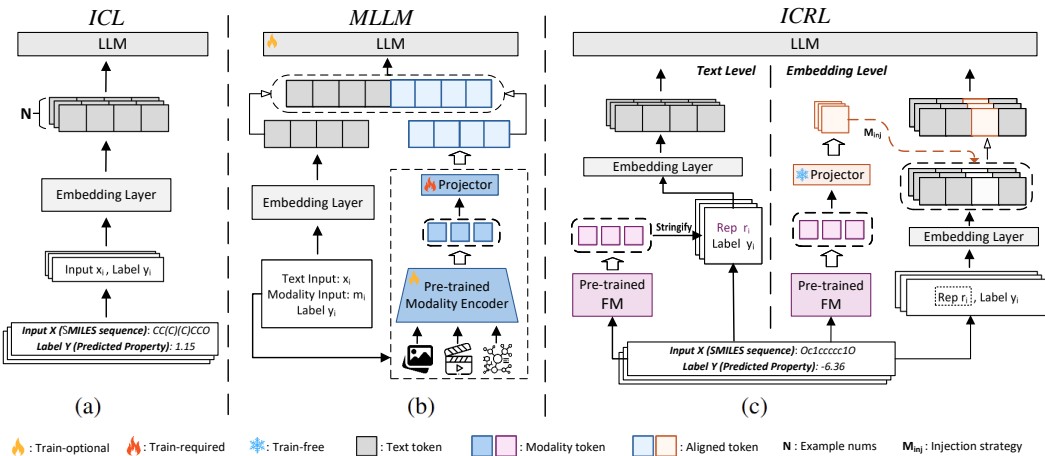

Figure 1: Comparison of (a) ICL, (b) multi-modal LLM, and (c) in-context representation learning.

In-context learning (ICL), a core capability of large language models (LLMs), enables task adaptation at inference time through few-shot examples [56, 12], offering a promising path toward flexible, training-free generalization. In this study, we explore how ICL can be extended to process representations from modality-specific foundation models (FMs) during inference. To this end, we propose **I**n-**C**ontext **R**epresentation **L**earning (ICRL)—a proof-of-concept designed to enhance LLM adaptability and enable a single model to generalize efficiently across diverse modalities (Fig. 1(c)). In contrast to ICL where text-label pairs (Fig. 1(a)) are included in the text prompt for the LLM to conduct few-shot prediction, ICRL replaces the sample text (e.g. SMILES of a molecule[4]) with the representation from an external FM (Fig. 1(c)). Note that our primary objective is not to outperform ICL but to investigate the feasibility of adaptively integrating non-text FM representations into a text-based LLM in a *training-free* manner. We study ICRL and its mechanisms on a battery of molecular tasks and organize our findings around 3 core research questions as follows.

---

**RQ1 (Sec. 2 & 3): How to Map FM Representations into an LLM in a Training-free Manner?**

---

We first study ICRL designs that integrate the FM representation vectors directly in the text prompt as few-shot examples. To fit these high-dimensional vectors as text within LLM's limited context window, we employ dimensionality reduction (i.e., PCA) and find this simple strategy surprisingly effective. We also investigate using a projection module to interface between the FM representations and a text-based LLM's embedding layer. To enable inference-time integration of the non-text modality, we also examine several training-free projection methods that map FM representations into the LLM's embedding space without fine-tuning and compare their empirical performances. Notably, our analysis indicates that an approach based on optimal transport theory, which aligns the distribution of FM representations with that of LLM embeddings, yields promising results.

---

**RQ2 (Sec. 4): What Factors affect ICRL Performance?**

---

We found that key parameters affecting standard ICL, such as the number of few-shot examples, similarly influence ICRL. Additionally, we analyze various parameters in ICRL to gain deeper insights into their impact on performance.

---

[4]To ensure a fair comparison, we use molecular datasets where SMILES strings encode structural information in a textual format naturally compatible with LLMs, avoiding the need for architectural changes or fine-tuning. This allows us to study non-text modality integration into ICL pipelines in a clean, training-free setting. Related modalities (e.g., protein sequences) and preliminary results on vision and speech are discussed in Sec. 7.

> **RQ3 (Sec. 5): What are the Mechanisms behind ICRL?**

Our findings show that ICRL performance improves when projected FM representations closely resemble their corresponding text embeddings, while larger deviations can harm performance. Additionally, we observed an inverse correlation between the similarity of ICRL-projected representations across different few-shot examples and ICRL performance, implying that excessive uniformity among projected representations may hinder effectiveness. Lastly, we found that when traditional ICL is present, the mode of ICRL representations shifts, leading them to be treated as pause tokens [17].

The main contributions of this paper are as follows: (i) To the best of our knowledge, the proposed in-context representation learning is the first *training-free approach* to integrate non-text modalities into a text-based LLM. (ii) We explore various design choices and analyze their impact on ICRL performance across a range of molecular domain tasks. (iii) We present mechanistic insights behind ICRL to show how the distribution of projected representation affects performance.

## 2 How to Map FM Representations into an LLM in a Training-free Manner?

In this section, we first discuss the proposed ICRL framework, which encompasses two different locations to integrate FM representations into a text-based LLM: (1) introduction of the representation as a *string* in the prompt text and (2) injection of the FM features into the LLM *embedding* spaces. Then, we introduce several methods that constitute the proposed ICRL framework; detailed algorithmic descriptions of the overall pipeline and each injection strategy are provided in Appendix D.

### 2.1 Preliminaries

**In-Context Learning & LLM.** Consider the general ICL framework [12]: given a set of text inputs $\mathbf{X} = [x_1, \ldots, x_n]$, i.e., molecular SMILES sequences, and the task label $y \in \mathcal{Y}$, a pre-trained LLM $\psi$ predicts by selecting the candidate with the highest score based on a demonstration set $E$. This set comprises the instruction $I$ and $k$ demonstration examples: $E = \{I, (x_1, y_1), \ldots, (x_k, y_k)\}$, where each $(x_i, y_i)$ represents a few-shot learning exemplar. The final prediction $\hat{y}$ is computed using a scoring function $f$ over the entire input sequence:

$$\hat{y} = \arg \max_{y_j \in \mathcal{Y}} f_\psi(y_j, E, \mathbf{X}). \tag{1}$$

where $\mathbf{X}$ represents a set of text examples. Within this framework, the LLM $\psi$ implements the above formulation through sequential token processing. Specifically, for each input sequence $x_i$, the embedding layer of LLM $\psi_e$ converts it into a sequence of token embeddings: $g_i = \psi_e(x_i) \in \mathbb{R}^{t_i \times d_{LLM}}$, where $t_i$ is the number of tokens corresponding to $x_i$, and $d_{LLM}$ is the dimensionality of each token embedding. For convenience, the distribution of LLM outputs is denoted as: $\mathcal{D}_{LLM} \in \mathbb{R}^{T \times d_{LLM}}$, where $T = \sum_{i=1}^{n} t_i$ is the total number of tokens across all inputs.

**Foundation Model.** We define the FM as a mapping: $\phi : \mathbf{X} \to \mathbb{R}^{N \times d_{FM}}$, and the extracted representations are structured as: $\mathbf{H} \in \mathbb{R}^{N \times M \times d_{FM}}$, where $M$ is the number of extracted feature vectors, and $d_{FM}$ is the corresponding dimensionality. In this article, we focus on the sentence-level classification feature, i.e., $M = 1$, and denote the distribution of the extracted representations as $\mathcal{D}_{FM} \in \mathbb{R}^{N \times d_{FM}}$. Further details on feature extraction schemes can be found in Appendix F.6.

**Projector.** We employ a simple multilayer linear model (MLM) to align the FM and LLM embedding spaces, defined as $P : \mathbb{R}^{d_{FM}} \to \mathbb{R}^{d_{LLM}}$. The projected embedding distribution is denoted as $\mathcal{D}_{Proj}$.

### 2.2 In-Context Representation Learning

Unlike standard ICL, which constructs examples $(x_i, y_i)$ only from textual inputs, and Multi-modal Large Language Model (MLLM) methods requiring supervised training for modality alignment, ICRL directly injects adjusted FM representations into LLMs without training, i.e., $(r_i, y_i)$, bypassing raw input $x$. We classify our methods into two injection levels and discuss their design and theoretical foundations in the following subsections.

#### 2.2.1 Text-Level Injection

**PCA.** A straightforward approach to integrating FM representations into LLMs is to embed the high-dimensional vectors directly into the prompt as strings. However, these high-dimensional embeddings

often surpass the context window limitations of most models and make them incompatible with ICL approaches. To address this challenge, we implement PCA for dimensionality reduction. This transformation maps the original embeddings to a lower-dimensional space: $\mathcal{D}_{PCA} \in \mathbb{R}^{N \times d_{Reduced}}$, where $d_{Reduced} \ll d_{FM}$. Let $\mathbf{W}_{PCA} \in \mathbb{R}^{d_{FM} \times d_{Reduced}}$, the reduced-dimensional embeddings $H_{pca} = H \times W_{PCA}$ are subsequently converted into PCA strings $S_{pca}$, enabling their seamless integration into the prompt while retaining the most critical features of the original representations.

### 2.2.2 Embedding-Level Injection

While dimensionality reduction enables the FM representations to fit within the text prompt, the vector strings still occupy a substantial portion of the prompt and incur information loss, thereby limiting their effectiveness. To address these limitations and enhance the LLM's understanding of these representations, we propose several embedding-level injection methods that directly inject features into the LLM's text embedding layer:

**Zero-Pad.** The simplest approach applies zero padding to FM representations to match the dimensionality of the LLM embedding space, offering a straightforward solution that preserves the original representation components. Then, we apply a normalization step (as detailed in Appendix G.1) to adjust the mean and variance of the padded representation to match the average mean and variance of the LLM's embeddings before feeding it into the LLM. This prevents the generation of irrelevant or nonsensical outputs due to input embeddings with out-of-distribution statistics.

**Random Projection.** We employ a randomly initialized MLM as a projector module to address the dimension mismatch problem. In contrast to previous studies [29], this projector is **untrained** and devoid of activation functions, which is mathematically equivalent to simple matrix multiplication, making it a lightweight and efficient solution. Then the projected features are directly concatenated with the embeddings of the rest of the example.

**Optimal Transport Alignment.** Directly using random projectors may result in a distribution mismatch between the LLM's embeddings and the mapped FM representations. As a mathematical framework designed to align two distributions, optimal transport (OT) provides a viable solution to resolve this mismatch [53]. In this approach, we extract the tokens corresponding to the original input $x_i$ or the representation string obtained by the PCA model. These token embeddings serve as the target distribution in OT to adjust the projected results $\mathbf{H}_{proj} = P(\mathbf{H})$. The alignment process is formalized as follows: Let $\mathcal{D}_{proj} \in \mathbb{R}^{N \times d_{LLM}}$ and $\mathcal{D}_{tar} \in \mathbb{R}^{N' \times d_{LLM}}$ represent the source and target distributions, respectively, where $N'$ denote the number of tokens in the target distribution. The final objective can be formulated as:

$$\min_{\gamma \in \Pi(\mu, \nu)} \int_{\mathcal{D}_{proj} \times \mathcal{D}_{tar}} c(u, v) \, \partial \gamma(u, v), \tag{2}$$

where $\gamma \in \Pi(\mu, \nu)$ is the transport plan that defines how to map the points in the source distribution to the target one, $\mu$ and $\nu$ are the marginal distributions over $\mathcal{D}_{\text{proj}}$ and $\mathcal{D}_{\text{tar}}$, respectively. $c(u, v)$ is the function that measures the cost of moving an element $u$ from $\mathcal{D}_{proj}$ to $v$ in $\mathcal{D}_{tar}$.

For practical implementation, we align the mean and variance of each dimension of $\mathcal{D}_{proj}$ to match the corresponding dimension of $\mathcal{D}_{tar}$, i.e., for each dimension $j$, we have:

$$shift_j = \bar{\mathbf{v}}_j - \bar{\mathbf{u}}_j \quad \text{and} \quad scale_j = \frac{\sigma_{t,j}}{\sigma_{p,j}}, \tag{3}$$

where $\bar{\mathbf{u}}_j$, $\bar{\mathbf{v}}_j$, $\sigma_{p,j}$ and $\sigma_{t,j}$ represent the means and standard deviations of the $j$-th dimension of $\mathcal{D}_{proj}$ and $\mathcal{D}_{tar}$, respectively. The final aligned embeddings $\mathbf{H}_{aligned}$ are obtained by applying the following transformation:

$$OT(\mathcal{D}_{proj}, \mathcal{D}_{tar}) = scale \cdot \mathbf{H}_{proj} + shift. \tag{4}$$

Given the differing target distributions, we propose two OT-based alignment methods:

**OT - Embed.** To enhance the interpretability of the representation for the LLM, a suitable alignment target is the LLM embedding of input text features (e.g., SMILES). Specifically, for each input $x_i$, the target embedding $\psi_e(x_i)$ is computed as the mean of its token-level embeddings. The adjusted embeddings can then be represented as: $\mathbf{H}_{\text{aligned}} = OT(\mathbf{H}_{\text{proj}}, \psi_e(\mathbf{X}))$.

**OT - PCA.** Another OT variant uses the embeddings of the stringified, dimensionally reduced FM representations $S_{\text{PCA}}$ as the target distribution, as this approach more closely captures the

token embeddings associated with FM representations. This can be expressed as: $\mathbf{H}_{\text{aligned}} = OT(\mathbf{H}_{\text{proj}}, \psi_e(\mathbf{S}_{\text{pca}}))$. Note that the computation of OT $shift$ and $scale$ parameters needs to be performed only once and takes negligible time, the subsequent use only requires quick adjustments. We describe the OT method in more detail in Alg. 5.

**Random Noise.** To verify whether the model is learning from the FM representation, we conduct an ablation study by replacing informative FM features with random noise.

**Theoretical support for linear projector.** In designing the projector, we initially experimented with a commonly used two-layer MLP. However, the mapped representations show greater similarity, suggesting a noticeable loss of information [31]. Here, we conduct a theoretical analysis to examine how nonlinear activations in the projector layers can affect the original embedding geometry.

Following [31], given a linear layer with random initialization. Let $\mathbf{W} \in \mathbb{R}^{d \times d}$ be a random square weight matrix such that each entry $\mathbf{W}_{k,l}$ is an i.i.d. Gaussian random variable, i.e., $\mathbf{W}_{k,l} \sim \mathcal{N}(0, \frac{1}{d})$. We consider affine transformations of the form $\mathbf{W}\mathbf{x} + \mathbf{b}$, where $\mathbf{b} \in \mathbb{R}^d$ is a random bias vector. Our objective is to demonstrate that such random linear mappings preserve the norms and angles of high-dimensional vectors, thereby retaining the underlying variability of the original embeddings.

**Theorem 1** (Concentration of Norm Under Random Linear Projector). *Let $\boldsymbol{u} \in \mathbb{R}^d$ be a fixed vector, and that $\boldsymbol{W} \in \mathbb{R}^{d \times d}$ has i.i.d. entries $\boldsymbol{W}_{k,l} \sim \mathcal{N}(0, 1/d)$, independent of a bias vector $\boldsymbol{b} \in \mathbb{R}^d$. Then for any $\delta_1 \in (0,1)$, with probability at least $1 - \delta_1$, there exists $\epsilon_1 = O(\sqrt{\log(1/\delta_1)/d})$ such that:*

$$\left| \|\boldsymbol{W}\boldsymbol{u} + \boldsymbol{b}\|^2 - \left( \|\boldsymbol{b}\|^2 + \|\boldsymbol{u}\|^2 \right) \right| \leq \epsilon_1 \left( \|\boldsymbol{b}\|^2 + \|\boldsymbol{u}\|^2 \right). \tag{5}$$

*Proof Sketch.* Each coordinate $(\mathbf{W}\mathbf{u} + \mathbf{b})_k$ is a sum of Gaussian variables with variance on the order of $\|\mathbf{u}\|^2/d$, shifted by $\mathbf{b}_k$. By applying classical concentration inequalities (e.g., Chebyshev's), the squared norm $\|\mathbf{W}\mathbf{u} + \mathbf{b}\|^2$ concentrates around its mean $\|\mathbf{b}\|^2 + \|\mathbf{u}\|^2$. We provide the full derivation and proof for all theorems in Appendix E.1 and E.2. □

**Theorem 2** (Preservation of Cosine Similarity). *Let $\boldsymbol{u}, \boldsymbol{v} \in \mathbb{R}^d$ be any two fixed vectors, and $\boldsymbol{W} \in \mathbb{R}^{d \times d}$ have i.i.d. entries $\boldsymbol{W}_{k,l} \sim \mathcal{N}(0, 1/d)$. Then for any $\delta_2 \in (0,1)$, there exists a small $\epsilon_2 = O(\sqrt{\log(1/\delta_2)/d})$ such that with high probability at least $1 - \delta_2$, we have:*

$$\left| \cos(\boldsymbol{W}\boldsymbol{u}, \boldsymbol{W}\boldsymbol{v}) - \cos(\boldsymbol{u}, \boldsymbol{v}) \right| \leq \epsilon_2. \tag{6}$$

**Remark.** We set the bias term $\mathbf{b} = \mathbf{0}$ in our theoretical analysis and subsequent experiments. This choice enables exact matching between pre- and post-projection cosine similarities, as any non-zero bias would introduce additional terms that obscure this relationship.

**Corollary 1.** *Nonlinear activations may distort vector angles and inflate similarities. Formally, let $\sigma(\cdot)$ be a nonlinear activation (e.g., ReLU, sigmoid). Then with high probability:*

$$\left| \cos(\sigma(\mathbf{W}\mathbf{u}), \sigma(\mathbf{W}\mathbf{v})) - \cos(\mathbf{u}, \mathbf{v}) \right| > \left| \cos(\mathbf{W}\mathbf{u}, \mathbf{W}\mathbf{v}) - \cos(\mathbf{u}, \mathbf{v}) \right|. \tag{7}$$

This effect arises because both norms and dot products tend to concentrate under random linear mappings $\mathbf{x} \mapsto \mathbf{W}\mathbf{x}$, thus preserving the geometric distinctions (i.e., angles) among points in high-dimensional space [31]. In contrast, nonlinear activations with range constraints or sparsity-inducing properties (e.g., ReLU, sigmoid) may suppress variation and exaggerate alignment, leading to inflated cosine similarities even for originally dissimilar inputs. Empirical evidence and theoretical proof supporting this corollary are presented in Sec. 4 and Appendix E.3.

## 3 Experimental Results

In addition to constructing examples from representations, i.e. $(r_i, y_i)$, ICRL can also utilize both the original textual input and its corresponding FM features to form examples like $(x_i, r_i, y_i)$. Extensive experiments show that adding representations can further enhance ICL performance.

### 3.1 Experiments Setup

**Datasets.** We evaluate our ICRL method on five molecular datasets: ESOL [11], Caco_wang [54], AqSolDB [52], LD50_Zhu [70], and AstraZeneca [58]. For larger datasets, following [2], we

Table 1: RMSE (↓) Comparison of ICRL Across Datasets. **Bold**/ Underline: best/second-best value among the Embedding Injection methods. Ran-Noi and Ran-Pro denote the Random Noise and Random Projection methods, respectively.

| Dataset | Text Injection | | Embedding Injection | | | | |
|---|---|---|---|---|---|---|---|
| | ICL | PCA | Zero-Pad | Ran-Noi | Ran-Pro | OT-Embed | OT-PCA |
| ESOL | 1.16 ±1.9e-2 | 1.11 ±2.3e-4 | 1.73 ±4.5e-2 | 1.41 ±3.5e-3 | 1.69 ±4.4e-2 | **1.19** ±1.8e-3 | 1.24 ±4.0e-3 |
| Caco2_Wang | 0.83 ±8.4e-4 | 0.95 ±2.4e-3 | 1.04 ±4.1e-3 | 1.03 ±4.4e-3 | 1.03 ±1.3e-3 | 0.89 ±2.4e-3 | **0.88** ±1.5e-3 |
| AqSolDB | 1.92 ±3.9e-4 | 2.91 ±1.5e-2 | 4.01 ±5.3e-2 | 3.95 ±4.8e-3 | 4.02 ±8.0e-2 | 4.06 ±1.3e-1 | **3.25** ±5.9e-2 |
| LD50_Zhu | 0.99 ±1.7e-4 | 1.06 ±2.5e-4 | 1.28 ±9.0e-4 | 1.21 ±1.8e-3 | 1.29 ±1.3e-3 | 1.18 ±1.2e-3 | **1.14** ±1.3e-4 |
| AstraZeneca | 1.37 ±2.4e-3 | 1.39 ±5.1e-5 | 1.55 ±2.0e-3 | 1.50 ±1.3e-2 | 1.54 ±9.2e-4 | **1.46** ±3.9e-4 | 1.47 ±5.4e-4 |

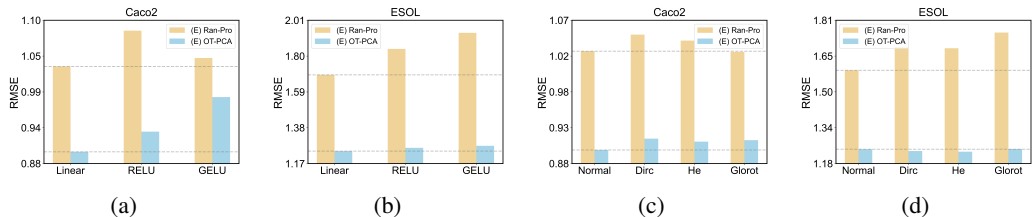

|  (a)  |  (b)  |  (c)  |  (d)  |

Figure 2: (a) and (b) present the performance with and without activation functions in the projector across two datasets, evaluated by RMSE (↓), demonstrating that linear projectors can achieve superior results. (c) and (d) depict the impact of different projector initialization strategies, indicating that these choices have minimal influence on performance.

randomly select 1,000 samples for the test set. Additional experiments and discussions on protein and drug-target interaction datasets can be found in Appendix F.7.

**Implementation & evaluation.** Unless stated otherwise, raw text input features are omitted in ICRL. We use Uni-Mol [68] to generate molecular representations and Llama-3.1-70B-Instruct [18] for inference. During inference, we set the number of examples (shots), the PCA target dimensionality to 20, and the batch query size to 3. The projector is a two-layer MLM with 64 hidden units per layer. To ensure result stability, each method is evaluated using ten random seeds, with final results derived from the top three runs. Details about the dataset and implementation are in Appendices C and G.1.

### 3.2 Which Injection Method is more effective?

This subsection evaluates various ICRL methods across different scenarios, analyzing their strengths and limitations [5]. It is important to note that our aim is not to claim state-of-the-art performance, but rather as an initial probe into *training-free* approaches for LLMs to leverage FM representations. Given that PCA and OT processes incur a negligible fraction of computational cost (take less than **two seconds**, with only **one additional token per sample** in the input window) compared to training-based approaches, we focus on comparing several representation injection strategies under similar conditions, and include ICL results as a reference. A detailed discussion is provided in Sec. 7.

**Comparison of representation injection approaches.** As shown in Tables 1, 24 & 25, the text-level injection approach PCA outperforms other embedding-level methods on most datasets. These results demonstrate that LLMs can effectively interpret and utilize features injected in this manner, while its reliance on a large context window limits scalability. In contrast, the OT-based approaches achieve comparable results with minimal context window usage, i.e., one token per FM representation. Conversely, the Zero-Pad and Random Projection methods perform poorly, often falling below that of Random noise, indicating that these simplistic techniques fail to generate suitable representations.

**ICRL performance with original text features (SMILES).** Incorporating SMILES sequences with ICRL reveals that most embedding-level injection methods could improve performance over ICL with only SMILES strings (Table 2, 22 & 23). Notably, on the ESOL dataset, the OT-PCA method

---

[5]The conclusions we proposed are consistent across different metrics, i.e., Pearson's correlation coefficient (Pearson $r$) and Root Mean Square Error (RMSE). Additional details can be found in Appendix G.2.1.

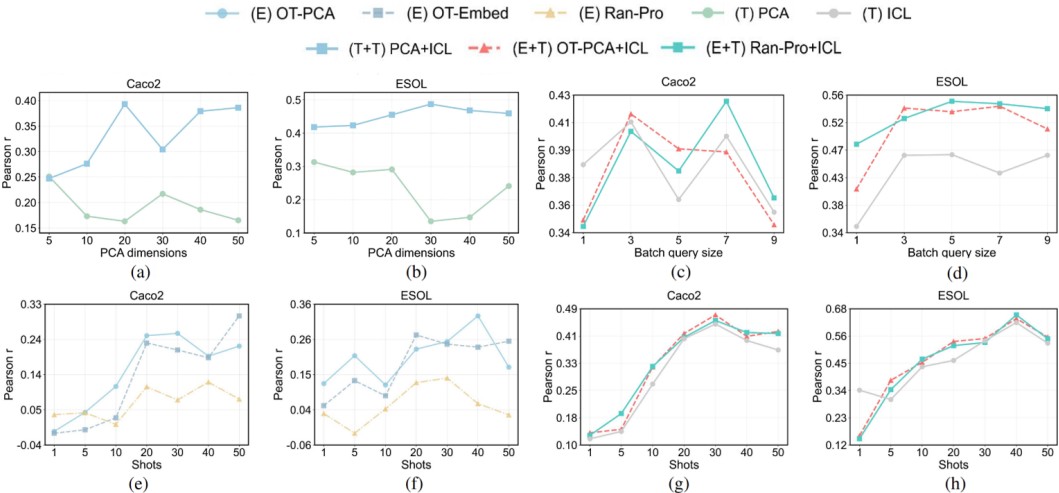

Figure 3: (a) and (b): performance of various methods under different PCA dimensions. (c) and (d): different methods behave similarly across batch query size settings. (e) - (h): Pearson correlation (↑) with increasing example count, demonstrating enhanced learning via OT-adjusted representations and feature injection. E and T denote different injection levels.

achieves a significant 16.6% improvement compared to using textual input alone (measured by Pearson r). Surprisingly, while LLMs exhibit a strong ability to interpret PCA strings and OT-based representations, their combined benefits with text features are less pronounced compared to simpler methods. Specifically, the Random Noise method consistently outperforms the baseline across all datasets when textual input is included, and Zero-Pad delivers superior results in most cases. In contrast, OT-based methods show inconsistent gains, and the PCA method—despite its prior strong performance—degrades performances across all datasets. Further analysis is provided in Sec. 5.

# 4    What Factors affect ICRL Performance?

**Model Capability.** The effectiveness of ICRL is closely linked to the capacity of the underlying pre-trained LLM. As shown in Tables 20 and 21, larger models are generally better at leveraging FM-derived representations and handling in-context prompts. In contrast, smaller models (i.e., Llama-3.2-3B-Instruct) exhibit noticeable performance degradation on both ICL and ICRL (see Appendix F.8). Despite this, ICRL achieves performance comparable to, and in some cases even surpassing ICL in these smaller models. This suggests that, under capacity constraints, OT-aligned FM embeddings serve as an informative input representation compared to the baseline SMILES strings (see Appendix F.9). Consequently, ICRL demonstrates strong potential as a lightweight and effective approach for enabling small-scale LLMs to leverage foundation model knowledge, providing a promising alternative to costly supervised learning in resource-constrained settings.

**ICRL projector schemes.** As our projector is untrained, its structural design—specifically, the inclusion of activation functions and initialization choices—plays a crucial role. We evaluate the effects of incorporating ReLU and GELU activation functions between two linear layers and explore four common initialization methods: Glorot initialization [16], He initialization [20], Dirac initialization–which structures the weight matrix to approximate an identity matrix, and standard normal initialization. Results in Figs. 2(a) and 2(b) demonstrate that activation functions consistently degrade performance, which aligns with our theoretical analysis. Additionally, Figs. 2(c) and 2(d) suggest that while alternative initialization methods may provide minor improvements, the standard normal initialization consistently yields the best overall performance. Furthermore, initialization choices have a relatively minor impact compared to activation function settings.

**PCA dimensions.** As the dimensionality of PCA affects the length of text-level injected representations, we conducted an ablation study to further investigate its impact. Interestingly, in scenarios without text input, the PCA method did not yield better performance with increasing representation length (Figs. 3(a) and 3(b)). In most cases, performance even declined, suggesting that longer representations do not necessarily enhance the model's ability to interpret them. In contrast, when

Table 2: Pearson Correlation (↑) Comparison Across Datasets. **Bold**/ Underline: best/second-best value compared with ICL.

| Dataset | Baseline | | ICRL (Ours) | | | | |
| --- | --- | --- | --- | --- | --- | --- | --- |
| | Text | Text | | | Embedding | | |
| | ICL | PCA+ICL | Zero-Pad+ICL | Ran-Noi+ICL | Ran-Pro+ICL | OT-Embed+ICL | OT-PCA+ICL |
| ESOL | 0.465 ±9.2e-4 | 0.455 ±1.2e-4 | 0.526 ±2.1e-4 | 0.540 ±1.6e-3 | 0.525 ±6.5e-5 | 0.508 ±1.7e-4 | **0.542** ±5.4e-4 |
| Caco2_Wang | 0.411 ±1.3e-3 | 0.393 ±9.2e-2 | 0.410 ±4.6e-6 | 0.420 ±1.1e-4 | 0.405 ±1.6e-5 | **0.429** ±1.1e-3 | 0.394 ±5.7e-4 |
| AqSolDB | 0.596 ±5.1e-5 | 0.549 ±3.2e-4 | **0.606** ±6.8e-6 | 0.597 ±1.1e-5 | 0.600 ±2.4e-5 | 0.569 ±5.7e-4 | 0.589 ±3.9e-5 |
| LD50_Zhu | 0.378 ±1.2e-5 | 0.356 ±1.9e-4 | **0.393** ±8.6e-6 | 0.379 ±5.4e-6 | 0.392 ±7.3e-5 | 0.361 ±1.2e-5 | 0.362 ±7.8e-5 |
| AstraZeneca | 0.266 ±2.3e-5 | 0.227 ±3.1e-5 | **0.272** ±4.8e-5 | 0.267 ±2.1e-5 | 0.269 ±1.9e-5 | 0.269 ±2.1e-4 | 0.271 ±6.6e-5 |

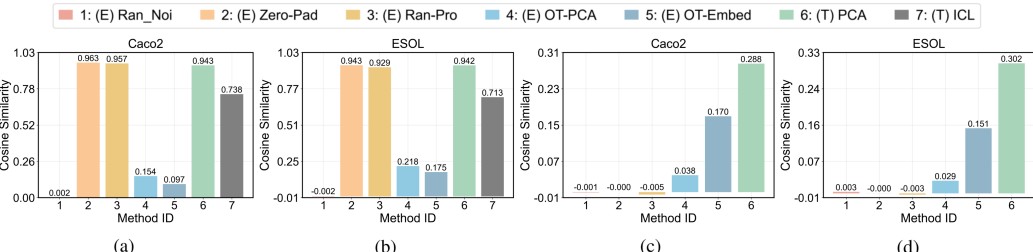

Figure 4: (a) and (b) illustrate the mean cosine similarity of ICRL representations of different molecules in different ICRL methods. (c) and (d) present the mean pairwise cosine similarity between the ICRL representations and their corresponding SMILES text embeddings.

both textual and representational inputs were considered, longer representations contributed to a deeper understanding of SMILES sequences to some extent. Additionally, for embedding-level injection methods, the length of the injected item remains fixed, making this parameter only marginally influential on the target distribution. Detailed results and analyses are provided in Appendix F.10.

**Do key ICL parameters affect ICRL similarly?** ICL has been shown to benefit from more examples and batch query processing [23]. This raises a key question: does ICRL exhibit similar behavior? To investigate, we conduct ablation experiments on these parameters, leading to the following findings:

(1) **ICRL benefits from an increased number of examples.** As illustrated in Figs. 3(e) and 3(f),

OT-based methods demonstrate a noticeable upward trend with an increasing number of examples, suggesting that LLMs can effectively interpret injected representations and leverage additional examples to enhance the performance of downstream tasks. In contrast, unadjusted representations show limited gains. Since ICRL requires significantly less context space, this finding suggests that embedding-level injection methods have the potential to achieve comparable performance to standard ICL by using a larger number of examples.

(2) **ICRL behaves similarly to ICL.** Figs. 3(e) to 3(h) illustrate the performance trends when incorporating both textual input and corresponding representations compared to using textual input alone under varying conditions. Specifically, both approaches demonstrate synchronized performance changes with increasing batch query sizes, and as the number of examples grows, consistent improvements are observed across all methods. These results indicate that the inclusion of representations does not alter the overall trends observed in textual-only approaches. Instead, it generally enhances the LLM's capability to interpret SMILES sequences within the provided examples.

## 5 What are the Mechanisms Behind ICRL?

In this section, we further analyze and discuss ICRL's distinct behaviors based on several observations:

*(1) High similarity between ICRL representations degrades performance.* Specifically, the FM features are typically confined to a narrower space [31], leading to highly similar mapped embeddings (Figs. 4(a) and 4(b)). Thus, the LLM perceives minimal differences between samples, leading to

random predictions based on the example label distribution, as further analyzed in Appendix F.11. However, representation similarity alone does not fully explain the results. For instance, methods like Random Noise exhibit poor performance despite their low similarity, whereas OT-based methods, which demonstrate moderate diversity, achieve superior outcomes. This underscores the importance of representation-text alignment in effective utilization.

*(2) Improved FM features do not necessarily lead to better improvement when text features are included.* Interestingly, although the OT-based method shows the best ICRL performance, it does not consistently achieve optimal results when integrated with textual features. Instead, simpler methods often provide more effective enhancements when combined with text input, as demonstrated in Table 2. An analysis of attention weights (Fig. 5) reveals that the model predominantly focuses on the SMILES sequences, which are more familiar and occupy a significantly larger portion of the context window compared to the injected representations. This suggests that the LLM tends to ignore the injected representations rather than actively learning from them.

**The dual operational modes of ICRL representations in different scenarios.** When prompts consist solely of ICRL representations as input features, the demonstrations differ significantly from the data encountered during pre-training. As a result, the model operates primarily in a *task learning* mode [33], relying heavily on the few-shot ICRL exemplars for prediction of test samples. This mode poses a challenge when the few-shot examples are insufficient, making both the prediction and the representation interpretation difficult. As shown in Figs. 3(e) and 3(f), when fewer than ten examples are provided, most methods yield nearly random predictions. However, as the number of examples increases, ICRL performance improves significantly, indicating that the model gradually acquires the ability to interpret and leverage the injected representations when provided with sufficient contextual samples [1]. In this case, ICRL performance depends on two critical factors: (i) the diversity of representations, which enables meaningful mappings to labels, and (ii) the alignment of projected FM representations with the LLM's embedding distribution. Both conditions are essential for the model to effectively utilize the injected features and function optimally in the *task learning* mode.

In contrast, when examples incorporate both representations and text inputs, the model may shift towards a *task retrieval* mode [33] due to the presence of SMILES strings in the LLM's pretraining data. In this mode, predictions are guided by the interaction between the model's pre-training priors and the contextual examples, allowing it to effectively leverage prior knowledge when encountering familiar information [33]. This explains the inferior performance of the PCA+ICL method: the injected representations, being textual tokens, interfere with the model's interpretation of other text inputs, such as SMILES strings [15]. Conversely, injecting more distinctive embedding-level features, such as random noise, may function similarly to a pause token, allowing additional "thoughts" that improve performance [17]. These findings underscore the importance of the uniqueness of injected representations. By aligning more effectively with the model's retrieval mechanism, distinct representations enhance the model's ability to leverage its prior knowledge and improve performance.

# 6 Related Work

**Representation-based In-context Learning.** Recent work shows that LLMs can reorganize internal representations to capture task semantics purely from in-context examples [40]. Most related to our work, Vector-ICL [71], designs pre-training and finetuning to train projectors to align external models' representations into the LLM embedding space. In contrast, our proposed approach aims to study training-free methods to derive the projector to avoid costly supervised training.

**In-context Learning & Multi-modal LLMs.** ICL is a notable emergent property of LLMs [56], enabling them to perform tasks by conditioning on a few examples without requiring parameter updates [12]. MLLMs extend LLMs by integrating multiple data modalities [28, 65, 42], allowing for more comprehensive reasoning. The Appendix B provides more detailed related work.

# 7 Further Discussions

**Performance–Cost Trade-off.** To clarify the trade-off between accuracy and efficiency, we compare ICRL with representative fine-tuning pipelines, including instruction-tuning (I-FT), supervised pretraining (S-PT) + finetuning, and unsupervised pretraining (PT) + finetuning. While these methods achieve good performance, they typically require hours to weeks of GPU training. In contrast, ICRL is training-free and requires only a lightweight CPU alignment step of about 2 seconds.

Table 3: Performance–cost comparison between ICRL and recent fine-tuning pipelines.

| Method | Type | Resource | Training Time | ESOL (RMSE) | Lipo (RMSE) | Avg |
|---|---|---|---|---|---|---|
| MolecularGPT [36] | I-FT | 4×A800-80G | <1 day | 1.471 | 1.157 | 1.314 |
| GIMLET [67] | S-PT + FT | 2–4 GPUs | ∼1 day | 1.132 | 1.345 | 1.239 |
| SELFormer [64] | PT | 2×A5000 | ∼2 weeks | 1.357 | 3.192 | 2.275 |
| | PT + FT | 2×A5000 | ∼2 weeks | 0.682 | 1.005 | 0.844 |
| GPT-MolBERTa [5] | PT + FT | 2–4 GPUs | ∼2 weeks | 0.477±0.01 | 0.758±0.01 | 0.612 |
| OT-PCA (ours) | Training-free | CPU only | ∼2 sec | 1.140±0.01 | 1.349±0.01 | 1.245 |
| OT-PCA + ICL (ours) | Training-free | CPU only | ∼2 sec | 1.094±0.01 | 1.277±0.01 | 1.186 |

As shown in Table 3, while large-scale PT+FT methods achieve the lowest RMSE, they require substantial training cost. In contrast, ICRL runs in only a few seconds on CPU and still matches or even surpasses some lightweight tuning pipelines. (Detailed setup are provided in Appendix F.1.)

**Lightweight Trainable Projectors.** We also investigate whether lightweight training can enhance projector performance by exploring three strategies: caption-based pretraining on LPM24, contrastive alignment between SMILES and FM embeddings, and a combined multi-task variant. While these methods can improve captioning quality on the pretraining domain, they fail to transfer effectively to regression tasks such as ESOL and AqSolDB. In some cases, training even destabilizes the ICL process, leading to incomplete outputs or degraded accuracy. These findings indicate that, under limited data and without domain-specific supervision, lightweight trainable projectors are less reliable than the training-free OT-PCA, underscoring the robustness and practicality of our approach. (Implementation details and results are given in Appendix F.2.)

**Performance under Challenging Tasks.** Tasks such as drug–target interaction and protein-related property prediction are particularly challenging for ICL, where subtle sequence variations critically determine labels but remain hard for general-purpose LLMs to capture [2, 26], often leading to near-random performance. ICRL mitigates this limitation by injecting representation-based inputs, allowing the model to capture informative signals even when textual features are homogeneous.

Our additional experiments on molecular QA and caption generation confirm this trend: ICRL consistently outperforms ICL under difficult tasks, yet it still falls short of domain-specific expert models. This highlights ICRL as a practical and lightweight alternative when ICL struggles, while leaving room for future integration with specialized approaches (Appendix F.3).

**Cross-Modality Generalizability.** ICRL provides a lightweight framework for enabling LLMs to process non-text modalities such as vision and audio without requiring paired data or modality-specific training. Even in the absence of supervision, OT-aligned embeddings from models like ViT and wav2vec2 support meaningful few-shot predictions (Appendix F.4). While ICRL does not reach the performance of resource-intensive supervised methods in domains where large-scale task-specific or multimodal LLMs already exist, it serves as a practical alternative in scenarios where such models are scarce, costly, or difficult to deploy. Representative examples include molecular property prediction [59], sensor-based human activity recognition [30], and biomedical applications such as protein–ligand binding [45], where pretrained multimodal LLMs are not readily available.

In these settings, the lack of modality-specific pretrained models, limited labeled data, and domain complexity often make end-to-end fine-tuning infeasible. By directly injecting representation-level features, ICRL enables LLMs to exploit modality-specific information with no additional supervision or architectural modification. Importantly, our findings across molecular and protein datasets confirm that the link between representation diversity and downstream performance is *modality-agnostic*, underscoring ICRL's potential as a general approach for extending LLMs beyond text (Appendix F.7).

# 8 Conclusion

We propose ICRL to explore whether LLMs can leverage modality-specific FM features without supervised training. Results show its feasibility and potential for generalizable multimodal reasoning.

**Limitations and Future Work.** Despite its efficiency and generalizability, ICRL underperforms compared to supervised methods due to the lack of task-specific training. In future work, we plan to explore lightweight training strategies to improve its performance while preserving efficiency.

## Acknowledgement

This research is supported by the National Research Foundation, Singapore under its National Large Language Models Funding Initiative (AISG Award No: AISG-NMLP-2024-001), NTU Start Up Grant and by the Ministry of Education, Singapore, under its Academic Research Fund Tier 1 (RG22/24). Any opinions, findings and conclusions or recommendations expressed in this material are those of the author(s) and do not reflect the views of National Research Foundation, Singapore. The computational work for this article was partially performed on resources of the National Supercomputing Centre (NSCC), Singapore (https://www.nscc.sg).

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

## Appendix Contents

## A   Author Contributions

**Tianle Zhang:** Led the project; designed and implemented various representation-level ICRL variants; conducted most experiments and analysis; wrote the main manuscript.

**Wanlong Fang:** Assisted with vision-related experiments; contributed to project discussions and paper revisions.

**Jonathan Woo:** Designed and implemented text-level ICRL variants; developed the initial project codebase; contributed to project discussions and paper revisions; assisted in designing the theoretical analysis framework.

**Paridhi Latawa:** Prepared and processed protein-related datasets; contributed to project discussions and paper revisions.

**Deepak A. Subramanian:** Prepared and processed small-molecule-related datasets; contributed to project discussions and paper revisions.

**Alvin Chan:** Provided overall supervision and guidance throughout the project; advised on the ICRL framework; contributed to manuscript writing.

# B Detailed Related work

**In-Context Learning (ICL)**: ICL is a remarkable emergent property of LLMs, enabling them to perform various tasks by conditioning on a few input-output examples, without requiring parameter updates or fine-tuning [7, 55]. It has been explored extensively in terms of studying factors that influence the performance of ICL such as prompting strategies, amount and order of few-shot exemplars[50, 23, 19, 12]. However, it remains unknown how we can apply in-context learning to integrate high-dimensional representations from external non-LLM models. Our work is pioneering in its focus on enabling LLMs to directly learn from and utilize representations derived from diverse modalities beyond text from external models. This novel approach has the potential to radically expand the utility of LLMs in domains requiring the integration of multiple modalities and knowledge, thus opening new avenues for the application of LLMs. In a recent work [60], it has been shown that LLMs can shift from their original semantic representations to new ones that aligned with the graph's structure given as few-shot learning examples in the text prompt, suggesting that scaling context can enable LLMs to reorganize their knowledge to accommodate novel contexts and tasks. In contrast, our work here focuses on the potential to integrate representations from external models during inference time through few-shot exemplars.

**Foundation Models (FMs)**: Foundation Models are large-scale, pre-trained deep learning models designed to generalize across diverse tasks by learning from massive datasets using self-supervised learning techniques. These models, such as GPT3 [43, 7], leverage billions of parameters to develop general-purpose representations that can be fine-tuned for specific tasks or domains, revolutionizing fields like natural language processing, computer vision, and biomedical research [7]. For molecular and protein analysis, structure-based FMs, such as GearNet [66], Uni-Mol [69], and ESMFold [32] extend this paradigm by incorporating domain-specific 2D/3D molecular representations. Although user-friendly, LLMs typically lack specialized training in areas like protein sequences or chemical structures. Such specialized domains FMs offers a way to enable LLMs to efficiently acquire and apply specialized knowledge. Our work use ICL to integrate domain-specific expertise from FMs pre-trained on specialized datasets, enhancing both usability and computational efficiency.

**Tool Use in LLMs**: Current research on tool use with LLMs, such as the implementation of APIs for external functionalities in applications like Chemcrow, demonstrates the capacity of LLMs to interface with external tools to perform specific tasks [6, 62, 41]. However, in the current approaches, LLMs utilize only the final outputs of external models as the external tool. There is currently no work looking into using the richer, underlying representations from these external models for more sophisticated LLM inference. Our work aims to bridge this gap by enabling LLMs to access and leverage deep learning model representations, enhancing their inferential capabilities to address specialized tasks beyond what the LLMs are trained on.

**Multi-modal Large Language Models (MLLMs):** Multi-modal Large Language Models are designed to process and integrate multiple data types, including images, text, audio, and more [63, 43]. Existing research in the realm of MLLMs, such as Vision-Language Models (VLM) [3]and Tx-LLM[8], has significantly advanced our understanding of how MLLMs can process and integrate information across different input modalities. However, these models typically require extensive supervised training for both the LLM and their projection layers to handle multi-modal tasks [27]. This process not only demands substantial computational resources but also necessitates expertise in AI, making it less accessible to researchers without a background in machine learning. The requirement for multi-task training means the entire model must be retrained with both old and new data to avoid catastrophic forgetting—a significant limitation when only new data is introduced[39]. Our work addresses these limitations by proposing a novel in-context representation learning framework that simplifies the integration of multi-modal data, thereby reducing the need for extensive computational resources and specialized AI knowledge.

# C Datasets Details

## C.1 Molecular Dataset Description

**Molecular Domain:** In the molecular domain, we evaluate the performance of our ICRL framework on seven key molecular property prediction regression tasks. These tasks are taken from Therapeutic

Data Commons (TDC) Absorption, Distribution, Metabolism, and Excretion (ADME) property prediction benchmarks and the Toxicity property prediction benchmarks.

TDC is a platform consisting of AI-based datasets, benchmarks, and processing tools for therapeutic machine learning [21]. The ADME tasks are a set of single-instance prediction tasks in the molecular domain that examine the ADME properties of small-molecule drug chemicals. The Toxicity property prediction tasks aim to predict drug toxicity properties. All molecular datasets are as follows.

- ESOL [11]: ESOL is made up of water solubility data (log solubility in mols per litre) for common organic small molecules. Input a chemical compounds SMILES string, predict the solubility of chemical compounds.

- Caco-2 [54]: The human colon epithelial cancer cell line, Caco-2, is used as an in vitro model to simulate human intestinal tissue. Experimental measurements of the rate at which drugs pass through Caco-2 cells can approximate the rate of drug permeation through human intestinal tissue. Input a drug SMILES string, predict the Caco-2 cell effective permeability.

- AqSolDB [52]: Aqueous solubility measures a drug's ability to dissolve in water. Poor water solubility can lead to slow drug absorption, inadequate bioavailability, and even toxicity. More than 40% of new chemical entities are poorly soluble. Input a drug SMILES string, predict the activity of solubility.

- LD50Zhu [70]: Acute toxicity LD50 measures the most conservative dose that can lead to lethal adverse effects. The higher the dose, the more lethal of a drug. Input a drug SMILES string, predict its acute toxicity.

- AstraZeneca [58]: Lipophilicity measures a drug's ability to dissolve in a lipid environment (e.g., fats, oils). High lipophilicity is often associated with a high rate of metabolism, poor solubility, rapid turnover, and low absorption. Input a drug SMILES string, predict the activity of lipophilicity.

### C.2 Protein Dataset Description

**Protein Domain:** In the protein domain, we evaluate the performance of the ICRL framework on four key protein property prediction regression tasks. These tasks are taken from two key protein representation learning benchmarks: TAPE (Tasks Assessing Protein Embeddings) and PEER (Protein Sequence Understanding).

The TAPE benchmark consists of five tasks spanning different domains of protein biology [44]. We focused on the Fluorescence and Stability tasks because they are whole protein-sequence-level regression tasks, in contrast to amino acid-level regression, pairwise amino acid regression, or classification. The Fluorescence dataset and Stability dataset descriptions are as follows:

- Fluorescence: The Fluorescence task is a regression-based task that predicts the log-fluorescence intensity of an input protein [47]. The train set consists of a small neighborhood of the GFP protein, and the test set has distant GFP proteins.

- Stability: The Stability task is a regression task that measures the most extreme circumstances in which a protein maintains its folded state above a concentration threshold, considered a proxy for stability [46]. The train set comprises a broad set of proteins, and the test set comprises one-mutation neighborhoods of sampled proteins.

**Drug-Target Interaction Domain:**

In the Drug-Target Interaction (DTI) domain, we evaluate the performance of our framework on two key molecular-protein cross-domain property prediction regression tasks. These tasks are taken from the TDC Multi-Instance Prediction Problem. The drug-target interaction (DTI) prediction task evaluates the interaction activity of small-molecule drug compounds.

BindingDB[35, 22] is a collection of various assays which is a part of the DTI prediction task, which takes in a target amino acid sequence and SMILES string and then predicts the binding affinity. Since different assays use different metrics, two separate BindingDB datasets in Ki and IC50 units were used.

- BindingDB_Ki is a database containing experimental data on the binding affinity (Ki values) of small molecules to protein targets.

- BindingDB_IC50 is a publicly available database containing experimental data on the binding affinities of small molecules to proteins, specifically in terms of IC50 values. IC50 measures the concentration of a substance required to inhibit 50% of a biological target.

## D  Algorithmic descriptions

In this section, we provide a detailed description of the overall pipeline and all representation injection algorithms.

---

**Algorithm 1** ICRL Framework

---

1: **Input:** Training and test dataset $Q_{train}$ and $Q_{test}$, FM $\phi$, LLM $\psi_e$, batch query size $b$, injection strategy $M_{inj}$, number of examples $k$, empty example set $E$
2: **Stage 1: Construct demonstration set**
3: Extract representations from FM: $\mathbf{H} = \phi(Q_{train})$
4: **if** $M_{inj}$ requires additional parameters **then**
5:     Derive parameters from $\{x_i\}_{i=1}^n \subseteq Q_{train}$ and $\mathbf{H}$
6: **end if**
7: Sample $\{(x_i, y_i)\}_{i=1}^n$ from $Q_{train}$
8: **for** each sampled $(x_i, y_i, h_i)$ **do**
9:     Apply injection strategy: $r_i = M_{inj}(h_i)$
10:     Add example $(r_i, y_i)$ to $E$
11: **end for**
12: **Stage 2: Batch Inference**
13: **for** each test batch $B_j = \{(x_j^{(t)}, y_j^{(t)})\}_{t=1}^b \subseteq Q_{test}$ **do**
14:     Initialize empty batch queries $Q_j$
15:     **for** each sample $t = 1$ to $b$ **do**
16:         Extract representation: $h_j^{(t)} = \phi(x_j^{(t)})$
17:         Apply injection strategy: $r_j^{(t)} = M_{inj}(h_j^{(t)})$
18:         Add query $(r_j^{(t)})$ to $Q_j$
19:     **end for**
20:     Generate predictions: $\hat{Y}_j = \{\hat{y}_j^{(t)}\}_{t=1}^b = \psi(E, Q_j)$
21: **end for**
22: **Output:** Predictions $\hat{Y}$.

---

---

**Algorithm 2** PCA method

---

1: **Input:** Extracted representations $\mathbf{H}$, PCA model $\mathbf{W}_{PCA}$ fitted by $Q_{train}$ and string conversion function $S$
2: Compute reduced-dimensional embeddings: $\mathbf{H}_{pca} = \mathbf{H} \times \mathbf{W}_{PCA}$
3: Convert reduced embeddings into text strings: $\mathbf{S}_{pca} = S(\mathbf{H}_{pca})$
4: **Output:** $\mathbf{S}_{pca}$

---

---

**Algorithm 3** Zero-Pad

---

1: **Input:** Extracted representations $\mathbf{H}$, LLM embedding space dimension $d_{LLM}$
2: Compute padding size: pad_size $= d_{LLM} - d_{FM}$
3: **for** each $h_i$ in $\mathbf{H}$ **do**
4:     Apply zero padding to $\mathbf{H}$ to match the LLM embedding dimension: $\mathbf{h_i}_{padded} = \text{Pad}(\mathbf{h_i}, \text{pad\_size})$
5: **end for**
6: **Output:** $\mathbf{H}_{padded}$

---

**Algorithm 4** Random Noise

---

1: **Input:** Training and test dataset $Q_{train}$, LLM $\psi_e$, LLM embedding dimension $d_{LLM}$
2: **for** each $x_i$ in $Q_{train}$ **do**
3:     Compute a random hash seed using $x_i$: $h_{i_{seed}} = \text{Hash}(x_i)$
4:     Generate a unique random vector for each input based on the hash seed: $\mathbf{h}_i = \text{Random}(h_{i_{seed}}, d_{LLM})$
5: **end for**
6: **Output:** $\mathbf{H}_{noise}$

---

**Algorithm 5** Optimal Transport Alignment

---

1: **Input:** Extracted and projected representations $\mathbf{H}$ and $\mathbf{H}_{proj}$, input texts $Q_{train}$, LLM embedding function $\psi_e$, alignment mode $mode$, PCA model $\mathbf{W}_{PCA}$ fitted by $Q_{train}$ and string conversion function $S$
2: **Stage 1: Get the OT parameters**
3: **if** $mode = \text{``}embed\text{''}$ **then**
4:     Compute input embeddings: $\mathbf{H}_{tar} = \psi_e(Q_{train})$
5: **else if** $mode = \text{``}pca\text{''}$ **then**
6:     Get PCA strings: $\mathbf{S}_{pca} = S(\mathbf{H} \times \mathbf{W}_{PCA})$
7:     Compute PCA embeddings: $\mathbf{H}_{tar} = \psi_e(\mathbf{S}_{pca})$
8: **end if**
9: Initialize empty vectors $shift, scale \in \mathbb{R}^{d_{LLM}}$
10: **for** each dimension $j$ **do**
11:     $\bar{\mathbf{u}}_j = mean(\mathbf{H}_{proj}[:,j])$, $\bar{\mathbf{v}}_j = mean(\mathbf{H}_{tar}[:,j])$
12:     $\sigma_{p,j} = std(\mathbf{H}_{proj}[:,j])$, $\sigma_{t,j} = std(\mathbf{H}_{tar}[:,j])$
13:     $shift[j] = \bar{\mathbf{v}}_j - \bar{\mathbf{u}}_j$, $scale[j] = \sigma_{t,j}/\sigma_{p,j}$
14: **end for**
15: **Stage 2: Implement alignment**
16: $\mathbf{H}_{aligned} = scale \cdot \mathbf{H}_{proj} + shift$
17: **return** $\mathbf{H}_{aligned}, shift, scale$

---

# E Detailed Proofs

In this section, we provide a complete derivation of the concentration result stated in Theorem 1 and Theorem 2. For convenience, we restate the theorem and then present the step-by-step analysis.

## E.1 Proof of Theorem 1

**theorem**: Given a fixed vector $\mathbf{u} \in \mathbb{R}^d$, suppose $\mathbf{W} \in \mathbb{R}^{d \times d}$ has i.i.d. entries $\mathbf{W}_{k,l} \sim \mathcal{N}(0, 1/d)$, and $b \in \mathbb{R}^d$ is a bias vector (fixed or random and independent of $\mathbf{W}$). Then there exist small constants $\epsilon_1, \delta_1 > 0$ such that, with probability at least $1 - \delta_1$,

$$\left| \|\mathbf{W}\mathbf{u} + \mathbf{b}\|^2 - (\|\mathbf{b}\|^2 + \|\mathbf{u}\|^2) \right| \leq \epsilon_1 \left( \|\mathbf{b}\|^2 + \|\mathbf{u}\|^2 \right). \tag{8}$$

### E.1.1 Step 1: Coordinate-wise Distribution

Consider the affine transformation $\mathbf{x} \mapsto \mathbf{W}\mathbf{x} + \mathbf{b}$. For each coordinate $k \in \{1, 2, \ldots, d\}$,

$$(\mathbf{W}\mathbf{u} + \mathbf{b})_k = \sum_{\ell=1}^{d} \mathbf{W}_{k,\ell}\, \mathbf{u}_\ell + \mathbf{b}_k. \tag{9}$$

Because $\mathbf{W}_{k,\ell} \sim \mathcal{N}(0, 1/d)$ are i.i.d., the sum $\sum_{\ell=1}^{d} \mathbf{W}_{k,\ell}\, \mathbf{u}_\ell$ is Gaussian with mean 0 and variance $\frac{\|\mathbf{u}\|^2}{d}$. Hence each coordinate $(\mathbf{W}\mathbf{u} + \mathbf{b})_k$ is distributed as

$$\mathbf{Y}_k := (\mathbf{W}\mathbf{u} + \mathbf{b})_k \sim \mathcal{N}\left( \mathbf{b}_k, \frac{\|\mathbf{u}\|^2}{d} \right). \tag{10}$$

### E.1.2 Step 2: Expectation and First Moments of the Squared Norm

Let $\mathbf{Z} = \|\mathbf{W}\mathbf{u} + \mathbf{b}\|^2 = \sum_{k=1}^{d} \mathbf{Y}_k^2$, since $\mathbf{Y}_k \sim \mathcal{N}(\mathbf{b}_k, \frac{\|\mathbf{u}\|^2}{d})$, for each $k$, we have $\mathbb{E}[\mathbf{Y}_k^2] = \mathbf{b}_k^2 + \frac{\|\mathbf{u}\|^2}{d}$, summation over $k$ yields:

$$\mathbb{E}[\mathbf{Z}] = \sum_{k=1}^{d} \left( \mathbf{b}_k^2 + \frac{\|\mathbf{u}\|^2}{d} \right) = \|\mathbf{b}\|^2 + \|\mathbf{u}\|^2. \tag{11}$$

Denote this sum by

$$M = \|\mathbf{b}\|^2 + \|\mathbf{u}\|^2, \tag{12}$$

our task is to show that $\mathbf{Z}$ remains close to $M$ with high probability as $d$ grows.

### E.1.3 Step 3: Concentration of the Squared Norm (Refined Analysis)

We now give a more precise argument about the variance of $\mathbf{Z}$ and how it influences the bound from Chebyshev's inequality.

**3.1. Exact Variance Computation.** Since $\mathbf{Z} = \sum_{k=1}^{d} \mathbf{Y}_k^2$ and each $\mathbf{Y}_k$ is independent across $k$: $\mathrm{Var}(\mathbf{Z}) = \sum_{k=1}^{d} \mathrm{Var}(\mathbf{Y}_k^2)$. For $\mathbf{Y}_k \sim \mathcal{N}(\mu_k, \sigma^2)$ with $\mu_k = \mathbf{b}_k$ and $\sigma^2 = \|\mathbf{u}\|^2/d$, one has

$$\mathrm{Var}(\mathbf{Y}_k^2) = \mathbb{E}[\mathbf{Y}_k^4] - (\mathbb{E}[\mathbf{Y}_k^2])^2 = 2\,\sigma^4 + 4\,\mu_k^2 \sigma^2. \tag{13}$$

Substituting $\mu_k = \mathbf{b}_k$ and $\sigma^2 = \|\mathbf{u}\|^2/d$, we get

$$\mathrm{Var}(\mathbf{Y}_k^2) = 2 \left( \frac{\|\mathbf{u}\|^2}{d} \right)^2 + 4 \left( \frac{\|\mathbf{u}\|^2}{d} \right) (\mathbf{b}_k^2). \tag{14}$$

Summing over $k$ from 1 to $d$,

$$\mathrm{Var}(\mathbf{Z}) = \sum_{k=1}^{d} \left( 2 \frac{\|\mathbf{u}\|^4}{d^2} + 4 \frac{\|\mathbf{u}\|^2}{d} \mathbf{b}_k^2 \right) = \frac{2 \|\mathbf{u}\|^4}{d} + \frac{4 \|\mathbf{u}\|^2 \|\mathbf{b}\|^2}{d}. \tag{15}$$

Thus we have:

$$\mathrm{Var}(\mathbf{Z}) = \frac{2 \|\mathbf{u}\|^4 + 4 \|\mathbf{u}\|^2 \|\mathbf{b}\|^2}{d}, \quad \mathbb{E}[\mathbf{Z}] = M = \|\mathbf{u}\|^2 + \|\mathbf{b}\|^2. \tag{16}$$

**3.2. Ratio of Variance to Square of the Mean.** To apply Chebyshev precisely, we look at

$$\frac{\mathrm{Var}(\mathbf{Z})}{(\mathbb{E}[\mathbf{Z}])^2} = \frac{2\|\mathbf{u}\|^4 + 4\|\mathbf{u}\|^2\|\mathbf{b}\|^2}{d(\|\mathbf{u}\|^2 + \|\mathbf{b}\|^2)^2}. \tag{17}$$

Provided $\|\mathbf{u}\|$ and $\|\mathbf{b}\|$ do not scale with $d$, this ratio shrinks on the order of $1/d$. Thus, the relative standard deviation $\sqrt{\mathrm{Var}(\mathbf{Z})}/\mathbb{E}[\mathbf{Z}]$ behaves like $1/\sqrt{d}$. Hence, as $d \to \infty$, $\mathbf{Z}$ concentrates around its mean $M$.

**3.3. Chebyshev's Inequality for $\epsilon_1, \delta_1$.** From Chebyshev's inequality:

$$\Pr\big(|\mathbf{Z} - \mathbb{E}[\mathbf{Z}]| \geq \alpha\big) \leq \frac{\mathrm{Var}(\mathbf{Z})}{\alpha^2}. \tag{18}$$

Set $\alpha = \epsilon_1(\|\mathbf{u}\|^2 + \|\mathbf{b}\|^2) = \epsilon_1 M$. Then

$$\Pr\Big(|\mathbf{Z} - M| \geq \epsilon_1 M\Big) \leq \frac{\mathrm{Var}(\mathbf{Z})}{\epsilon_1^2 M^2} = \frac{2\|\mathbf{u}\|^4 + 4\|\mathbf{u}\|^2\|\mathbf{b}\|^2}{d(\|\mathbf{u}\|^2 + \|\mathbf{b}\|^2)^2 \epsilon_1^2} = \frac{C_{u,b}}{d\,\epsilon_1^2}, \tag{19}$$

where

$$C_{u,b} = \frac{2\|\mathbf{u}\|^4 + 4\|\mathbf{u}\|^2\|\mathbf{b}\|^2}{(\|\mathbf{u}\|^2 + \|\mathbf{b}\|^2)^2}. \tag{20}$$

Hence, for any fixed $\epsilon_1$, the failure probability $\delta_1$ satisfies

$$\delta_1 = \Pr\Big(|\mathbf{Z} - M| \geq \epsilon_1 M\Big) \leq \frac{C_{u,b}}{d\,\epsilon_1^2}. \tag{21}$$

As $d$ increases, $\delta_1 \to 0$. This establishes a high-probability guarantee of the form

$$\Pr\Big(\big|\|\mathbf{W}\mathbf{u} + \mathbf{b}\|^2 - (\|\mathbf{u}\|^2 + \|\mathbf{b}\|^2)\big| \leq \epsilon_1(\|\mathbf{u}\|^2 + \|\mathbf{b}\|^2)\Big) \geq 1 - \delta_1, \tag{22}$$

with explicit constants that depend on $\|\mathbf{u}\|$, $\|\mathbf{b}\|$, and $d$.

**3.4. Uniform Dependence on $\|\mathbf{u}\|$ and $\|\mathbf{b}\|$.** If $\|\mathbf{u}\|$ or $\|\mathbf{b}\|$ grows with $d$, the coefficient $C_{u,b}$ might also grow, thus weakening the rate at which $\delta_1$ decays. However, if $\|\mathbf{u}\|$ and $\|\mathbf{b}\|$ remain bounded independently of $d$, then $C_{u,b}$ is constant, and the probability of failure vanishes on the order of $1/d$. This completes our proof of Theorem 1

**Remarks.**

- The key structural property is that $\mathbb{E}[\mathbf{W}^T\mathbf{W}] = I_d$, similar statements hold under any isotropic, sub-Gaussian distribution for $\mathbf{W}$. This also explains why the normal initialization achieves the best performance, since it is the one that best fits the theoretical assumptions.

- If $b$ is itself random and independent of $\mathbf{W}$, conditioning on one and integrating out the other yields a nearly identical analysis; the main requirement is that $b$ not be too large or adversarially correlated with $\mathbf{W}$, e.g., in practice, we avoid this problem by setting $b$ directly to 0.

**E.2  Proof of Theorem 2**

theorem: Let $\mathbf{u}, \mathbf{v} \in \mathbb{R}^d$ be any two fixed vectors, and let $\mathbf{W} \in \mathbb{R}^{d \times d}$ have i.i.d. entries $\mathbf{W}_{k,l} \sim \mathcal{N}(0, 1/d)$. Define $\mathbf{u}' = \mathbf{W}\mathbf{u}$, $\mathbf{v}' = \mathbf{W}\mathbf{v}$. Then there exist constants $\epsilon_2, \delta_2 > 0$ such that, with probability at least $1 - \delta_2$,

$$\big|\cos(\mathbf{u}', \mathbf{v}') - \cos(\mathbf{u}, \mathbf{v})\big| \leq \epsilon_2. \tag{23}$$

Equivalently,

$$\cos(\mathbf{W}\mathbf{u}, \mathbf{W}\mathbf{v}) \approx \cos(\mathbf{u}, \mathbf{v}) \quad \text{with high probability.} \tag{24}$$

### E.2.1 Step 1: Concentration of norms.

By Theorem 1, for each of the vectors $\mathbf{u}$ and $\mathbf{v}$, their images under $\mathbf{x} \mapsto \mathbf{Wx}$ satisfy

$$\|\mathbf{Wu}\|^2 \approx \|\mathbf{u}\|^2, \qquad \|\mathbf{Wv}\|^2 \approx \|\mathbf{v}\|^2. \tag{25}$$

More precisely, there exist $\epsilon_1, \delta_1 > 0$ such that with probability at least $1 - \delta_1$,

$$\left| \|\mathbf{Wu}\|^2 - \|\mathbf{u}\|^2 \right| \leq \epsilon_1 \|\mathbf{u}\|^2, \quad \left| \|\mathbf{Wv}\|^2 - \|\mathbf{v}\|^2 \right| \leq \epsilon_1 \|\mathbf{v}\|^2. \tag{26}$$

Taking square roots and using standard bounds yields

$$\|\mathbf{u'}\| = \|\mathbf{Wu}\| \approx \|\mathbf{u}\|, \quad \|\mathbf{v'}\| = \|\mathbf{Wv}\| \approx \|\mathbf{v}\|. \tag{27}$$

### E.2.2 Step 2: Concentration of the dot product.

Consider $(\mathbf{Wu})^\top (\mathbf{Wv}) = \mathbf{u'}^\top \mathbf{v'}$, we have:

$$\mathbf{u'}^\top \mathbf{v'} = (\mathbf{Wu})^\top (\mathbf{Wv}). \tag{28}$$

Since $\mathbf{W}$ has mean-zero i.i.d. Gaussian entries, we have:

$$\mathbb{E}[(\mathbf{Wu})^\top (\mathbf{Wv})] = \mathbf{u}^\top \mathbf{v}. \tag{29}$$

By concentration inequalities for sums of Gaussians, there exist $\epsilon_2', \delta_2' > 0$ such that with probability at least $1 - \delta_2'$,

$$\left| (\mathbf{Wu})^\top (\mathbf{Wv}) - \mathbf{u}^\top \mathbf{v} \right| \leq \epsilon_2' \|\mathbf{u}\| \|\mathbf{v}\|. \tag{30}$$

### E.2.3 Step 3: Combining the bounds to preserve cosine similarity.

For $\mathbf{u'}$ and $\mathbf{v'}$, we have $\cos(\mathbf{u'}, \mathbf{v'}) = \frac{\mathbf{u'}^\top \mathbf{v'}}{\|\mathbf{u'}\| \|\mathbf{v'}\|}$. From (27) and (30), with high probability, we have:

$$\mathbf{u'}^\top \mathbf{v'} \approx \mathbf{u}^\top \mathbf{v}, \quad \|\mathbf{u'}\| \approx \|\mathbf{u}\|, \quad \|\mathbf{v'}\| \approx \|\mathbf{v}\|. \tag{31}$$

Thus we have:

$$\cos(\mathbf{u'}, \mathbf{v'}) = \frac{\mathbf{u'}^\top \mathbf{v'}}{\|\mathbf{u'}\| \|\mathbf{v'}\|} \approx \frac{\mathbf{u}^\top \mathbf{v}}{\|\mathbf{u}\| \|\mathbf{v}\|} = \cos(\mathbf{u}, \mathbf{v}). \tag{32}$$

Therefore,

$$\left| \cos(\mathbf{Wu}, \mathbf{Wv}) - \cos(\mathbf{u}, \mathbf{v}) \right| \leq \epsilon_2 \tag{33}$$

for some $\epsilon_2 > 0$, with probability at least $1 - \delta_2$, where $\delta_2 := \delta_1 + \delta_2'$, thus we complete the proof.

### E.3 Proof of Corollary 1

Below, we present a more formulaic argument showing that a random Gaussian linear map $\mathbf{x} \mapsto \mathbf{Wx}$ followed by a standard nonlinear activation $\sigma : \mathbb{R} \to \mathbb{R}$ (e.g., ReLU, leaky ReLU, sigmoid, etc.) leads to *larger angle distortion* than the purely linear case. In other words, with high probability, the angle distortion $\epsilon_{2,\sigma}$ for nonlinearly transformed embeddings is larger than the purely linear case $\epsilon_{2,\text{lin}}$.

**Coordinate-Level Analysis.** Consider a single coordinate, $z_1 = (\mathbf{Wu})_k, z_2 = (\mathbf{Wv})_k$. The original product is $z_1 z_2$, whereas after applying $\sigma$ we have $\sigma(z_1) \sigma(z_2)$. If $\sigma$ compresses negative values more strongly than positive ones (*e.g.*, ReLU, sigmoid), or otherwise saturates different portions of the real line, then $\sigma(z_1) \sigma(z_2)$ can deviate significantly from $z_1 z_2$.

Taking ReLU as an example, the $z_1 z_2$ and $\sigma(z_1) \sigma(z_2)$ will differ by

$$\Delta_k = \max\{z_1, 0\} \max\{z_2, 0\} - z_1 z_2. \tag{34}$$

If $z_1$ and $z_2$ have opposite signs, $\Delta_k > 0$; this pushes the new vectors into the positive orthant and tends to align them more.

**Closed-Form for Correlated Gaussians.** Due to the properties of random Gaussian matrices, for any coordinate $k$, $(z_1, z_2)$ follows a bivariate normal with correlation $\rho > 0$, i.e. $(Z_1, Z_2) \sim \mathcal{N}(\mathbf{0}, \Sigma)$,

where $\Sigma = \begin{pmatrix} 1 & \rho \\ \rho & 1 \end{pmatrix}$ and $0 < \rho < 1$. Taking ReLU as an example, we can define $Z_{1+} = \max\{Z_1, 0\}$ and $Z_{2+} = \max\{Z_2, 0\}$, thus we have:

$$\mathrm{Corr}(Z_{1+}, Z_{2+}) = \frac{\mathbb{E}[Z_{1+} Z_{2+}]}{\sqrt{\mathbb{E}[Z_{1+}^2]\,\mathbb{E}[Z_{2+}^2]}}. \tag{35}$$

A standard integral shows $\mathbb{E}[Z_+^2] = \frac{1}{2}$, so we can simplify the definition as: $\mathrm{Corr}(Z_{1+}, Z_{2+}) = 2\,\mathbb{E}[Z_{1+} Z_{2+}]$. When $\rho = 0$, symmetry implies $\mathbb{E}[Z_{1+} Z_{2+}] = \frac{1}{4}$, giving a correlation of $\frac{1}{2}$. As $\rho \to 1$, $\mathbb{E}[Z_{1+} Z_{2+}] \to \frac{1}{2}$, so the correlation approaches 1. For $0 < \rho < 1$, as $\mathbb{E}[Z_{1+} Z_{2+}]$ *increases* with $\rho$, staying above $\frac{1}{4}$ and thus making: $\mathrm{Corr}(Z_{1+}, Z_{2+}) > \rho$. As correlation in $\mathbb{R}^d$ directly relates to cosine similarity, we conclude that

$$\cos\big(\max\{Z_1, 0\}, \max\{Z_2, 0\}\big) > \cos(Z_1, Z_2), \tag{36}$$

i.e. ReLU pushes already-aligned (positively correlated) coordinates even closer together, leading to "angle inflation."

**Summary and Conclusion (for General $\sigma$).** Because generic activations $\sigma$ (e.g. ReLU, leaky ReLU, sigmoid, tanh) discard or shrink certain parts of the real line (often negative values), they systematically inflate pairwise similarities in a high-dimensional random setting. Therefore, we have

$$\big|\cos\big(\sigma(\mathbf{Wu}), \sigma(\mathbf{Wv})\big) - \cos(\mathbf{u}, \mathbf{v})\big| > \big|\cos(\mathbf{Wu}, \mathbf{Wv}) - \cos(\mathbf{u}, \mathbf{v})\big|, \tag{37}$$

meaning the angle distortion $\epsilon_{2,\sigma}$ for such non-linearities exceeds the purely linear case $\epsilon_{2,\mathrm{lin}}$.

# F More Experiments

## F.1 Performance-Efficiency Trade-offs in ICRL

A key motivation behind ICRL is to explore the feasibility of enabling LLMs to understand non-text modalities (e.g., molecular, vision, audio) in a training-free manner. This contrasts with prior methods such as Vector-ICL [71], Florence-VL [9], and ICL-Reps [61], which rely on supervised training or projection tuning. In this section, we provide a comprehensive analysis of the trade-offs between performance and efficiency across model scale, input modalities, and computational cost.

**Comparison with Similar Methods.** As shown in Table 4, ICRL is unique in being both training-free and capable of accepting non-text inputs. In contrast, MLLMs (Florence-VL) and Vector-ICL require large-scale supervised training or projection tuning. These approaches also consume substantially more resources in terms of data and time.

Table 4: Comparative overview of multimodal adaptation methods.

| Method | Non-text Input | Supervised Training | Data Requirements | Time Cost |
|---|---|---|---|---|
| MLLM [9] | ✓ | ✓ | High | High |
| Vector-ICL [71] | ✓ | ✓ | High | High |
| ICL-Reps [61] | ✓ | ✓ | High | High |
| DA-ICL [37] | ✗ | ✓ | High | High |
| ICL (text only) | ✗ | ✗ | Low | Low |
| **ICRL (Ours)** | ✓ | ✗ | Low | Low |

**Cost–Performance Comparison with Fine-Tuning Pipelines.** We compare ICRL against recent molecular property prediction pipelines, including instruction-tuning via Q-LoRA (I-FT) [36], supervised pretraining followed by finetuning (S-PT + FT) [67], and unsupervised pretraining followed by finetuning (PT + FT) [64, 5]. Evaluation is conducted on ESOL and Lipophilicity datasets, with RMSE as the metric. To ensure fairness, we implement ICRL using the LLaMA-3.1-8B-Instruct model—comparable in inference cost to those used in prior work and feasible on a single GPU. Evaluation is based on RMSE. Notably, ICRL achieves even better performance with larger models; e.g., on ESOL, it reaches 0.839 RMSE using LLaMA-3.1-70B. Cost information for baselines is extracted from original papers when available.

**Analysis.** As shown in Table 3, under comparable inference cost, ICRL achieves performance comparable to or even superior to lightweight tuning, while requiring *no training* and only $\sim2$ seconds of CPU computation. Although large-scale PT+FT pipelines (e.g., GPT-MolBERTa) still achieve the lowest RMSE, their training requires weeks on multiple GPUs, in sharp contrast to the negligible overhead of ICRL. This result highlights ICRL's practicality for low-resource and rapid-deployment scenarios, where retraining is prohibitive.

Improved understanding of small molecules does not necessarily translate to better performance on downstream tasks. In analogy to LLM pretraining, the pretraining in baseline methods is primarily intended to enhance molecular understanding. However, as shown in [64], the standalone pretrained model performs significantly worse than ICRL on downstream prediction, suggesting that pretraining alone mainly serves as a warm-up for subsequent supervised tuning.

**Context Window Usage.** One of the primary advantages of ICRL is its low context window usage. As summarized in Table 5, standard ICL methods require dozens to hundreds of tokens to represent non-text samples such as SMILES strings, amino acid sequences, images, or audio. In contrast, all variants of representation-level ICRL compress the input into a single embedding vector, which is injected as a single token—dramatically reducing input length and improving inference efficiency.

Table 5: Approximate per-sample context window usage across methods and modalities.

| Method | SMILES (ICL) | AA Sequence (ICL) | Vision (MLLM) | Audio (MLLM) | ICRL (Ours) |
|---|---|---|---|---|---|
| **Tokens** | 6–21 | 103–214 | 29–576 | 75–1500 | **1** |

**Inference Overhead.** Although ICRL includes additional steps such as PCA and OT alignment, these operations are lightweight. Table 6 shows that the PCA and OT steps collectively take about 2 seconds on CPU—faster than a single forward pass. Importantly, OT alignment only needs to be computed once per domain and reused thereafter via a simple matrix multiplication during inference.

Table 6: Comparison of computational cost across adaptation strategies.

| Method | Pre-training | Fine-tuning | PCA Step | OT Step |
|---|---|---|---|---|
| **Time Cost** | Days–Weeks | Hours–Days | $\sim$1.2 sec | $\sim$0.9 sec |
| **GPU Requirement** | High | Medium | None | None |

**Conclusion.** While ICRL may not achieve state-of-the-art performance in all tasks, it provides a promising direction for enabling lightweight, training-free multimodal reasoning. By drastically reducing context window usage and inference cost, it offers an efficient alternative to conventional multimodal systems, especially under low-resource constraints.

### F.2 Lightweight Learnable Projectors

To further examine whether lightweight training can improve projector performance, we conduct all projector training on the LPM24 dataset [14]. We design three loss variants: (1) *caption-based pretraining*, where the projector is optimized to generate molecular descriptions from FM embeddings; (2) *contrastive learning*, inspired by CLIP [43], which minimizes the distance between LLM hidden states when receiving molecular inputs in SMILES form versus projected FM embeddings; and (3) a *combined* loss that integrates both caption and contrastive objectives. This setup ensures that the LLM learns to treat projected embeddings as semantically consistent with textual SMILES representations, while also testing whether multi-task training enhances generalization.

All experiments use LLaMA-3.1-8B-Instruct as the base model, with a single-layer linear projector of hidden size 4096, consistent with [71]. Performance is evaluated on two tasks: molecular captioning on the LPM24 test set using BLEU-4, ROUGE-1, and ROUGE-L; and molecular property prediction on ESOL and Solubility_AqSolDB using RMSE. Other hyperparameters and evaluation protocols follow the main paper to ensure comparability.

**Analysis.** All three projector-training strategies perform worse than the training-free OT-PCA baseline. In regression tasks, they yield higher RMSE, and in some cases also harm ICL behavior

Table 7: Molecular captioning results with learnable projectors.

| Method | BLEU-4 | ROUGE-1 | ROUGE-L |
|---|---|---|---|
| Caption-only | 35.29 | 0.551 | 0.373 |
| Contrastive-only | 21.42 | 0.320 | 0.237 |
| Caption + Contrastive | 37.31 | 0.592 | 0.369 |

Table 8: Regression results with learnable projectors.

| Method | ESOL (RMSE) | AqSolDB (RMSE) |
|---|---|---|
| Caption-only | 1.256 | 3.030 |
| Contrastive-only | 1.372 | 2.915 |
| Caption + Contrastive | 1.213 | 3.805 |
| OT-PCA (ours) | **1.140** | **2.411** |
| OT-PCA + ICL (ours) | **1.094** | **2.385** |

(e.g., incomplete outputs, overfitting). Even when captioning quality improves, this does not transfer to downstream prediction. These results indicate that, under limited data and lack of domain knowledge, lightweight training is unstable and cannot match the robustness of training-free alignment.

### F.3 Cross-Tasks Generalizability of ICRL

To examine whether ICRL generalizes beyond regression tasks, we extend evaluation to two additional task types: molecular QA and captioning. For QA, we adopt the MoleculeQA benchmark [38], which contains four categories (Structure, Source, Property, Application), and report accuracy in each as well as the average score. For captioning, we use the ChEBI-20 dataset [13], where performance is measured with BLEU-4, ROUGE-1, and ROUGE-L. All experiments are conducted with LLaMA-3.1-8B-Instruct as the base model, baseline results in the table are taken from [38] and [13], respectively.

Table 9: Molecular QA results on MoleculeQA benchmark.

| Method | Structure | Source | Property | Application | Avg |
|---|---|---|---|---|---|
| Llama-2-7B-chat (L-FT) | 28.75 | 39.84 | 31.33 | 27.71 | 31.54 |
| ICL | 35.03 | 27.04 | 24.62 | 28.69 | 28.85 |
| OT-PCA (ours) | 51.32 | 37.66 | 33.71 | 31.02 | 38.43 |
| OT-PCA + ICL (ours) | 50.60 | 43.52 | 23.47 | 29.97 | 36.89 |
| MolT5-base (FT) | 58.01 | 65.85 | 45.14 | 42.24 | 55.39 |

**Analysis.** On the QA benchmark, ICRL significantly outperforms both standard ICL and fine-tuned general-purpose LLMs, showing that OT-aligned embeddings can be directly interpreted and utilized by a text-based LLM. Notably, embedding-only injection sometimes surpasses the combination with textual features, suggesting that the injected representations can provide a clearer and more informative signal than raw SMILES strings for molecular reasoning.

On the captioning task, ICRL also improves over ICL by enhancing the model's ability to generate semantically relevant outputs, despite the base LLM lacking molecular grounding. However, the performance gap with expert models such as MolT5 remains, indicating that while ICRL extends to generative tasks, domain-specific pretraining is still advantageous in settings requiring highly specialized knowledge.

### F.4 Cross-Modal Generalizability of ICRL

To further examine the generalizability of ICRL beyond molecular and protein datasets, we conducted experiments on vision and audio datasets, which are more distant from the natural language domain. These modalities do not natively conform to sequence-based formats, making ICL particularly challenging.

Table 10: Molecular captioning results on ChEBI-20 dataset.

| Method | BLEU-4 | ROUGE-1 | ROUGE-L |
|---|---|---|---|
| ICL | 0.133 | 0.393 | 0.310 |
| OT-PCA (ours) | 0.147 | 0.353 | 0.274 |
| OT-PCA + ICL (ours) | 0.196 | 0.407 | 0.353 |
| MolT5-base (FT) | 0.457 | 0.634 | 0.578 |

We evaluated ICRL on four datasets: ImageNet and CIFAR-100 (vision), ESC-50 and VGGSound (audio). Representations were extracted using ViT-B/16 for vision and wav2vec2-base-960h for audio. Due to computational constraints, we randomly selected 50 classes per dataset and followed the default settings described in the main paper. Because LLaMA-3-70B-Instruct does not support direct image/audio input, standard ICL is not applicable. Hence, we use random guessing as a baseline for comparison. All experiments were repeated 10 times with different random seeds, and we report the top-3 classification accuracy averaged across runs.

As shown in Table 11, naive strategies such as random noise or zero padding performed worse than random guessing, suggesting that LLMs cannot interpret these representations. In contrast, OT-based embeddings consistently outperformed random guessing, demonstrating that such embeddings preserve enough structure to be understood by LLMs in a training-free manner.

Table 11: Top-3 classification accuracy (%) on vision and audio datasets using LLaMA-based ICRL. OT-based embeddings clearly outperform baselines such as random noise or zero padding, indicating effective cross-modal generalization.

| Dataset | Random Guess | Random Noise | Zero Pad | OT-Embed | OT-PCA |
|---|---|---|---|---|---|
| ImageNet | 2.00 | 1.01 | 0.32 | 14.21 | 17.21 |
| CIFAR-100 | 2.00 | 0.63 | 0.91 | 13.73 | 15.62 |
| ESC-50 | 2.00 | 1.39 | 1.26 | 17.65 | 16.75 |
| VGGSound | 2.00 | 0.76 | 1.44 | 12.39 | 18.23 |

## F.5 Visualization of attentional weights

Figs. 5a and 5b present attention-weight heatmaps for the Ran-Pro+ICL and ICL methods, using the 20th attention head of the final layer as an example. The results show that the model's attention is predominantly concentrated on the SMILES string, suggesting that the injected representations are not the primary focus of the model's learning process. We set the shot count to 5 and batch query size to 1 in this experiment.

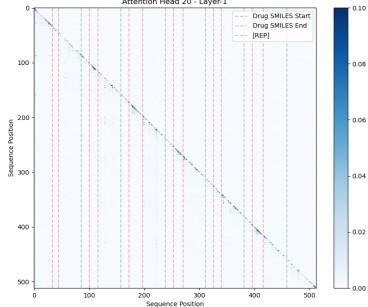
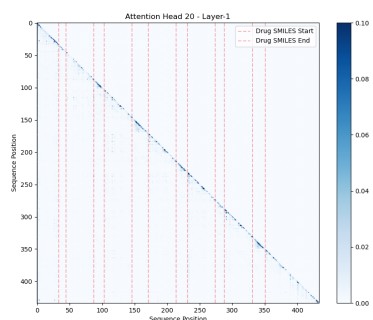

(a)        (b)

Figure 5: (a) and (b) are the attention-weight heatmaps for the Ran-Pro+ICL and ICL methods on the ESOL dataset, respectively.

## F.6 Feature extraction for different layers of FM

In this study, we primarily utilize CLS features derived from the FM. However, previous research has shown that using CLS features may not be the optimal choice for improving downstream task performance [51]. As shown in Table 12, under the OT-PCA method, most representations from other layers are difficult to interpret, possibly due to their higher similarity. Even the OT method struggles to effectively adjust these representations. However, when combined with ICL, using features from shallower layers typically yields better performance, suggesting that these features exhibit lower similarity to textual embeddings and can serve as more effective identity tokens.

In the ablation study of PCA, we observed that increasing the length of injected representations significantly impacts the performance of ICRL. Therefore, we further explored the effect of this factor on non-CLS features. As shown in Table 13, increasing the length did not lead to significant performance improvements. This indicates that representation length is not a critical factor for embedding-level injection methods. Moreover, longer representations may weaken their role as identity tokens, leading to performance degradation when combined with ICL.

Table 12: OT-PCA and OT-PCA+ICL across different layers (one layer) on ESOL. **Bold** = best value (highest for Pearson/Spearman, lowest for RMSE); Underline = second-best.

| Method | Layer | Pearson | Spearman | RMSE |
|---|---|---|---|---|
| OT-PCA | 0 | -0.017 ± 1.6e-2 | 0.010 ± 2.1e-2 | 1.432 ± 1.2e-2 |
| | 1 | 0.051 ± 4.0e-4 | 0.102 ± 6.2e-3 | 1.350 ± 1.9e-2 |
| | 5 | 0.078 ± 3.2e-4 | 0.148 ± 3.7e-4 | 1.235 ± 4.1e-4 |
| | 10 | 0.052 ± 4.0e-3 | 0.101 ± 5.9e-3 | 1.336 ± 2.1e-2 |
| | -1 | 0.078 ± 8.6e-4 | 0.122 ± 2.8e-3 | **1.220 ± 5.7e-3** |
| | cls | **0.227 ± 7.3e-4** | **0.235 ± 1.5e-4** | 1.243 ± 4.0e-3 |
| OT-PCA+ICL | 0 | **0.609 ± 1.0e-3** | 0.595 ± 2.4e-3 | **0.889 ± 1.3e-3** |
| | 1 | 0.586 ± 3.7e-3 | 0.588 ± 4.7e-3 | 0.901 ± 1.1e-3 |
| | 5 | 0.607 ± 1.3e-3 | **0.663 ± 3.9e-3** | 0.901 ± 1.5e-3 |
| | 10 | 0.602 ± 1.4e-3 | 0.581 ± 3.3e-3 | 0.898 ± 2.8e-5 |
| | -1 | 0.606 ± 3.4e-3 | 0.588 ± 1.2e-3 | 0.898 ± 6.8e-3 |
| | cls | 0.542 ± 5.4e-4 | 0.552 ± 1.6e-3 | 1.135 ± 2.8e-3 |

Table 13: OT-PCA and OT-PCA+ICL across different layers (1 layer 3 repeat) on ESOL. **Bold** = best value (highest for Pearson/Spearman, lowest for RMSE); Underline = second-best.

| Method | Layer | Pearson | Spearman | RMSE |
|---|---|---|---|---|
| OT-PCA | 0 | 0.075 ± 2.3e-3 | 0.117 ± 4.4e-3 | 1.321 ± 2.5e-2 |
| | 1 | 0.053 ± 5.7e-3 | 0.088 ± 1.1e-2 | 1.331 ± 2.2e-2 |
| | 10 | 0.029 ± 9.6e-3 | 0.081 ± 1.3e-2 | 1.336 ± 2.1e-2 |
| | -1 | 0.114 ± 4.4e-4 | 0.177 ± 6.7e-4 | 1.268 ± 2.4e-2 |
| | cls | **0.223 ± 3.1e-4** | **0.232 ± 8.4e-4** | **1.210 ± 6.0e-4** |
| OT-PCA+ICL | 0 | 0.602 ± 1.8e-3 | 0.595 ± 4.4e-3 | 0.894 ± 6.6e-4 |
| | 1 | 0.579 ± 4.0e-3 | 0.582 ± 6.0e-3 | 0.920 ± 2.1e-3 |
| | 10 | **0.600 ± 1.9e-3** | **0.608 ± 2.2e-3** | **0.898 ± 1.3e-3** |
| | -1 | 0.598 ± 1.3e-3 | 0.603 ± 1.9e-3 | 0.892 ± 3.9e-3 |
| | cls | 0.493 ± 3.8e-3 | 0.527 ± 1.5e-3 | 0.898 ± 2.8e-5 |

## F.7 Experiments on the protein task and the DTI task

We also conducted experiments on drug-target interaction (DTI) and protein-related tasks. However, due to the poor performance of standard ICL on these datasets, with Pearson correlation coefficients below 0.2 [2]. Specifically, when evaluating the models on a test set consisting of 1,000 samples, we

observed that all methods performed close to random guessing. Regardless of whether representations or protein sequences were used, the models failed to learn the corresponding tasks. This can be attributed to the fact that such datasets typically focus on protein sequences within a fixed domain, resulting in high similarity between sequences and their FM-derived representations, with similarity scores approaching 0.99. To further investigate the performance of ICRL on such datasets, we conducted experiments using a randomly sampled test set of 90 samples for reference purposes.

For the DTI task, due to the limitations of the context window, the number of shots was set to 10, while the other experimental settings remained consistent with those described in the main text. When using Pearson r as the evaluation metric, we observed that ICRL still demonstrates relatively strong performance across most datasets, as shown in Tables 14 to 17, sometimes even surpassing ICL. Additionally, we provide results based on other evaluation metrics for reference. However, it is important to note that these findings are based on small-sample experiments. In larger datasets, the performance of all methods tends to converge and show minimal differences.

**V2 for protein.** To address the issue of protein sequence similarity, we conducted a simple method by extracting tokens preceding the CLS token from the ESM model. These tokens, which contain specific information about the protein sequences, were used as input features for ICRL. This approach demonstrated some improvement in small-sample datasets, as shown in Tables 16 and 17. Unfortunately, we found that it fails to resolve the fundamental issue of random guessing when applied to larger test sets.

**Effect of different FMs.** We also compared different protein FMs with varying levels of representation similarity. Using ESM2, which produces highly similar embeddings (average similarity ∼0.98), both ICL and ICRL performed poorly and often degenerated to random guessing. In contrast, ProtBert generates more diverse embeddings (average similarity ∼0.92), leading to consistently better results on stability and fluorescence prediction (see Tables 18 and 19). These results highlight that the diversity of FM-derived representations is a critical factor for ICRL effectiveness: when embeddings are overly homogeneous, the benefit of representation-based inputs diminishes, whereas more distinguishable embeddings allow ICRL to extract useful signals and surpass standard ICL. Importantly, this observation is consistent with our analyses on small-molecule datasets, indicating that the conclusion is *modality-agnostic*.

Table 14: Pearson comparison ↑ on the different DTI datasets without ICL. **Bold/** Underline: best/second-best value among the Embedding Injection methods. *: 1000 test samples.

| Dataset | String Injection | | Embedding Injection | | | |
| --- | --- | --- | --- | --- | --- | --- |
| | ICL | PCA | Random | Rep | Embed+OT | PCA+OT |
| BindingDB_IC50 | 0.149 ±6.9e-2 | 0.009 ±5.4e-4 | 0.092 ±3.6e-4 | 0.051 ±1.6e-3 | **0.177** ±2.4e-3 | 0.023 ±1.5e-4 |
| BindingDB_Ki | 0.172 ±8.5e-4 | 0.195 ±8.2e-4 | 0.170 ±4.6e-4 | 0.182 ±3.3e-4 | 0.113 ±6.1e-4 | **0.184** ±7.8e-4 |
| BindingDB_Ki* | 0.045 ± 2.10e-3 | -0.004 ± 3.8-4 | **0.084** ± 1.2e-3 | 0.010 ±2.1e-3 | -0.006 ±4.6e-4 | 0.037 ±2.4e-4 |

Table 15: Pearson ↑ comparison on DTI task datasets. **Bold/** Underline: best/second-best value compared with ICL.(highest for *Ser_cor*↑), *: 1000 test samples.

| Dataset | Baseline | ICRL (Ours) | | | | |
| --- | --- | --- | --- | --- | --- | --- |
| | Text ICL | Text PCA+ICL | Embedding | | | |
| | | | Ran-Noi+ICL | Ran-Pro+ICL | OT-Embed+ICL | OT-PCA+ICL |
| BindingDB_IC50 | 0.149 ±6.9e-4 | 0.124 ±2.8e-3 | 0.148 ±6.5e-3 | 0.081 ±4.5e-3 | **0.164** ±5.7e-3 | 0.163 ±2.8e-3 |
| BindingDB_Ki | 0.172 ±8.5e-4 | 0.241 ±1.1e-3 | 0.303 ±7.5e-4 | 0.268 ±1.3e-3 | 0.263 ±1.1e-3 | **0.333** ±8.9e-4 |
| BindingDB_Ki* | 0.045 ±2.1e-3 | 0.039 ±9.8e-4 | 0.084 ±1.2e-3 | **0.088** ±3.1e-3 | 0.078 ±7.9e-3 | 0.074 ±3.9e-7 |

## F.8 Experiments with Llama-3.2-3B-Instruct

To further investigate the reasons behind the failure of the DTI task, we conducted experiments using smaller models, i.e. Llama-3.2-3B-Instruct. As shown in Table 20, when employing smaller models, even on simpler tasks such as solubility prediction in the ESOL dataset, the standard ICL approach performs poorly. Notably, under these conditions, ICRL outperforms ICL, suggesting that the poor

Table 16: Pearson comparison ↑ on the different ESM task datasets without ICL. **Bold/** Underline: best/second-best value among the Embedding Injection methods. OOM: out of memory.

| Dataset | String Injection | | Embedding Injection | | | |
|---|---|---|---|---|---|---|
| | ICL | PCA | Random | Rep | Embed+OT | PCA+OT |
| Fluorescence | 0.237 ±1.9e-5 | 0.098 ±5.1e-4 | 0.114 ±8.4e-5 | 0.078 ±2.6e-4 | 0.051 ±4.7e-5 | **0.154** ±9.4e-5 |
| Stability | 0.130 ±7.3e-4 | 0.057 ±8.7e-4 | 0.127 ±1.2e-4 | **0.172** ±7.5e-4 | 0.097 ±2.9e-4 | 0.143 ±4.9e-4 |
| Stability_v2 | 0.130 ±4.1e-6 | OOM | 0.187 ±9.1e-7 | **0.212** ±7.2e-6 | 0.173 ±5.6e-6 | 0.180 ±7.3e-6 |

Table 17: Pearson ↑ comparison on ESM task datasets. **Bold/** Underline: best/second-best value compared with ICL. *: 1000 test samples. OOM: out of memory.

| Dataset | Baseline | | ICRL (Ours) | | | | |
|---|---|---|---|---|---|---|---|
| | Text ICL | Text PCA+ICL | Embedding | | | | |
| | | | Ran-Noi+ICL | Ran-Pro+ICL | OT-Embed+ICL | OT-PCA+ICL | |
| Fluorescence | 0.237 ±1.9e-5 | **0.164** ±2.1e-4 | 0.159 ±7.5e-5 | 0.129 ±4.5e-5 | 0.083 ±5.9e-5 | 0.151 ±2.4e-5 | |
| Stability | 0.130 ±7.3e-4 | 0.133 ±5.2e-3 | 0.131 ±4.4e-4 | 0.138 ±3.7e-4 | **0.141** ±2.3e-4 | 0.094 ±7.8e-5 | |
| Stability_v2 | 0.130 ±4.1e-6 | OOM | 0.111 ±1.7e-4 | **0.217** ±4.6e-4 | 0.177 ±1.4e-3 | 0.194 ±2.9e-5 | |

Table 18: Protein results using ESM2 as FM encoder (high similarity).

| ESM2 (sim ∼0.98) | ICL | OT-Embed | OT-PCA | OT-Embed+ICL | OT-PCA+ICL |
|---|---|---|---|---|---|
| Stability | 0.720 | 0.712 | 0.703 | 0.827 | 0.642 |
| Fluorescence | 0.995 | 1.322 | 1.222 | 0.997 | 0.987 |

Table 19: Protein results using ProtBert as FM encoder (more diverse embeddings).

| ProtBert (sim ∼0.92) | ICL | OT-Embed | OT-PCA | OT-Embed+ICL | OT-PCA+ICL |
|---|---|---|---|---|---|
| Stability | 0.720 | 0.631 (↓0.081) | 0.644 (↓0.059) | 0.673 (↓0.154) | 0.577 (↓0.065) |
| Fluorescence | 0.995 | 1.230 (↓0.092) | 1.044 (↓0.178) | 0.984 (↓0.013) | 0.949 (↓0.038) |

performance of ICRL in the DTI task stems from the model's unfamiliarity with the task, primarily due to insufficient pre-training knowledge.

Table 20: Pearson, Spearman, and RMSE results under the Llama 3B model. The symbol * indicates that no normalization was applied. The results demonstrate a clear advantage of ICRL.

| | ICL | PCA+OT | PCA+OT_ICL | Rep* | Rep+ICL* |
|---|---|---|---|---|---|
| Pearson | 0.048 ±9.9e-5 | 0.100 ±9.4e-4 | 0.133 ±2.0e-3 | 0.077 ±1.4e-3 | 0.140 ±6.2e-3 |
| Spearman | 0.045 ±5.6e-4 | 0.092 ±4.6e-4 | 0.117 ±6.9e-3 | 0.070 ±3.3e-3 | 0.126 ±4.3e-3 |
| RMSE | 1.317 ±2.2e-3 | 1.398 ±5.5e-3 | 1.341 ±2.1e-2 | 1.303 ±6.3e-3 | 1.244 ±5.8e-3 |

## F.9 Generalization Across LLMs

To further analyze the generalizability of ICRL across different language model capacities, we evaluated three LLaMA models of varying sizes: 3B, 8B, and 70B, on two molecular property prediction datasets—ESOL and Caco2. Each experiment was repeated 10 times with different random seeds, and we report the average of the top 3 runs for robustness.

As shown in Table 21, across both datasets and all model sizes, OT-based embeddings were consistently better interpreted and leveraged by the models, particularly by smaller models (e.g., 3B). This may be attributed to their more limited pretraining capacity, which makes external structured

representations more useful. Notably, ICRL even surpasses ICL in some scenarios, reinforcing its effectiveness and the dual-mode framework discussed in Sec. 5.

Table 21: RMSE(↓) results across three LLaMA models on the ESOL and Caco2 datasets. OT-based methods shows consistent improvements over baseline methods, especially in smaller models.

| Model Size | ESOL | | | | | Caco2 | | | | |
|---|---|---|---|---|---|---|---|---|---|---|
| | ICL | Ran-Noi | Zero-Pad | OT-Embed | OT-PCA | ICL | Ran-Noi | Zero-Pad | OT-Embed | OT-PCA |
| 3B | 1.313 | 1.817 | 2.013 | **1.299** | 1.315 | 1.027 | 1.207 | 1.193 | **0.971** | **0.965** |
| 8B | **1.179** | 1.764 | 1.837 | 1.186 | 1.177 | **0.892** | 1.103 | 1.098 | 0.903 | 0.902 |
| 70B | **1.158** | 1.412 | 1.727 | 1.199 | 1.243 | **0.832** | 1.026 | 1.037 | 0.899 | 0.898 |

These additional results confirm that OT-adjusted representations can be effectively interpreted and utilized across a spectrum of LLMs, thereby reinforcing the general utility of our method.

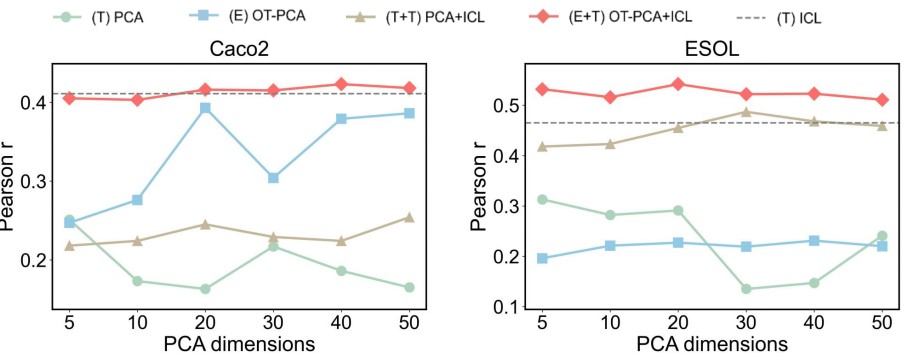

Figure 6: (a) and (b) present the performance of various methods under different PCA parameter settings, measured using Pearson correlation (higher is better).

## F.10 More Analysis about PCA Ablation.

The experimental results in Figs. 6, demonstrate that the performance of the text-level injection method is significantly affected by this parameter, particularly in non-ICL scenarios. In contrast, the embedding-level injection methods exhibit apparent stability, regardless of whether they are combined with ICL.

This phenomenon arises because, in text-level injection methods, modifying the PCA dimension directly alters the length of the injected strings, which can substantially impact the final inference results. On the other hand, in embedding-level injection methods, this change only introduces minor variations to the target distribution. Consequently, the OT-PCA method remains insensitive to this parameter.

## F.11 More Analysis about Representation Similarity.

In our experiments, we observed that when representation similarity is excessively high, the decoded representations exhibit nearly identical character compositions. Consequently, the model's output values are also highly similar, differing primarily in sequence and minor variations in decimal places due to the specific representation values. As illustrated in Figs. 10 and 11, which present examples of the Random Projection method, the model's responses to different queries show only sequential differences. The high degree of similarity in the representations can be attributed, in part, to the inherent characteristics of FM representations in chemical datasets. More about the visualization results of the similarity analysis in Sec. 5 are shown in Figs. 7 and 8.

This issue is particularly prominent in protein and drug-target interaction datasets, where the similarity values tend to cluster closely together, often reaching cosine similarity levels as high as 0.99.

# G  More Details

Due to the page limitation, we present more details about settings, results, and analysis here.

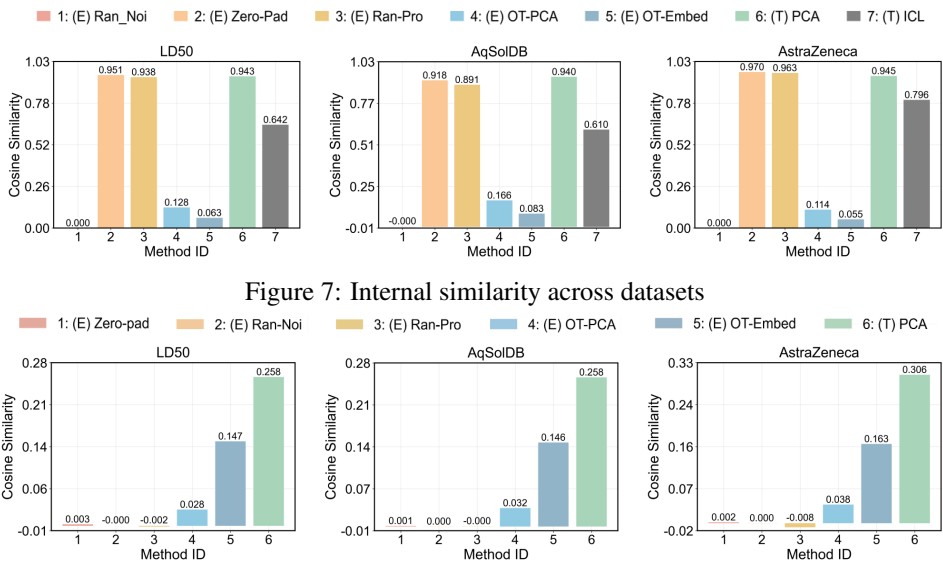

Figure 7: Internal similarity across datasets

Figure 8: Similarity to text embedding

## G.1 Experimental Setup

**Normalization.** First, we compute the overall average mean and variance from the non-zero elements of the embedding matrix of the LLM, ensuring that the statistics are representative of meaningful data. Specifically, we calculate the mean and variance for each embedding vector, then take the average across all embeddings to obtain a single reference mean and variance. Next, we normalize the input embeddings by adjusting their mean and variance to align with the target values derived from the LLM's embeddings. This is done by: (1) Calculate the mean and variance of the input embeddings. (2) Shift the embeddings by subtracting their mean and scaling them to match the target variance. (3) Finally, add the target mean to the scaled embeddings to ensure statistical consistency with the LLM's embeddings.. This normalization step ensures that the input embeddings are statistically aligned with the LLM's internal representations, improving compatibility and potentially enhancing model performance.

In our experiments, we observed that applying normalization to representations obtained through random methods occasionally resulted in NaN or Inf values. To simplify the process, we opted to disable normalization across all datasets. For the AqSolDB and LD50 Zhu datasets, a similar issue was observed with the OT-Embed method. However, given its relatively low occurrence—appearing in only 2-3 out of 10 random seeds—we conducted non-normalized experiments only for the OT-Embed method on these two datasets.

It is noteworthy that while normalization was consistently disabled across our experiments, in non-ICL scenarios, omitting normalization generally led to improved performance, particularly for the OT+PCA method. In such cases, the Pearson correlation coefficient improved by approximately 0.1 to 0.2, likely because normalization disrupts the distribution adjustments made by the OT method. Conversely, in ICL-inclusive settings, normalization proved beneficial in mitigating noise in representations, thereby enhancing retrieval performance.

**System prompt.** In all experiments presented in the main text, we consistently utilize a simple system prompt: *You are a drug expert. The answer should be different from the examples; DO NOT COPY ANY FLOAT VALUE*. For illustration, Fig. 9 provides an example of a decoded prompt for the OT+PCA+ICL approach.

## G.2 In-Context Example Sampling

For all regression and classification tasks, we adopt a *stratified sampling* strategy to select in-context examples from the training set. Specifically, for a given number of examples $k$, the training data are partitioned into $k$ equal-sized bins according to the label distribution, and one example is randomly

drawn from each bin. This ensures that the selected demonstrations cover a diverse range of labels and avoids over-representing particular regions of the label space. For question answering and caption generation tasks, we randomly sample $k$ examples uniformly from the training split at each run, following the original benchmark protocols. Beyond these default strategies, we also explored more informed selection methods. For example, using *Tanimoto similarity* to choose in-context examples that are closer to the test instance in the molecular embedding space can further improve ICRL performance. As discussed in Sec. 4, many techniques that enhance ICL quality—such as similarity-based sampling—also benefit ICRL in a consistent manner.

Considering the computational cost of running inference with 10 repeated trials, we further adopt the following rule for constructing candidate pools: 1. when the training dataset contains fewer than 1000 samples, we directly use the entire training set as the candidate pool; 2. when the dataset contains more than 1000 samples, we uniformly down-sample to 1000 candidates before applying the sampling procedure. This ensures a consistent and tractable evaluation budget across datasets of different sizes.

### G.2.1 Differences in different assessment indicators.

We employed three evaluation metrics: RMSE, Spearman correlation coefficient, and Pearson correlation coefficient. Among these, RMSE is the most sensitive metric, whereas improving the Pearson coefficient poses the greatest challenge. For instance, PCA may achieve a lower RMSE than ICL, yet its Pearson coefficient is lower, indicating that PCA should not be considered superior to ICL. Therefore, we primarily use RMSE to accurately assess performance differences in ICRL, while Pearson correlation is employed to demonstrate ICRL's contribution to enhancing ICL performance. The complete evaluation results are provided in Tables 22–25 for reference.

Table 22: Spearman ↑ comparison on the different datasets. **Bold/** Underline: best/second-best value compared with ICL.

| Dataset | Baseline | | ICRL (Ours) | | | | |
|---|---|---|---|---|---|---|---|
| | Text ICL | Text PCA+ICL | Zero-Pad+ICL | Ran-Noi+ICL | Embedding Ran-Pro+ICL | OT-Embed+ICL | OT-PCA+ICL |
| ESOL | 0.464 ±2.5e-3 | 0.460 ±7.4e-4 | 0.554 ±5.2e-4 | **0.562** ±5.0e-4 | 0.526 ±1.4e-4 | 0.522 ±8.7e-4 | 0.552 ±1.6e-3 |
| Caco2_Wang | 0.416 ±2.3e-3 | 0.387 ±1.8e-4 | 0.407 ±3.8e-4 | **0.426** ±3.5e-4 | 0.410 ±1.3e-5 | 0.409 ±3.3e-3 | 0.401 ±1.4e-3 |
| AqSolDB | 0.610 ±7.5e-5 | 0.551 ±5.0e-4 | **0.612** ±2.5e-5 | 0.594 ±3.6e-5 | 0.599 ±1.1e-4 | 0.586 ±1.0e-4 | 0.592 ±9.4e-5 |
| LD50_Zhu | 0.395 ±8.3e-5 | 0.382 ±5.0e-4 | **0.411** ±2.3e-5 | 0.399 ±3.3e-6 | 0.408 ±1.2e-4 | 0.382 ±2.3e-5 | 0.380 ±1.4e-4 |
| AstraZeneca | 0.233 ±2.3e-4 | 0.189 ±1.4e-5 | 0.239 ±2.8e-5 | 0.235 ±6.6e-6 | 0.234 ±1.6e-5 | **0.250** ±2.8e-4 | 0.244 ±1.9e-4 |

Table 23: RMSE ↓ comparison on the different datasets. **Bold/** Underline: best/second-best value compared with ICL (lowest for *RMSE↓*).

| Dataset | Baseline | | ICRL (Ours) | | | | |
|---|---|---|---|---|---|---|---|
| | Text ICL | Text PCA+ICL | Zero-Pad+ICL | Ran-Noi+ICL | Embedding Ran-Pro+ICL | OT-Embed+ICL | OT-PCA+ICL |
| ESOL | 1.158 ±1.9e-2 | 1.135 ±2.8e-3 | 1.085 ±4.9e-3 | 1.152 ±6.5e-3 | **1.037** ±4.5e-3 | 1.154 ±5.7e-3 | 1.135 ±2.8e-3 |
| Caco2_Wang | 0.832 ±8.4e-4 | 0.842 ±1.1e-3 | 0.830 ±8.8e-4 | 0.841 ±7.5e-4 | 0.839 ±1.3e-3 | **0.815** ±1.1e-3 | 0.888 ±8.9e-4 |
| AqSolDB | 1.917 ±3.9e-4 | 2.029 ±3.3e-3 | **1.910** ±5.2e-4 | 1.944 ±4.1e-4 | 1.941 ±3.0e-4 | 1.963 ±1.8e-3 | 1.986 ±1.1e-4 |
| LD50_Zhu | 0.986 ±1.7e-4 | 0.998 ±2.8e-4 | **0.966** ±2.0e-4 | 0.984 ±1.6e-4 | 0.995 ±7.7e-4 | 1.002 ±4.6e-4 | 1.004 ±1.7e-4 |
| AstraZeneca | 1.366 ±2.4e-3 | **1.319** ±9.2e-4 | 1.353 ±2.2e-4 | 1.399 ±1.1e-3 | 1.372 ±2.3e-3 | 1.372 ±1.2e-3 | 1.395 ±4.2e-3 |

Table 24: Pearson comparison ↑ on the different datasets without ICL. **Bold**/ Underline: best/second-best value among the Embedding Injection methods.

| Dataset | Text Injection | | Embedding Injection | | | | |
|---|---|---|---|---|---|---|---|
| | ICL | PCA | Zero-Pad | Ran-Noi | Ran-Pro | OT-Embed | OT-PCA |
| ESOL | 0.465 ±9.2e-4 | 0.291 ±1.1e-3 | 0.155 ±3.7e-3 | 0.123 ±2.6e-4 | 0.125 ±1.3e-3 | **0.270** ±8.2e-3 | 0.227 ±7.3e-4 |
| Caco2_Wang | 0.411 ±1.3e-3 | 0.163 ±1.5e-4 | 0.122 ±1.2e-3 | 0.114 ±1.3e-4 | 0.113 ±2.6e-4 | 0.226 ±1.3e-3 | **0.245** ±4.8e-4 |
| AqSolDB | 0.596 ±5.2e-5 | 0.075 ±3.6e-3 | 0.027 ±3.2e-4 | -0.026 ±8.0e-4 | 0.035 ±5.4e-4 | 0.115 ±6.4e-3 | **0.215** ±2.2e-3 |
| LD50_Zhu | 0.378 ±1.2e-5 | 0.128 ±1.6e-4 | 0.047 ±9.0e-5 | 0.029 ±1.2e-4 | 0.026 ±1.3e-5 | 0.064 ±5.6e-4 | **0.079** ±9.9e-5 |
| AstraZeneca | 0.266 ±2.3e-5 | 0.027 ±4.9e-4 | 0.041 ±2.8e-5 | 0.046 ±6.7e-5 | **0.049** ±3.3e-5 | 0.018 ±8.4e-5 | 0.043 ±1.6e-4 |

Table 25: Spearman comparison ↑ on the different datasets without ICL. **Bold**/ Underline: best/second-best value among the Embedding Injection methods.

| Dataset | String Injection | | Embedding Injection | | | | |
|---|---|---|---|---|---|---|---|
| | ICL | PCA | Zero-Pad | Ran-Noi | Ran-Pro | OT-Embed | OT-PCA |
| ESOL | 0.464 ±2.5e-3 | 0.299 ±8.4e-4 | 0.177 ±8.8e-3 | 0.127 ±3.6e-4 | 0.151 ±1.3e-3 | 0.231 ±6.0e-3 | **0.235** ±1.5e-4 |
| Caco2_Wang | 0.416 ±2.3e-3 | 0.175 ±8.2e-7 | 0.124 ±1.4e-3 | 0.113 ±2.6e-4 | 0.107 ±3.8e-4 | **0.245** ±6.0e-4 | 0.240 ±2.8e-4 |
| AqSolDB | 0.610 ±7.5e-5 | 0.056 ±3.8e-3 | 0.025 ±3.2e-4 | -0.026 ±1.6e-3 | 0.045 ±3.2e-4 | 0.116 ±4.9e-3 | **0.203** ±1.3e-3 |
| LD50_Zhu | 0.395 ±8.3e-5 | 0.165 2.5e-4 | 0.054 ±9.9e-6 | 0.039 ±1.8e-4 | 0.027 ±2.6e-5 | 0.090 ±1.6e-4 | **0.091** ±1.4e-4 |
| AstraZeneca | 0.233 ±2.3e-4 | 0.028 ±8.8e-4 | 0.035 ±3.2e-4 | **0.041** ±2.7e-4 | 0.040 ±2.0e-5 | 0.015 ±7.5e-5 | 0.040 ±1.8e-4 |

Table 26: Spearman comparison ↑ on the different DTI datasets without ICL. **Bold**/ Underline: best/second-best value among the Embedding Injection methods. *: 1000 test samples.

| Dataset | String Injection | | Embedding Injection | | | |
|---|---|---|---|---|---|---|
| | ICL | PCA | Random | Rep | Embed+OT | PCA+OT |
| BindingDB_IC50 | 0.156 ±1.4e-2 | 0.001 ±5.3e-4 | 0.080 ±1.5e-4 | 0.035 ±6.6e-3 | **0.156** ±2.1e-3 | 0.035 ±1.5e-4 |
| BindingDB_Ki | 0.152 ±3.5e-4 | 0.185 ±8.2e-4 | **0.184** ±1.1e-4 | 0.157 ±1.7e-4 | 0.104 ±2.0e-4 | 0.172 ±1.9e-4 |
| BindingDB_Ki* | 0.030 ±1.7e-3 | -0.004 ±2.8e-4 | **0.064** 1.2e-3 | 0.007 ±1.7e-3 | 0.002 ±1.6e-4 | 0.031 ±3.2e-4 |

Table 27: RMSE comparison ↓ on the different DTI datasets without ICL. **Bold**/ Underline: best/second-best value among the Embedding Injection methods.*: 1000 test samples.

| Dataset | String Injection | | Embedding Injection | | | |
|---|---|---|---|---|---|---|
| | ICL | PCA | Random | Rep | Embed+OT | PCA+OT |
| BindingDB_IC50 | 1.740 ±1.7e-3 | 1.659 ±5.6e-4 | 1.767 ±8.5e-4 | 1.754 ±2.4e-3 | **1.726** ±4.5e-4 | 1.743 ±4.4e-4 |
| BindingDB_Ki | 1.575 ±3.3e-3 | 1.555 ±7.2e-3 | 1.665 ±4.2e-4 | **1.578** ±1.3e-4 | 1.678 ±2.3e-4 | 1.655 ±4.9e-4 |
| BindingDB_Ki* | 1.631 ±2.1e-3 | 1.646 ±2.8e-4 | **1.612** ±9.2e-3 | 1.683 ±7.2e-3 | 1.716 ±5.0e-4 | 1.675 ±3.8e-4 |

Table 28: Spearman ↑ comparison on DTI task datasets. **Bold**/ Underline: best/second-best value compared with ICL, *: 1000 test samples.

| Dataset | Baseline | | ICRL (Ours) | | | | |
|---|---|---|---|---|---|---|---|
| | Text ICL | Text PCA+ICL | Ran-Noi+ICL | Ran-Pro+ICL | OT-Embed+ICL | OT-PCA+ICL |
| BindingDB_IC50 | 0.156 ±1.4e-4 | 0.120 ±2.8e-4 | 0.133 ±4.5e-5 | 0.057 ±4.7e-4 | 0.158 ±4.9e-4 | **0.163** ±2.8e-5 |
| BindingDB_Ki | 0.152 ±3.5e-5 | 0.200 ±1.3e-3 | 0.267 ±8.5e-4 | 0.223 ±1.7e-4 | 0.229 ±2.9e-4 | **0.310** ±2.8e-5 |
| BindingDB_Ki* | 0.030 ±1.7e-3 | 0.017 ±5.6e-4 | **0.070** ±8.8e-4 | 0.069 ±8.2e-5 | 0.064 ±4.6e-5 | 0.055 ±4.2e-5 |

Table 29: RMSE ↓ comparison on DTI task datasets. **Bold**/ Underline: best/second-best value compared with ICL, *: 1000 test samples.

| Dataset | Baseline | | ICRL (Ours) | | | |
| | Text ICL | Text PCA+ICL | Ran-Noi+ICL | Ran-Pro+ICL | OT-Embed+ICL | OT-PCA+ICL |
| | | | Embedding | | | |
|---|---|---|---|---|---|---|
| BindingDB_IC50 | 1.740 ±1.7e-3 | 1.739 ±2.8e-4 | 1.630 ±4.5e-5 | **1.637** ±4.7e-4 | 1.615 ±4.9e-4 | 1.748 ±2.8e-5 |
| BindingDB_Ki | 1.575 ±3.3e-3 | 1.572 ±1.3e-3 | 1.486 ±8.5e-4 | 1.544 ±1.7e-4 | 1.524 ±2.9e-4 | **1.465** ±2.8e-5 |
| BindingDB_Ki* | 1.631 ±2.1e-3 | 1.679 ±2.3e-3 | 1.645 ±1.6e-3 | **1.617** ±4.6e-4 | 1.659 ±1.5e-3 | 1.675 ±2.5e-5 |

Table 30: Spearman ↑ comparison on the different ESM datasets without ICL. **Bold**/ Underline: best/second-best value among the Embedding Injection methods. OOM: out of memory.

| Dataset | String Injection | | Embedding Injection | | | |
| | ICL | PCA | Random | Rep | Embed+OT | PCA+OT |
|---|---|---|---|---|---|---|
| Fluorescence | 0.204 ±1.7e-4 | 0.046 ±5.4e-3 | 0.132 ±8.7e-4 | 0.083 ±4.1e-4 | 0.096 ±7.7e-3 | **0.159** ±9.5e-4 |
| Stability | 0.133 ±7.7e-3 | 0.056 ±4.7e-4 | 0.170 ±1.3e-3 | **0.199** ±8.4e-4 | 0.095 ±2.8e-3 | 0.149 ±7.9e-4 |
| Stability_v2 | 0.133 ±4.2e-5 | OOM | 0.163 ±8.2e-6 | **0.187** ±7.7e-6 | 0.170 ±4.6e-6 | 0.166 ±6.3e-6 |

Table 31: RMSE comparison↓ on the different ESM datasets without ICL. **Bold**/ Underline: best/second-best value among the Embedding Injection methods. OOM: out of memory.

| Dataset | String Injection | | Embedding Injection | | | |
| | ICL | PCA | Random | Rep | Embed+OT | PCA+OT |
|---|---|---|---|---|---|---|
| Fluorescence | 1.213 ±2.7e-4 | 0.046 ±7.4e-3 | **1.429** ±8.1e-4 | 1.575 ±1.6e-4 | 1.613 ±3.2e-3 | 1.523 ±6.7e-4 |
| Stability | 0.735 ±7.9e-3 | 0.649 ±2.7e-4 | 1.009 ±1.6e-3 | 0.973 ±8.8e-4 | **0.894** ±6.5e-3 | 0.952 ±4.1e-4 |
| Stability_v2 | 0.735 ±7.2e-5 | OOM | 0.894 ±7.7e-6 | 0.878 ±7.7e-6 | 0.898 ±4.9e-6 | **0.862** ±5.8e-6 |

Table 32: Spearman ↑ comparison on ESM task datasets. **Bold**/ Underline: best/second-best value compared with ICL. *: 1000 test samples. OOM: out of memory.

| Dataset | Baseline | | ICRL (Ours) | | | |
| | Text ICL | Text PCA+ICL | Ran-Noi+ICL | Ran-Pro+ICL | OT-Embed+ICL | OT-PCA+ICL |
| | | | Embedding | | | |
|---|---|---|---|---|---|---|
| Fluorescence | 0.204 ±1.7e-4 | 0.137 ±4.1e-4 | 0.129 ±7.0e-5 | 0.061 ±6.5e-5 | 0.110 ±7.4e-5 | **0.138** ±3.7e-5 |
| Stability | 0.133 ±7.9e-3 | 0.146 ±5.7e-5 | 0.125 ±4.1e-4 | **0.153** ±6.7e-4 | 0.142 ±2.1e-4 | 0.086 ±7.7e-5 |
| Stability_v2 | 0.133 ±4.2e-5 | OOM | 0.111 ±1.9e-4 | **0.192** ±4.1e-4 | 0.172 ±4.4e-3 | 0.184 ±8.9e-5 |

Table 33: RMSE comparison ↓ on the different ESM datasets without ICL. **Bold**/ Underline: best/second-best value compared with ICL. OOM: out of memory

| Dataset | Baseline | | ICRL (Ours) | | | |
| | Text ICL | Text PCA+ICL | Ran-Noi+ICL | Ran-Pro+ICL | OT-Embed+ICL | OT-PCA+ICL |
| | | | Embedding | | | |
|---|---|---|---|---|---|---|
| Fluorescence | 1.213 ±2.7e-4 | 1.462 ±7.1e-3 | 1.407 ±7.0e-3 | 1.467 ±5.5e-4 | 1.479 ±7.9e-4 | **1.370** ±6.7e-4 |
| Stability | 0.735 ±7.9e-5 | 0.848 ±7.7e-5 | 0.770 ±4.8e-6 | 0.780 ±6.1e-4 | **0.716** ±2.3e-4 | 0.848 ±7.9e-4 |
| Stability_v2 | 0.735 ±7.2e-5 | OOM | **0.719** ±2.8e-5 | 0.720 ±5.1e-5 | 0.738 ±4.7e-5 | 0.780 ±8.4e-5 |

**Prompt**

<|begin_of_text|><|start_header_id|>system<|end_header_id|>

You are a drug expert. The answer should be different from the examples; DO NOT COPY ANY FLOAT VALUE.

<|eot_id|><|start_header_id|>user<|end_header_id|>

Drug SMILES: < ON=C1CCCCC1 >
Given the SMILES sequence of the drug molecule above, answer the following question using the specified format.
Question: What is the Solubility of the drug molecule above?
Molecular vector representation: [REP]771[/REP]
Answer: -0.85

Drug SMILES: < O=c1cc[nH]c(=O)[nH]1 >
Given the SMILES sequence of the drug molecule above, answer the following question using the specified format.
Question: What is the Solubility of the drug molecule above?
Molecular vector representation: [REP]727[/REP]
Answer: -1.48

Drug SMILES: < CCOC(=O)C1=C(COCCN2C(=O)c3ccccc3C2=O)NC(C)=C(C(=O)OC)C1c1ccccc1Cl >
Given the SMILES sequence of the drug molecule above, answer the following question using the specified format.
Question: What is the Solubility of the drug molecule above?
Molecular vector representation: [REP]738[/REP]
Answer: -4.5
......
Drug SMILES: < CCOC(=O)c1nc(C(Cl)(Cl)Cl)n(-c2ccc(Cl)cc2Cl)n1 >
Given the SMILES sequence of the drug molecule above, answer the following question using the specified format.
Question: What is the Solubility of the drug molecule above?
Molecular vector representation: [REP]742[/REP]
Answer: -5.65

Drug SMILES: < Clc1cccc(Cl)c1 >
Given the SMILES sequence of the drug molecule above, answer the following question using the specified format.
Question: What is the Solubility of the drug molecule above?
Molecular vector representation: [REP]453[/REP]
Answer: -3.07

Drug SMILES: < CN(C)C(=O)Nc1ccc(Cl)c(Cl)c1 >
Given the SMILES sequence of the drug molecule above, answer the following question using the specified format.
Question 1: What is the Solubility of the drug molecule above?
Molecular vector representation: [REP]649[/REP]
Please respond with the following format for each question:
---BEGIN FORMAT TEMPLATE FOR QUESTION 1---
Answer 1: [Your Answer Here for Question 1]
---END FORMAT TEMPLATE FOR QUESTION 1---

Drug SMILES: < CC1(N)CN(c2c(F)cc3c(=O)c(C(=O)O)cn(C4CC4)c3c2Cl)C1 >
Given the SMILES sequence of the drug molecule above, answer the following question using the specified format.
Question 2: What is the Solubility of the drug molecule above?
Molecular vector representation: [REP]742[/REP]
Please respond with the following format for each question:
---BEGIN FORMAT TEMPLATE FOR QUESTION 2---
Answer 2: [Your Answer Here for Question 2]
---END FORMAT TEMPLATE FOR QUESTION 2---

Drug SMILES: < CC(=O)Oc1ccc(-c2ccccc2)cc1 >
Given the SMILES sequence of the drug molecule above, answer the following question using the specified format.
Question 3: What is the Solubility of the drug molecule above?
Molecular vector representation: [REP]673[/REP]
Please respond with the following format for each question:
---BEGIN FORMAT TEMPLATE FOR QUESTION 3---
Answer 3: [Your Answer Here for Question 3]
---END FORMAT TEMPLATE FOR QUESTION 3---<|eot_id|><|start_header_id|>assistant<|end_header_id|>

**Output**

---BEGIN FORMAT TEMPLATE FOR QUESTION 1---
Answer 1: -2.93
---END FORMAT TEMPLATE FOR QUESTION 1---

---BEGIN FORMAT TEMPLATE FOR QUESTION 2---
Answer 2: -2.14
---END FORMAT TEMPLATE FOR QUESTION 2---

---BEGIN FORMAT TEMPLATE FOR QUESTION 3---
Answer 3: -1.21
---END FORMAT TEMPLATE FOR QUESTION 3---

Figure 9: Decoded prompt example and output for ESOL task.

**Prompt**

<|begin_of_text|><|start_header_id|>system<|end_header_id|>

You are a drug expert. The answer should be different from the examples; DO NOT COPY ANY FLOAT VALUE.

<|eot_id|><|start_header_id|>user<|end_header_id|>

Drug SMILES: < O=c1c(-c2ccc(O)cc2)coc2cc(OC3OC(CO)C(O)C(O)C3O)ccc12 >
Given the SMILES sequence of the drug molecule above, answer the following question using the specified format.
Question: What is the permeation rate through Caco-2 cells of the drug molecule above?
Molecular vector representation: [REP]-des[/REP]
Answer: -6.36

Drug SMILES: < COC(=O)c1ccccc1-c1c2ccc(=N)cc-2oc2cc(N)ccc12 >
Given the SMILES sequence of the drug molecule above, answer the following question using the specified format.
Question: What is the permeation rate through Caco-2 cells of the drug molecule above?
Molecular vector representation: [REP]-des[/REP]
Answer: -5.94

Drug SMILES: < Cc1ccc(N2CCN(C(=O)[C@H](C)Cc3ccc(Cl)cc3C)CC2)c([C@@H](NC(=O)CCN(C)C)C(C)C)c1 >
Given the SMILES sequence of the drug molecule above, answer the following question using the specified format.
Question: What is the permeation rate through Caco-2 cells of the drug molecule above?
Molecular vector representation: [REP]BBBB[/REP]
Answer: -4.8
......
Drug SMILES: < C[C@@H]1O[C@@H](O[C@@H]2C(=O)c3c(O)cc(O)cc3O[C@H]2c2ccc(O)c(O)c2)[C@H](O)[C@H](O)[C@H]1O >
Given the SMILES sequence of the drug molecule above, answer the following question using the specified format.
Question: What is the permeation rate through Caco-2 cells of the drug molecule above?
Molecular vector representation: [REP]-des[/REP]
Answer: -6.82

Drug SMILES: < CN(C)CCCN1c2ccccc2Sc2cccccc21 >
Given the SMILES sequence of the drug molecule above, answer the following question using the specified format.
Question: What is the permeation rate through Caco-2 cells of the drug molecule above?
Molecular vector representation: [REP]-des[/REP]
Answer: -4.38

Drug SMILES: < COc1cc(OC)c2c(=O)cc(-c3ccccc3)oc2c1 >
Given the SMILES sequence of the drug molecule above, answer the following question using the specified format.
Question: What is the permeation rate through Caco-2 cells of the drug molecule above?
Molecular vector representation: [REP]-des[/REP]
Answer: -4.63

Drug SMILES: < CCN(CC)CCCC(C)Nc1ccnc2cc(Cl)ccc12 >
Given the SMILES sequence of the drug molecule above, answer the following question using the specified format.
Question 1: What is the permeation rate through Caco-2 cells of the drug molecule above?
Molecular vector representation: [REP]BBBB[/REP]
Please respond with the following format for each question:
---BEGIN FORMAT TEMPLATE FOR QUESTION 1---
Answer 1: [Your Answer Here for Question 1]
---END FORMAT TEMPLATE FOR QUESTION 1---

Drug SMILES: < Clc1ccc2c(c1)CCc1cccnc1C2=C1CCNCC1 >
Given the SMILES sequence of the drug molecule above, answer the following question using the specified format.
Question 2: What is the permeation rate through Caco-2 cells of the drug molecule above?
Molecular vector representation: [REP]-des[/REP]
Please respond with the following format for each question:
---BEGIN FORMAT TEMPLATE FOR QUESTION 2---
Answer 2: [Your Answer Here for Question 2]
---END FORMAT TEMPLATE FOR QUESTION 2---

Drug SMILES: < COC(=O)Nc1nc2ccc(C(=O)c3ccccc3)cc2[nH]1 >
Given the SMILES sequence of the drug molecule above, answer the following question using the specified format.
Question 3: What is the permeation rate through Caco-2 cells of the drug molecule above?
Molecular vector representation: [REP]-des[/REP]
Please respond with the following format for each question:
---BEGIN FORMAT TEMPLATE FOR QUESTION 3---
Answer 3: [Your Answer Here for Question 3]
---END FORMAT TEMPLATE FOR QUESTION 3---<|eot_id|><|start_header_id|>assistant<|end_header_id|>

**Output**

---BEGIN FORMAT TEMPLATE FOR QUESTION 1---
Answer 1: -5.68
---END FORMAT TEMPLATE FOR QUESTION 1---

---BEGIN FORMAT TEMPLATE FOR QUESTION 2---
Answer 2: -6.51
---END FORMAT TEMPLATE FOR QUESTION 2---

---BEGIN FORMAT TEMPLATE FOR QUESTION 3---
Answer 3: -5.69
---END FORMAT TEMPLATE FOR QUESTION 3---

Figure 10: Decoded prompt example and output for Caco2 task.

**Prompt**

<|begin_of_text|><|start_header_id|>system<|end_header_id|>

You are a drug expert. The answer should be different from the examples; DO NOT COPY ANY FLOAT VALUE.

<|eot_id|><|start_header_id|>user<|end_header_id|>

Drug SMILES: < O=c1c(-c2ccc(O)cc2)coc2cc(OC3OC(CO)C(O)C(O)C3O)ccc12 >
Given the SMILES sequence of the drug molecule above, answer the following question using the specified format.
Question: What is the permeation rate through Caco-2 cells of the drug molecule above?
Molecular vector representation: [REP]-des[/REP]
Answer: -6.36

Drug SMILES: < COC(=O)c1ccccc1-c1c2ccc(=N)cc-2oc2cc(N)ccc12 >
Given the SMILES sequence of the drug molecule above, answer the following question using the specified format.
Question: What is the permeation rate through Caco-2 cells of the drug molecule above?
Molecular vector representation: [REP]-des[/REP]
Answer: -5.94

Drug SMILES: < Cc1ccc(N2CCN(C(=O)[C@H](C)Cc3ccc(Cl)cc3C)CC2)c([C@@H](NC(=O)CCN(C)C)C(C)C)c1 >
Given the SMILES sequence of the drug molecule above, answer the following question using the specified format.
Question: What is the permeation rate through Caco-2 cells of the drug molecule above?
Molecular vector representation: [REP]BBBB[/REP]
Answer: -4.8
......
Drug SMILES: < C[C@@H]1O[C@@H](O[C@@H]2C(=O)c3c(O)cc(O)cc3O[C@H]2c2ccc(O)c(c2)[C@H](O)[C@H](O)[C@H]1O >
Given the SMILES sequence of the drug molecule above, answer the following question using the specified format.
Question: What is the permeation rate through Caco-2 cells of the drug molecule above?
Molecular vector representation: [REP]-des[/REP]
Answer: -6.82

Drug SMILES: < CN(C)CCCN1c2ccccc2Sc2ccccc21 >
Given the SMILES sequence of the drug molecule above, answer the following question using the specified format.
Question: What is the permeation rate through Caco-2 cells of the drug molecule above?
Molecular vector representation: [REP]-des[/REP]
Answer: -4.38

Drug SMILES: < COc1cc(OC)c2c(=O)cc(-c3ccccc3)oc2c1 >
Given the SMILES sequence of the drug molecule above, answer the following question using the specified format.
Question: What is the permeation rate through Caco-2 cells of the drug molecule above?
Molecular vector representation: [REP]-des[/REP]
Answer: -4.63

Drug SMILES: < CSCC[C@H](NC(=O)[C@H](Cc1ccccc1)NC(=O)CNC(=O)[C@@H](C)NC(=O)[C@@H](N)Cc1ccc(O)cc1)C(=O)O >
Given the SMILES sequence of the drug molecule above, answer the following question using the specified format.
Question 1: What is the permeation rate through Caco-2 cells of the drug molecule above?
Molecular vector representation: [REP]BBBB[/REP]
Please respond with the following format for each question:
---BEGIN FORMAT TEMPLATE FOR QUESTION 1---
Answer 1: [Your Answer Here for Question 1]
---END FORMAT TEMPLATE FOR QUESTION 1---

Drug SMILES: < CC[C@]1(O)C[C@@H]2CN(CCc3c([nH]c4ccccc34)[C@@](C(=O)OC)(c3cc4c(cc3OC)N(C)[C@H]3[C@@](O)(C(=O)OC)[C@H](OC(C)=O)[C@]5(CC)C=CCN6CC[C@]43[C@@H]65)C2)C1 >
Given the SMILES sequence of the drug molecule above, answer the following question using the specified format.
Question 2: What is the permeation rate through Caco-2 cells of the drug molecule above?
Molecular vector representation: [REP]-des[/REP]
Please respond with the following format for each question:
---BEGIN FORMAT TEMPLATE FOR QUESTION 2---
Answer 2: [Your Answer Here for Question 2]
---END FORMAT TEMPLATE FOR QUESTION 2---

Drug SMILES: < C[C@H](N[C@H](CCc1ccccc1)C(=O)O)C(=O)N1CCC[C@@H]1C(=O)O >
Given the SMILES sequence of the drug molecule above, answer the following question using the specified format.
Question 3: What is the permeation rate through Caco-2 cells of the drug molecule above?
Molecular vector representation: [REP]-des[/REP]
Please respond with the following format for each question:
---BEGIN FORMAT TEMPLATE FOR QUESTION 3---
Answer 3: [Your Answer Here for Question 3]
---END FORMAT TEMPLATE FOR QUESTION 3---<|eot_id|><|start_header_id|>assistant<|end_header_id|>

**Output**

---BEGIN FORMAT TEMPLATE FOR QUESTION 1---
Answer 1: -5.68
---END FORMAT TEMPLATE FOR QUESTION 1---

---BEGIN FORMAT TEMPLATE FOR QUESTION 2---
Answer 2: -6.51
---END FORMAT TEMPLATE FOR QUESTION 2---

---BEGIN FORMAT TEMPLATE FOR QUESTION 3---
Answer 3: -5.69
---END FORMAT TEMPLATE FOR QUESTION 3---

Figure 11: Decoded prompt example and output for Caco2 task.

