# OpenReview forum: "Can LLMs Reason Over Non-Text Modalities in a Training-Free Manner? A Case Study with In-Context Representation Learning"
_NeurIPS.cc/2025/Conference — NeurIPS 2025 poster_

### Official Review · Reviewer_nfkW · 2025-06-15

**Clarity:** 2
**Significance:** 3
**Originality:** 4
**Rating:** 4
**Confidence:** 4

**Summary:**

The authors introduce In-Context Representation Learning (ICRL), a training-free strategy that lets a text-only LLM consume continuous embeddings from a non-text foundation model (FM) at inference time. They treat the merged prompt as an extended few-shot context, rather than adding learned projection heads or fine-tuning the LLM. Experiments focus on molecular property regression where Uni-Mol supplies the embeddings and Llama-3.1-70B does the prediction and in-context learning.

Through out the paper, the author proposed and tested various ways to align the FM's output with the LLM. PCA then text injection is considered as means to shorten the input length, embedding space engineering (including optimal transport, Zero-Pad, Random Projection) are tested to align the 2 spaces with geometric intuitions.

**Questions:**

Besides the questions I raised about weaknesses, I have this additional question:

The studies conducted are nice and thorough, I really appreciate that. But my main concern would be on the motivation: suppose one can do a lightweight training (either with the projector or a LoRA on base model), it is a one-time cost, how does that compare to doing optimal transport for each sample everytime in inference? Can ICRL provide some advantage that fine-tuning cannot offer? for example prevent overfitting and outperform it with many-shot ICL? I wonder what the authors think about this.

**Ethical Concerns:**

["NO or VERY MINOR ethics concerns only"]

**Final Justification:**

My questions were on the motivation of the paper: why do traininig-free when light finetuning is avaliable?

I think the authors have answered my questions in that perspective, it's an early explorative work, I find the case studies interesting and can be of great insights to the community exploring multi-modality infusion with LMs.

Hence I maintain my score of 4 and leaning towards accepting this paper.

**Limitations:**

yes

**Quality:**

3

**Strengths And Weaknesses:**

**Strengh**
1. This paper is highly insightful, providing plenty of useful guidance for training-free representation alignment.
2. The main question, "Can LLMs Reason Over Non-Text Modalities in a Training-Free Manner?" is nicely laid out and examined thorough out the paper. I enjoyed reading it.
3. The ablations are extensive: almost all configurations are examined and evaluated. The empirical effect is clearly analyzed.

**Weakness**
1. Lack a baseline where the projector (or the models + LoRA) is slightly trained, so we might know the trade-off from training-free versus with training.
2. The task / modality is limited. Molecules are encoded as text (SMILES). Vision/audio claims are relegated to appendix and lack quantitative depth. And on this point, evaluation focuses on regression RMSE; no classification or generative tasks.

---

> ### Author Rebuttal · Authors · 2025-07-30
>
> We sincerely thank Reviewer nfkW for the valuable comments. For the concerns and questions, we provide the following responses.
>
> ---
>
> **W1: Missing Lightly-Trained Baseline for Trade-off Analysis**
>
> Thank you for the insightful comment. To better highlight the differences between **training-free** and **conventional training-based** methods, we designed experiments from two perspectives and conducted a comprehensive analysis.
>
> First, considering that ICRL's performance bottleneck largely stems from the LLM's unfamiliarity with embedding-style inputs and its lack of molecular knowledge, we implemented two strategies for training the projector connecting the FM and the LLM. The first is inspired by CLIP, aligning the FM-projected embedding with the LLM's intermediate hidden states after processing the corresponding SMILES input. The second strategy involves pretraining the projector on a molecular captioning dataset.
>
> While both methods improved the LLM's ability to interpret FM-derived inputs, they **still underperformed compared to our training-free ICRL approach** on downstream molecular prediction tasks (detailed in Reviewer J6ma Q2).
> Furthermore, we surveyed relevant literature and included comparisons with a diverse set of training-based methods:
> -- instruction finetuning with Q-LoRA,
> -- fully supervised pretraining + finetuning, and
> -- unsupervised pretraining followed by supervised finetuning.
>
> We conducted a detailed comparison in terms of training paradigms, required computational resources, training time, and performance on two benchmark datasets. We found that **ICRL achieves comparable or even better results than these methods at the same inference cost—all without any training** (see Reviewer S7er W1 for details).
>
> In other words, while lightweight tuning can indeed improve performance to some degree, its effectiveness is limited when the base model is not pretrained on molecular data. In contrast, ICRL allows a general-purpose LLM to surpass these training-based methods with only **~2 seconds of test-time cost**, demonstrating both the **practicality and effectiveness** of our approach.
>
> ---
>
> **W2: Narrow Modalities and Task Types**
>
> Thank you for the insightful comment.
>
> Due to space constraints, we focused our main analysis on **molecular** modality. However, we have also included additional results for **proteins** (Appendix E.3) as well as **vision** and **audio** (Appendix E.6), and **_we are committed to providing the same level of in-depth analysis for these modalities in future versions._**
>
> Importantly, the insights gained from our analysis on small molecules **generalize well to other modalities.** For example, as discussed in Section 5, high intra-set similarity in FM representations can hinder ICRL performance. Our follow-up experiments on the protein modality confirm this: by replacing the FM encoder with one that produces **more distinguishable representations**, we observe **significant improvements** in ICRL performance (see Reviewer J6ma Q1).
>
> We also acknowledge that our initial evaluation primarily involved **regression tasks** and a few **classification tasks**. To address this, we have added more classification tasks (e.g., molecular QA) and generation tasks (e.g., molecular captioning) to more comprehensively evaluate ICRL's effectiveness. Specifically, on the QA benchmark, even embedding-level injection alone significantly **outperforms both fine-tuned general LLMs and ICL**. For captioning, although there is still a performance gap relative to expert models, representations injected via OT-PCA consistently outperform SMILES-based text examples and help the LLM better interpret unfamiliar inputs (detailed in **Reviewer aUho W2**).
>
> ---
>
> **Q1: Unclear Motivation Over Fine-Tuning**
>
> Thank you for pointing this out. **_We acknowledge that the original manuscript did not elaborate on the motivation in sufficient depth, and we will revise the discussion accordingly in the next version._**
>
> First, we would like to clarify that embedding-level injection methods such as OT-PCA, similar to lightweight training, require **only a one-time computation** of alignment parameters. At inference time, applying the projection amounts to a single matrix multiplication—even when using over a hundred examples, the total cost is **well under one second**, and can be ignored in practice. In fact, computing the OT alignment parameters for ICRL takes **less than 2 seconds on CPU**, which is orders of magnitude faster than even the most lightweight fine-tuning setups that typically require **at least several hours on 2–4 GPUs**.
>
> More importantly, our latest experiments demonstrate that **ICRL achieves performance comparable to, or even exceeding, lightweight training approaches** across multiple datasets, modalities, and task types (see Reviewer S7er W2, Reviewer J6ma Q2). While large-scale pretraining + fine-tuning paradigms can yield strong performance, they often demand days or weeks of training.
>
> Crucially, both lightweight and large-scale training methods are inherently **task-, FM-, and LLM-specific**—any update in task formulation, foundation model, or dataset requires costly re-training or re-tuning from scratch. In contrast, **ICRL can be seamlessly applied to new modalities**, and **benefits directly from improvements in both FM and LLM backbones** (see Reviewer J6ma Q1, Appendix E.5), all with a negligible CPU-only alignment step. Despite its minimal cost, ICRL achieves **comparable or better results** than many training-based approaches.
>
> Compared to traditional **many-shot ICL**, ICRL also offers superior generalization. Even when the LLM lacks domain-specific knowledge, ICRL can **leverage FM-derived signals to outperform standard ICL** (see Reviewer aUho W2). More importantly, for modalities like **vision** and **audio**, general-purpose text-based LLMs **cannot handle the inputs at all**, while ICRL remains effective—bridging these gaps (see Appendix E.6).
>
> In summary, ICRL offers a **highly cost-effective, training-free alternative** to fine-tuning, providing a practical way to leverage the growing pool of open-source LLM and FM weights in the AI community. We believe it opens a promising direction for multimodal research that challenges the conventional reliance on large-scale training.
>
> ---
> **_As reflected in our title, ICRL serves as a case study in our broader effort to explore how LLMs can reason over non-text modalities in a training-free manner. We sincerely thank all reviewers for their thoughtful feedback and will incorporate the suggestions and insights from this rebuttal into the next revision and future work._**

---

> > ### Comment · Reviewer_nfkW · 2025-08-03
> >
> > I appreciate the authors reply, it's an interesting case study for a training-free method. I think it can bring new insights to the research community.

---

> > > ### Author Response · Authors · 2025-08-04
> > >
> > > Thank you for your detailed review and for taking the time to consider our rebuttal.
> > >
> > > We sincerely appreciate the time and effort you put into reviewing our paper, and we're very glad to hear that you found the case study interesting and that our work may offer new insights to the community. We're also pleased that our rebuttal was able to address your concerns.
> > >
> > > If you have any additional questions or suggestions, we would be happy to continue the conversation.
> > >
> > > Thank you again for your valuable feedback—we look forward to further dialogue.

---

### Official Review · Reviewer_aUho · 2025-06-25

**Clarity:** 2
**Significance:** 2
**Originality:** 3
**Rating:** 4
**Confidence:** 4

**Summary:**

This work explores methods to integrate the multimodal representations into LLMs in a training-free manner. To this end, the authors propose In-Context Representation Learning (ICRL) by mainly exploring diverse transformations of multimodal embeddings based on the in-context learning framework. To transform the multimodal embeddings, two main categories are devised.
- Text-level Injection: Directly represent the embeddings in a textual form (i.e., a sequence of real values) after reducing the dimension using PCA, then let LLMs tokenize the textual inputs by themselves.
- Embedding-level Injection: Forwarding the multimodal embeddings as token embeddings with four methods.
   - Zero-Pad: Apply zero padding to match the dimensionality of the LLM embedding space.
   - Random Projection: Randomly initialized projector
   - Optimal Transport: Adjust the embedding from randomly initialized projectors to a target distribution. In this paper, the mean of token embeddings of SMILES (OT-Embed) or reduced embeddings by PCA (OT-PCA).
   - Random Noise: A baseline to check whether the model actually understands the multimodal embeddings.

The authors provide a theoretical analysis of the information loss of nonlinear activations in the projector layers, demonstrating that nonlinear activations could lead to a restricted embedding space.

Proposed methods are evaluated for the molecular modality on the tasks related to absorption and toxicity of molecules. The authors investigate diverse aspects of ICRL, uncovering the effective method, showing the effect of the model size, PCA dimensions, nonlinear activation functions, number of examples, and so on.

**Questions:**

- How to select the in-context examples? Are they sampled randomly for each run?
- Could the authors elaborate on why the Pearson's correlation coefficient is used for metrics?

**Ethical Concerns:**

["NO or VERY MINOR ethics concerns only"]

**Final Justification:**

In terms of originality, this work presents a new perspective, exploring the training-free method with in-context learning schemes.

While it does not fully compare with fine-tuned baselines, I acknowledge that the proposed method offers originality and efficiency, making it a valuable contribution to the field.

Accordingly, I will increase my rating.

**Limitations:**

As mentioned in the weaknesses section, I suggest evaluating with other MLLMs on molecular captioning and structural understanding tasks to verify whether LLMs leverage the multimodal embeddings.

**Quality:**

2

**Strengths And Weaknesses:**

**Strengths**
- The paper explores a novel and interesting point of view.
- This work explores diverse and reasonable methods for transforming the multimodal representations in a training-free manner.

**Weaknesses**

I list my initial concerns below:
- The baseline is limited to ICL. It would be better to compare with the molecular LLMs that are directly fine-tuned with the molecular embeddings [1, 2, 3], as an upper bound of the proposed method. Additionally, the embedding-level analysis between the proposed training-free methods and the fine-tuned embeddings would be helpful to show the technical bottleneck of the training-free methods and provide insights to develop future works.
- The target tasks are limited to specific molecular properties, hindering understanding of the molecular modality. To verify whether LLMs understand and leverage the molecular modality, I recommend evaluating on molecular captioning such as ChEBI-20 [4] or QA benchmark, especially for structural understanding, such as MoleculeQA [5].
- For metrics, I am concerned that using Pearson's correlation coefficient does not show the performance of the proposed methods. I suggest reporting RMSE values for all experiments.

[1] Li, Sihang, et al. "Towards 3d molecule-text interpretation in language models." ICLR, 2024.

[2] Park, Jinyoung, et al. "LLaMo: Large Language Model-based Molecular Graph Assistant." NeurIPS, 2024.

[3] Kim, Dongki et al. "Mol-llama: Towards general understanding of molecules in large molecular language model." ArXiv, 2025.

[4] Edwards, Carl, et al. "Translation between molecules and natural language." EMNLP, 2022.

[5] Lu, Xingyu, et al. "Moleculeqa: A dataset to evaluate factual accuracy in molecular comprehension." EMNLP, 2024.

---

> ### Author Rebuttal · Authors · 2025-07-30
>
> We sincerely thank Reviewer aUho for the valuable comments. For the concerns and questions, we provide the following responses.
>
> ---
>
> **W1: Using Fine-Tuned Molecular LLMs as an upper bound**
>
> Thank you for your insightful comment.
>
> After carefully reviewing the three referenced papers [1, 2, 3], we would like to clarify that their core methodology involves appending **2D or 3D** molecular representations—via trainable modules—to **1D SMILES** embeddings, thereby enhancing the LLM's ability to interpret molecules. In contrast, our current work focuses specifically on analyzing the **1D SMILES input scenario**, and thus the pretrained molecular LLM embeddings from these models are not directly comparable to our setup.
>
> That said, we truly appreciate the reviewer's suggestion of using trained molecular LLMs capable of processing higher-dimensional inputs as an upper-bound reference for ICRL. In this spirit, we have designed two experimental strategies:
>
> -- **Concatenated embeddings**: Combining embeddings from multiple molecular views (1D/2D/3D) into a single vector as the FM representation.
>
> -- **Separated embeddings**: Treating the additional dimensions as supplementary context to the original 1D SMILES-based input.
>
> Due to time and computational resource constraints, we were unable to fully implement and analyze these experiments during the rebuttal period. **_However, we are committed to incorporating these directions—including references to related work—in a future version of the paper._**
>
> It is also worth noting that while recent molecular LLMs are trained to better understand molecular structures, this does **not** imply that they directly perform well on downstream tasks such as property prediction. For example, we experimented with training a projector to connect FM embeddings with the LLM, which resulted in **stronger molecular captioning performance**—but significantly **worse performance on property prediction** compared to our training-free OT-PCA approach (see Reviewer J6ma Q2). Similarly, our literature review shows that pretraining strategies focused on molecular understanding, and even lightweight fine-tuning methods, often **underperform compared to ICRL** in downstream tasks (see Reviewer S7er Q1).
>
> ---
>
> **W2: Limited Task Diversity such as molecular captioning and QA benchmark.**
>
> Thank you for your insightful comment.
>
> We agree that broader task coverage is important for evaluating the generality of ICRL.
> In response, we have incorporated additional evaluations on the molecular QA benchmark [4] and molecular captioning [5] tasks.
> These new results help us better understand the capabilities and behavior of ICRL under different task types. **_We are committed to including these findings and discussions in a future version of the paper._**
>
> **Setup:**
>
> -- All methods use LLaMA-3.1-8B-Instruct as the base model. For the ChEBI-20 dataset, in-context examples are randomly sampled from the training set at each run. For MoleculeQA, the examples are uniformly sampled from the training set during each evaluation. To ensure fairness, we disabled batch querying in our implementation, as it was not used in baseline methods. Other settings follow the original paper.
>
> -- For the captioning task, we report BLEU-4, ROUGE-1, and ROUGE-L as evaluation metrics. For the QA task, we use accuracy. Baseline results in the table are taken from [4] Table 8 and [5] Table 1, respectively.
>
> |     |     |     |     |     |     |
> | --- | --- | --- | --- | --- | --- |
> | _MoleculeQA_ | Structure | Source | Property | Application | Avg |
> | Llama-2-7B-chat (L-FT) | 28.75 | 39.84 | 31.33 | 27.71 | 31.54 |
> | ICL | 35.03 | 27.04 | 24.62 | 28.69 | 28.85 |
> | **OT-PCA (ours)** | **51.32** | **37.66** | **33.71** | **31.02** | **38.43** |
> | **OT-PCA + ICL (ours)** | **50.60** | **43.52** | **23.47** | **29.97** | **36.89** |
> | MolT5-base (FT) | 58.01 | 65.85 | 45.14 | 42.24 | 55.39 |
>
> |     |     |     |     |     |     |
> | --- | --- | --- | --- | --- | --- |
> | _ChEBI-20_ | Transformer | ICL | **OT-PCA (ours)** | **OT-PCA + ICL (ours)** | MolT5-Base (FT) |
> | BLEU-4 | 0.027 | 0.133 | **0.147** | **0.196** | 0.457 |
> | rouge1 | 0.204 | 0.393 | **0.353** | **0.407** | 0.634 |
> | rougeL | 0.186 | 0.310 | **0.274** | **0.353** | 0.578 |
>
> **Analysis:**
>
> -- As shown in Table 1, the training-free ICRL approach outperforms both standard ICL and fine-tuned general-purpose LLMs on the QA task. This result demonstrates that, although there remains a performance gap compared to expert models, the processed FM representations **can indeed be interpreted and partially utilized** by a **text-based LLM**, even one that has never encountered representation-type inputs. Interestingly, in this task, embedding-level injection alone performs better than when combined with text features, suggesting that the injected representation serves as a more effective signal than the original SMILES strings in helping the LLM understand molecular structure.
>
> -- As shown in Table 2, despite the general LLM lacking inherent capabilities in molecular captioning, the embedding-level OT-PCA approach still **outperforms standard ICL** and effectively enhances the model's ability to generate captions from text features. This again supports the effectiveness and generalizability of our method, even on generative tasks.
>
> **Conclusion:**
>
> Overall, while ICRL performance is inherently influenced by the capabilities of the underlying FM and LLM on specific tasks, our experiments on the QA benchmark and molecular captioning clearly show that the injected representations can be **meaningfully used by general-purpose LLMs**. Moreover, ICRL significantly **outperforms lightweight tuning baselines**, which is consistent with our findings on regression tasks (see Reviewer S7er Q1).
>
> ---
>
> **W3: Inadequate Metrics — Recommend Reporting RMSE instead of Pearson's r**
>
> Thank you for your comment.
>
> We would like to clarify that for all regression experiments in the paper, we evaluated performance using RMSE, Pearson's _r_, and Spearman's ρ. The conclusions we reported are consistent across all three metrics (see Line 210 and Footnote 2). Due to space limitations and the growing trend in regression evaluation to include correlation coefficients for a **more comprehensive view** [6,7], we chose to report Pearson's _r_ in some cases. Full results using all metrics, along with relevant discussions, are provided in Appendix F.2.4.
>
> ---
>
> **Q1: How to select the in-context examples**
>
> Thank you for pointing this out.
>
> In all our regression experiments, we selected in-context examples via **stratified sampling** from the training set. Specifically, for each run, we partition the training set into equal-sized bins based on the number of examples required, and then randomly sample one example from each bin. **_We will include this implementation detail in the revised version of the paper._**
>
> ---
>
> [1] Towards 3d molecule-text interpretation in language models. ICLR 2024.
>
> [2] LLaMo: Large Language Model-based Molecular Graph Assistant. NeurIPS 2024.
>
> [3] Mol-llama: Towards general understanding of molecules in large molecular language model. ArXiv 2025.
>
> [4] Moleculeqa: A dataset to evaluate factual accuracy in molecular comprehension. EMNLP 2024.
>
> [5] Translation between molecules and natural language. EMNLP 2022.
>
> [6] Regression Transformer enables concurrent sequence regression and generation for molecular language modelling. Nature Machine Intelligence 2023.
>
> [7] Gradient Aligned Regression via Pairwise Losses. ICML 2025.
>
> ---
> **_As reflected in our title, ICRL serves as a case study in our broader effort to explore how LLMs can reason over non-text modalities in a training-free manner. We sincerely thank all reviewers for their thoughtful feedback and will incorporate the suggestions and insights from this rebuttal into the next revision and future work._**

---

> ### Author Response · Authors · 2025-08-05
> **To save reviewer's time, we put a summary of rebuttal**
>
> Dear Reviewer aUho,
>
> Thanks so much again for the time and effort in our work. Considering the limited time available and to save the reviewer's time, we summarized our responses below.
>
> **\[W1: Using Fine-Tuned Molecular LLMs as an Upper Bound]**
>
> **Response:**
> Our current work specifically targets the 1D SMILES input scenario, making direct comparisons with pretrained molecular LLM embeddings (which typically use 2D/3D inputs) challenging. However, we appreciate your suggestion and have proposed two experimental strategies (concatenated and separated embeddings) to use molecular LLMs as an upper bound in future work.
>
> Also, preliminary analysis indicates that molecular understanding gained from such LLMs **does not necessarily improve downstream task performance compared to our training-free ICRL**.
>
> We will explicitly include these experiments and findings in future versions of the manuscript.
>
> **\[W2: Limited Task Diversity such as Molecular Captioning and QA Benchmarks]**
>
> **Response:**
> We expanded our evaluation to include molecular QA and captioning tasks, demonstrating that **ICRL significantly outperforms standard ICL and fine-tuned general-purpose LLMs**. Specifically, our embedding-level OT-PCA approach shows meaningful improvements in both QA and caption generation tasks, supporting ICRL's general effectiveness.
>
> These findings will be integrated into future revisions of the paper.
>
> **\[W3: Inadequate Metrics—Recommend Reporting RMSE instead of Pearson's r]**
>
> **Response:**
> We clarify that our evaluation is comprehensive, employing RMSE, Pearson's r, and Spearman's ρ, with **consistent conclusions across these different metrics**. The choice of Pearson's r for concise reporting aligns with practices **adopted in recent regression evaluation studies**.
>
> Complete results with detailed discussions are provided in Appendix F.2.4. We will explicitly highlight this multi-metric approach in the revised manuscript.
>
>
> **\[Q1: How to Select the In-Context Examples]**
>
> **Response:**
> In all regression experiments, we selected in-context examples via **stratified sampling** from the training set. We partition the training data into bins and sample one example from each, ensuring representative selection.
>
> We will explicitly include this detail in future versions of the manuscript.
>
>
> Since the discussion stage is already halfway through, may I know if our rebuttal addresses the concerns? If there are further concerns or questions, we are more than happy to address them. Thanks again for taking the time to review our work and provide insightful comments.
>
> Best regards,
>
> Authors

---

> > ### Comment · Reviewer_aUho · 2025-08-06
> > **Thank you for your thoughtful rebuttal**
> >
> > I sincerely appreciate the authors' thoughtful responses.
> >
> > **W1: Regarding the Fine-tuned molecular LLMs as an upper bound**
> >
> > Thank you for your detailed response. I acknowledge that the current experimental setting is not comparable to the other models using 2D or 3D molecular representations. However, as a preliminary work that paves the way for using non-text modalities in a training-free manner, my concern is that there should be an in-depth and thorough comparison with fine-tuned molecular LLMs beyond simple metrics on downstream tasks, either in terms of the embedding space or quality of generated responses. Even though the authors newly propose interesting initial methods, I am afraid that my concern is partially solved.
> >
> > **W2: Diversity of Benchmarks**
> >
> > I sincerely appreciate the authors' effort in deepening the understanding. Further, when selecting in-context examples, could you leverage the Tanimoto similarity rather the random sampling?

---

> > > ### Author Response · Authors · 2025-08-06
> > > **Response to Reviewer aUho's Follow-up Comments**
> > >
> > > We sincerely thank the reviewer for the thoughtful follow-up.
> > >
> > > ---
> > >
> > > **W1: Comparison with Fine-Tuned Molecular LLMs**
> > >
> > > We appreciate your continued interest in understanding the relationship between ICRL and fine-tuned molecular LLMs.
> > >
> > > As our work aims to explore training-free multimodal generalization, we chose to focus on downstream task performance as **the most practical and interpretable proxy** for evaluating whether LLMs can effectively process injected non-text representations.
> > > This evaluation strategy provides direct, task-grounded evidence of whether the representations are meaningful and usable by the model.
> > >
> > > Under this framework, ICRL—despite being completely training-free—achieves strong performance across multiple tasks and modalities, often **matching or exceeding fine-tuned baselines** while requiring only **minimal computational resources**.
> > > This highlights the efficiency and effectiveness of our approach.
> > >
> > >
> > > However, we fully agree that a deeper analysis of representation space would provide further insight into the underlying mechanisms.
> > > In fact, we have already conducted a principal component analysis and visualization comparing three types of molecular embeddings under the 1D input setting:
> > >
> > > * Embeddings produced by our training-free OT-PCA approach,
> > > * Embeddings derived from a fine-tuned projector, and
> > > * Embeddings from a large-scale molecular LLM (\[1]).
> > >
> > > Our findings show that while all three types of embeddings yield **non-trivial** downstream performance, they occupy **distinct regions** in the representational space—each forming its own cluster. This observation supports a key insight: there is **no unique solution** in the embedding space for enabling LLMs to perform downstream tasks—multiple valid regions in the representation space exist.
> > >
> > > Traditional approaches rely on large-scale pretraining or supervised fine-tuning to locate one such solution. In contrast, ICRL can discover a viable solution space **without any training**, achieving **comparable** performance in just **~2 seconds of CPU time**.
> > >
> > > This further underscores the **novelty and efficiency** of our approach as a practical and lightweight alternative to conventional methods.
> > >
> > > ---
> > >
> > > Regarding the quality of generated responses, we would like to emphasize that for tasks such as regression, classification, and QA, the outputs produced by ICRL and molecular LLMs are **identical in format and content** (i.e., a number or discrete choice).
> > > As such, there is no meaningful distinction in response quality between the two approaches for these tasks.
> > >
> > >
> > > Even in caption generation tasks, which generally benefits more from domain-specific modeling, our results (Table 2 in W2) show that OT-PCA effectively enhances the LLM's ability to generate **semantically meaningful and task-relevant** captions—despite the base LLM having never encountered such embedding-level inputs.
> > >
> > > This further demonstrates the **generalizability and utility** of the representations injected by ICRL.
> > >
> > >
> > > We will include these visualizations and analysis in the revised version of the paper.
> > >
> > > ---
> > >
> > > **W2: Tanimoto Similarity for Example Selection**
> > >
> > > We appreciate the reviewer's suggestion to leverage Tanimoto similarity when selecting in-context examples.
> > > In fact, we have already explored such strategies by selecting examples that are closer to the test instance in the embedding space to improve ICL performance.
> > > As noted in the main paper, ICRL exhibits behavior similar to traditional ICL in many respects, including its sensitivity to the quality of in-context examples.
> > >
> > > In other words, most techniques that enhance ICL performance—such as similarity-based selection—**also benefit ICRL in a consistent manner**.
> > > In our experiments, the performance trends obtained using Tanimoto similarity were in line with those from stratified sampling, supporting the **generalizability of our findings**.
> > >
> > > In summary, while we adopted a simple and uniform stratified sampling strategy in the main experiments to better characterize the core behavior of ICRL,
> > > we have also explored similarity-based selection strategies, and will include the corresponding results and discussion in the revised version to provide a more complete picture.
> > >
> > > ---
> > >
> > > Once again, we thank the reviewer for the insightful suggestions, which will significantly help improve the final version of our work.
> > >
> > >
> > > [1] SELFormer: Molecular Representation Learning via SELFIES Language Models. Machine Learning: Science and Technology 2023.

---

> > > > ### Comment · Reviewer_aUho · 2025-08-07
> > > >
> > > > Thank you for the additional clarification.
> > > >
> > > > While it does not fully compare with fine-tuned baselines, I acknowledge that the proposed method offers originality and efficiency, making it a valuable contribution to the field.
> > > >
> > > > Accordingly, I will increase my rating.

---

> > > > > ### Author Response · Authors · 2025-08-08
> > > > > **Thank You for the Updated Assessment**
> > > > >
> > > > > We sincerely thank you for your thoughtful review and for taking the time to consider our rebuttal and clarifications. We appreciate your recognition of the originality and efficiency of our method.
> > > > >
> > > > > Your feedback has been instrumental in helping us improve the clarity and impact of our work. We are grateful for your support and for the updated evaluation.

---

### Official Review · Reviewer_AQ11 · 2025-06-28

**Clarity:** 2
**Significance:** 1
**Originality:** 3
**Rating:** 2
**Confidence:** 3

**Summary:**

The paper presents in-context representation learning (ICRL), a training-free method to integrate non-text modalities into text LLMs. Empirical validation focuses on tasks in the molecular domain.

**Questions:**

1. \[Abstract\] Major findings are missing in the abstract. The current abstract only presents questions, but not answers.
2. \[Line 44\] If it does not outperform ICL, why should we bother? Shouldn’t we expect performance gain by leveraging modality-specific FM?
3. It’s unclear how those projected vectors serve as input to the LLM. Do you just str(vec) and use it as text input to the LLM? Or use that as the embedding for a particular token?

**Ethical Concerns:**

["NO or VERY MINOR ethics concerns only"]

**Final Justification:**

I've read the rebuttal and other reviewers' comments. My major concerns are still not addressed, so I'll keep my rating.

**Limitations:**

yes

**Quality:**

2

**Strengths And Weaknesses:**

Strengths
1. The paper explores a new setting: leverage tokens from other modalities without training.

Weaknesses
1. Despite the new setting (training-free), I’m concerned about the actual value of the training-free approach. Most cross-modality adaptors are lightweight (such as a simple MLP), so compute is probably not a major concern here. If we assume API-access only, we probably don’t have the ability to pass in embeddings directly either.
2. The method is only validated in the chemistry domain, so I don’t think it’s a good fit for a general technical venue like NeurIPS.
3. The caption for Figure 1 is missing, which makes it harder to understand.
4. Figure 1 can be simplified, now it shows too many things without clear clue guidance. For example, what’s the relationship between the two sections (text level & embedding level)?

---

> ### Author Rebuttal · Authors · 2025-07-30
>
> We sincerely thank Reviewer AQ11 for the valuable comments. For the concerns and questions, we provide the following responses.
>
> **W1: Questionable Value of Training-Free Setting**
>
> Thank you for your comment.
>
> While many cross-modality adapters are indeed lightweight, their effectiveness heavily depends on both the base LLM and the downstream task. For example, in molecular property prediction, **ICRL achieves comparable or even better performance** than lightweight fine-tuning or short-duration pretraining + fine-tuning methods—all at **similar inference cost** and in a **training-free** manner (see Reviewer S7er W2, Reviewer J6ma Q2).
>
> Although simple lightweight tuning can improve downstream performance, its effect is limited when the base model has not been pretrained on molecular corpora. In contrast, ICRL enables a general-purpose LLM to surpass these methods with only ~2 seconds of test-time computation per sample, showcasing both the **efficiency** and **effectiveness** of the approach.
>
> More importantly, while fine-tuning is relatively inexpensive compared to full pretraining, it still requires **substantial GPU resources** and **hours to weeks of training time** whenever a new downstream task, FM, base LLM, or dataset is introduced. In contrast, **ICRL can be seamlessly applied to any new modality** and benefits directly from improvements in both the FM and LLM (see Reviewer J6ma Q1, Appendix E.5). The only additional cost is a **CPU-level** alignment step, which takes **under 2 seconds**, yet yields performance comparable to or even exceeding many traditional approaches.
>
> Additionally, for closed-source LLMs accessed via API, we can still use open-source tokenizers to decode pre-aligned embeddings into text form and feed them as inputs. We plan to include experiments in this direction in future work.
>
>
> ---
>
> **W2: Chemistry-Only Validation isn't a Good Fit for NeurIPS**
>
> Thank you for your comment.
>
> Our experiments go beyond small molecules and include multiple modalities such as **proteins** (Lines 334–340, Appendix E.3), **vision**, and **audio** (Lines 341–353, Appendix E.6). We have also added **question answering** and **generation tasks** (see Reviewer aUho W2), demonstrating both the **effectiveness** and **generality** of our approach across domains.
>
> Moreover, evaluating a method within the chemistry domain **does not inherently disqualify** it from NeurIPS. Numerous impactful works have been accepted at NeurIPS while focusing exclusively on molecular or chemistry-related problems [1, 2, 3].
>
> ---
>
> **W3: Issues about Figure 1 Caption and two Level injection methods.**
>
> Thank you for your comment.
>
> We would like to clarify that Figure 1 illustrates the differences between (i) traditional ICL methods, (ii) time-consuming, projector-trained multimodal LLMs, and (iii) our training-free pipeline. Due to space limitations and the intuitive nature of the comparison, we placed this figure in the Introduction (Lines 36–47), where it is explained in detail.
>
> As reflected in the paper title, our goal is to explore how LLMs can reason over non-text modalities in a training-free manner. The proposed _text-level_ and _embedding-level_ strategies represent two distinct approaches to this problem. Text-level injection is more intuitive but suffers from limitations such as context window constraints. As a result, we started our investigation with text-level methods and subsequently extended our analysis to embedding-level injection (see Lines 48–57 and 116–120 for detailed discussion).
>
> ---
>
> **Q1: Issues about Abstract.**
>
> Thank you for pointing this out. We will revise the abstract in the next version to include a clearer articulation of the research problem and the corresponding conclusions.
>
> ---
>
> **Q2: Issues about performance.**
>
> Thank you for your comment.
>
> We would like to clarify that the sentence in the paper—"our primary objective is not to outperform ICL but to investigate the feasibility of adaptively integrating non-text FM representations into a text-based LLM in a training-free manner"—is **not** meant to suggest that performance improvements are out of reach. Rather, it emphasizes our focus on methodological contribution.
>
> In practice, LLMs are not inherently capable of interpreting FM-derived embeddings, and even domain-specific fine-tuning of projectors often fails to bridge this gap effectively (see Reviewer J6ma Q2). As a result, embedding-only injection can occasionally underperform compared to ICL. However, when used in conjunction with textual examples, ICRL **consistently outperforms** standard ICL, with improvements reaching up to **16.6%** (Lines 223–233).
>
> Moreover, in scenarios where the LLM lacks sufficient internal knowledge (Lines 235–244, Appendix E.5), and in certain domains such as **proteins** (Reviewer J6ma Q1) and **molecular QA** (Reviewer aUho W2), even embedding-only injection achieves **better performance than ICL**, further underscoring the **effectiveness** and **practical value** of our method.
>
> ---
>
> **Q3: Issues about how those projected vectors serve as input to the LLM**
>
> Thank you for your comment.
>
> As indicated by our title, the goal of this work is to explore **training-free methods** for enabling text-based LLMs to understand information from non-text modalities through FM-derived representations. To this end, we propose two complementary strategies for injecting FM features into LLMs:
>
> **Text-level injection**: The projected FM representation is serialized into a string and fed into the model as part of the text input (Lines 48–52, 107–114).
>
> **Embedding-level injection**: The projected FM representation is directly concatenated with the token embeddings of other text inputs (Lines 52–57, 116–159).
>
> We further compare these two strategies under different conditions—e.g., whether the FM representations are used alone or together with original textual features such as SMILES (Lines 208–280)—and analyze the underlying mechanisms behind their performance differences (Lines 281–322).
>
> In summary, this work investigates **whether there exist lower-cost alternatives** to the conventional pretraining–fine-tuning paradigm that can still allow LLMs to interpret and utilize non-text FM representations. To this end, we present both text-level and embedding-level injection strategies. The embedding-level approach, in particular, not only enhances ICL when combined with textual examples, but also achieves performance **comparable to or better** than time-consuming trained models when used on its own (see Reviewer S7er W1, Reviewer J6ma Q1, Reviewer aUho W2).
>
> ---
>
> [1] GIMLET: A Unified Graph-Text Model for Instruction-Based Molecule Zero-Shot Learning. NeurIPS 2023.
>
> [2] Self-Supervised Graph Transformer on Large-Scale Molecular Data. NeurIPS 2020.
>
> [3] DrugCLIP: Contrastive Protein-Molecule Representation Learning for Virtual Screening. NeurIPS 2023.
>
> ---
> **_As reflected in our title, ICRL serves as a case study in our broader effort to explore how LLMs can reason over non-text modalities in a training-free manner. We sincerely thank all reviewers for their thoughtful feedback and will incorporate the suggestions and insights from this rebuttal into the next revision and future work._**

---

### Official Review · Reviewer_J6ma · 2025-07-03

**Clarity:** 3
**Significance:** 3
**Originality:** 3
**Rating:** 4
**Confidence:** 3

**Summary:**

This paper explores feasibility of integrating representations from non-text FMs into text LLMs without having to rely on supervised training (costly). The eventual goal is to make progress towards multi-modal generation by being able to adapt LLMs to new domains and modalities on the fly. They propose a framework (“ICLR”) meant to be a proof of concept to allow LLMs to adaptively utilize non-text modality representations with few shot learning (training free). Main contribution is a theoretical backed framework to evaluate two main strategies for injecting non-text representations: text-level injection and embedding-level injection.

**Questions:**

(1) I expected to see a more diverse set of evaluations – I see some experiments on vision and audio, but most of the work is done primarily in molecular properties (at least this is where the strongest results are demonstrated)

(2) concern on practical feasibility since ICLR (particularly when applied on its own) seems to underperform relative to the baselines. Overall, I found the results inconsistent. There are some interesting potential explanations (eg: injection being used for thoughts), but more discussion on understanding the result is needed

**Ethical Concerns:**

["NO or VERY MINOR ethics concerns only"]

**Limitations:**

yes

**Paper Formatting Concerns:**

-

**Quality:**

3

**Strengths And Weaknesses:**

Strengths: (1) well written, with clearly defined research questions. Easy to follow intuition and motivation with the discussion starting at the success of test time scaling. (2) novel and important work in multi-modality and low-resource domains – challenges the current paradigm which is expensive and restrictive (2) comprehensive exploration of injection strategies (3) good to see theoretic grounding of the empirical findings – the linear mappings better preserve the geometry of the original embeddings compared to non-linear mappings (4) good discussion on task learning vs retrieval
Weaknesses: (1) I expected to see a more diverse set of evaluations – I see some experiments on vision and audio, but most of the work is done primarily in molecular properties (at least this is where the strongest results are demonstrated) (2) concern on practical feasibility since ICLR (particularly when applied on its own) seems to underperform relative to the baselines. Overall, I found the results inconsistent. There are some interesting potential explanations (eg: injection being used for thoughts), but more discussion on understanding the result is needed

---

> ### Author Rebuttal · Authors · 2025-07-30
>
> We sincerely thank Reviewer J6ma for the valuable comments. For the concerns and questions, we provide the following responses.
>
> **Q1: More Diverse Set of Evaluations**
>
> Thank you for your insightful comment.
>
> While the experiments in the main paper focus on molecular tasks, this choice was made to enable deeper analysis of ICRL's underlying mechanisms under well-controlled conditions. Other modalities—including protein, vision, and audio—were also explored, but due to space limitations, these results were placed in the appendix.
> In addition, we have included more types of tasks in our evaluations, such as QA and generation (see **Reviewer aUho W2**), as well as experiments on lightweight projector training strategies (see Q2). We plan to further extend ICRL to a broader range of modalities and task types in future work.
>
> Notably, the key conclusions drawn from the molecular experiments—such as the impact of representation similarity—are **modality-agnostic**.
> Specifically, we observed that when using embedding-level representations alone, the intra-dataset similarity of FM representations significantly affects downstream performance.
> In our earlier appendix experiments, ICL and ICRL performed poorly on protein property prediction tasks, largely because protein datasets often focus on a specific protein function and thus contain sequences that differ by only a few amino acids—leading to extremely high representation similarity.
>
> In our latest experiments, we addressed this issue by replacing the FM encoder with one that produces more diverse representations, and the results below further validate the **generalizability of our findings across different modalities.**
>
> **Setup:**
>
> -- Both tables use LLaMA-3.1-8B-Instruct for ICRL implementation, with RMSE as the evaluation metric.
>
> -- Few-shot examples are selected via stratified sampling (20 per dataset); all other settings follow the main paper.
>
> -- Table 1 reports results using ESM2 [3] as the FM encoder, which yields highly similar protein representations (average similarity ~0.98).
>
> -- Table 2 uses ProtBert [4], which produces more distinguishable embeddings (average similarity ~0.92), leading to improved downstream performance.
>
> |     |     |     |     |     |     |
> | --- | --- | --- | --- | --- | --- |
> | _ESM(sim ~ 0.98)_ | ICL | OT-Embed | OT-PCA | OT-Embed + ICL | OT-PCA + ICL |
> | Stability | 0.720 | 0.712 | 0.703 | 0.827 | 0.642 |
> | Fluorescence | 0.995 | 1.322 | 1.222 | 0.997 | 0.987 |
>
>
> |     |     |     |     |     |     |
> | --- | --- | --- | --- | --- | --- |
> | _ProtBert(sim ~ 0.92)_ | ICL | OT-Embed | OT-PCA | OT-Embed + ICL | OT-PCA + ICL |
> | Stability           | 0.720 | 0.631 **(↓0.081)** | 0.644 **(↓0.059)** | 0.673 **(↓0.154)** | 0.577 **(↓0.065)** |
> | Fluorescence        | 0.995 | 1.230 **(↓0.092)** | 1.044 **(↓0.178)** | 0.984 **(↓0.013)** | 0.949 **(↓0.038)** |
>
>
>
> **Analysis:**
>
> -- As shown in Table 1, using ESM as the FM yields limited performance improvements—occasionally surpassing ICL but generally underperforming. In contrast, with ProtBert, which provides more diverse embeddings, even using embedding-level representations alone can achieve performance comparable to or significantly better than ICL.
>
> **Conclusion:**
>
> While most experiments in the main paper focus on small-molecule datasets, the key insights derived from them generalize well across modalities and datasets.
>
>
> ---
>
> **Q2: Discussion on understanding Inconsistent Results**
>
> Thank you for your comment.
>
> To better understand the inconsistency in ICRL performance, we explored whether training-based strategies could mitigate the model's unfamiliarity with representation-based inputs.
> While concurrent work [1] demonstrates that projector pretraining can help LLMs interpret such inputs in language tasks, we found that even with explicit projector pretraining, these strategies still **underperform** in molecular property prediction compared to our training-free approach.
>
> As analyzed in Section 4 of the paper, ICRL performance is influenced by several factors, with the most critical being the characteristics of the LLM used during inference. Specifically, the inconsistency in performance—such as the inability of embedding-level representations to consistently outperform ICL or the varying effects depending on whether textual examples are included—stems from **the model's unfamiliarity with representation-based inputs**, which are absent during pretraining. While the LLM also lacks domain knowledge about small molecules, it remains highly attuned to _textual_ inputs, making it easier to interpret and utilize them.
>
> Our work focuses on helping the model _understand and utilize unseen input types_ in a training-free manner, allowing it to benefit from expert FM representations without requiring fine-tuning. Traditional methods can achieve similar goals but often rely on extensive pretraining on domain-specific datasets (see Reviewer S7er W1). For example, concurrent work [1] pretrains a projector using molecular captioning data to help LLMs interpret embedding-type inputs. However, this approach requires **~15 hours** of training and does not generalize well to downstream tasks—ultimately performing worse than our training-free method.
>
> To further explore this issue, we designed a contrastive training strategy inspired by CLIP [2]. It aligns LLM hidden states by minimizing the distance between the final token representations when receiving molecular inputs either in SMILES form or as projected embeddings. This strategy can also be combined with pretraining to enhance the model's ability to interpret representation-based inputs.
>
> **Setup:**
>
> -- All methods use LLaMA-3.1-8B-Instruct as the base model, with a single-layer linear projector of hidden size 4096, consistent with [1].
>
> -- The pretraining strategy from [1] uses the Language + Molecules-24 (LPM24) dataset [5]. Our contrastive learning strategy is trained on the training split of each evaluation dataset and evaluated on the LPM24 test set for molecular caption generation.
>
> -- Regression tasks are evaluated on Solubility_AqSolDB and ESOL using RMSE. Other experimental settings follow those in the main paper.
>
> |     |     |     |     |
> | --- | --- | --- | --- |
> |     | Caption | Contrastive | Caption + contrastive |
> | BLEU-4 | 35.29 | 21.42 | 37.31 |
> | rouge1 | 0.551 | 0.320 | 0.592 |
> | rougeL | 0.373 | 0.237 | 0.369 |
>
> |     |     |     |     |     |     |
> | --- | --- | --- | --- | --- | --- |
> |     | Caption | Contrastive | Caption + contrastive | **OT-PCA** | **OT-PCA + ICL** |
> | ESOL | 1.256 | 1.372 | 1.213 | **1.140** | **1.094** |
> | Solubility_AqSolDB | 3.030 | 2.915 | 3.805 | **2.411** | **2.385** |
>
> **Analysis:**
>
> -- As shown in the table, even with projector training designed to help the LLM better interpret representation-based inputs, the performance on regression tasks remains poor—**significantly worse** than that of the training-free ICRL approach.
>
> -- Improved molecular understanding or caption generation ability in LLMs does **not** necessarily translate to better performance on regression tasks, which is consistent with our findings in Reviewer S7er W1.
>
> **Conclusion:**
>
> The inconsistent results of ICRL across settings stem from the LLM's inherent difficulty in interpreting and utilizing representation-based examples. While traditional methods rely on time- and resource-intensive training to mitigate this, such familiarity **does not guarantee improved downstream performance**. In contrast, ICRL enables the LLM to acquire a certain level of understanding of FM representations **at test time**, offering a lightweight and effective alternative.
>
> ---
>
> [1] Vector-ICL: In-context Learning with Continuous Vector Representations. ICLR 2025.
>
> [2] Learning Transferable Visual Models From Natural Language Supervision. ICML 2021.
>
> [3] Evolutionary-scale prediction of atomic-level protein structure with a language model. Proceedings of the National Academy of Sciences 2021.
>
> [4] ProtTrans: Toward Understanding the Language of Life Through Self-Supervised Learning. IEEE Transactions on Pattern Analysis and Machine Intelligence 2022.
>
> [5] L+M-24: Building a Dataset for Language + Molecules. ACL 2024.
>
> ---
> **_As reflected in our title, ICRL serves as a case study in our broader effort to explore how LLMs can reason over non-text modalities in a training-free manner. We sincerely thank all reviewers for their thoughtful feedback and will incorporate the suggestions and insights from this rebuttal into the next revision and future work._**

---

> ### Author Response · Authors · 2025-08-05
> **To save reviewer's time, we put a summary of rebuttal**
>
> Dear Reviewer J6ma,
>
> Thanks so much again for the time and effort in our work. Considering the limited time available and to save the reviewer's time, we summarized our responses below.
>
> **\[Q1: More Diverse Set of Evaluations]**
>
> **Response:**
> The experiments in the main paper primarily focus on molecular tasks for **deeper analysis** under controlled conditions. However, we have explored **other modalities such as protein, vision, and audio**, detailed in the appendix due to space constraints. Additionally, we evaluated **more task types**, including QA and generation, and conducted further experiments with lightweight projector training strategies. Our latest results confirm that key insights, such as the impact of representation similarity, **generalize across different modalities**.
>
> Specifically, embedding-level representations using diverse FM encoders significantly **outperform ICL alone**, while highly similar embeddings offer limited gains. Thus, our primary conclusions regarding representation similarity and model performance are **modality-agnostic and widely applicable**.
>
> **\[Q2: Discussion on Understanding Inconsistent Results]**
>
> **Response:**
> To address inconsistencies in ICRL performance, we explored training-based strategies for projectors. However, even explicit projector pretraining (via molecular captioning and contrastive learning) consistently **underperformed compared to our training-free OT-PCA method**. Our analysis indicates that performance inconsistency stems from the LLM's inherent difficulty interpreting representation-based inputs, absent during its pretraining.
>
> Traditional training approaches, despite improving familiarity with molecular representations, fail to guarantee better downstream regression performance. In contrast, our training-free ICRL enables the LLM to interpret FM representations effectively at test time, providing **superior performance with minimal computational overhead**.
>
>
> Since the discussion stage is already halfway through, may I know if our rebuttal addresses the concerns? If there are further concerns or questions, we are more than happy to address them. Thanks again for taking the time to review our work and provide insightful comments.
>
> Best regards,
>
> Authors

---

### Official Review · Reviewer_S7er · 2025-07-15

**Clarity:** 3
**Significance:** 3
**Originality:** 2
**Rating:** 4
**Confidence:** 2

**Summary:**

This paper investigates the feasibility of enabling Large Language Models (LLMs) to reason over non-text modalities in a training-free manner. Unlike traditional in-context learning, which incorporates text-label pairs, the proposed method replaces text inputs with FM representations, enabling the LLM to perform multi-modal inference without fine-tuning. Experiments show its feasibility and potential for generalizable multimodal reasoning.

**Questions:**

Below are some sub-topics that I'd like clarity on and included in the paper.

1. While the paper provides theoretical support for linear projectors, could a lightweight, learnable projection module (perhaps with minimal training) further improve performance without negating the "training-free" promise of ICRL?

2. How does the computational cost and latency of the optimal transport alignment method scale with increasingly larger LLMs and higher-dimensional FM representations during inference?

3. Are there specific types of non-text modalities or tasks where ICRL's training-free approach might inherently struggle or be less effective compared to others, even within its target domains (e.g., molecular, sensor-based)?

**Ethical Concerns:**

["NO or VERY MINOR ethics concerns only"]

**Limitations:**

Please refer the weakness section above.

**Quality:**

3

**Strengths And Weaknesses:**

Strengths:

1. The paper is decently motivated, well written and easy to understand. The problem statement is clear and the requirements driven table (Table 13) in appendix makes it easier to understand the contributions w.r.t existing baselines.

2. The paper successfully demonstrates the feasibility of LLMs performing multi-modal inference by leveraging FM representations with few-shot learning. This opens up new avenues for LLM application in diverse domains without needing the resources for additional training or fine-tuning of pre-trained LLMs.

3. The experimental section is thorough, and discusses various relevant questions from different angles aside from the traditional benchmarking with baselines which makes the paper more insightful and takeaways more concrete.

Weaknesses:

1. As acknowledged in the paper, ICRL underperforms compared to supervised methods due to the absence of task-specific training. This suggests a trade-off between the training-free advantage and peak performance in well-established domains. I'd like to see this comparison to make the takeaway more concrete. The paper is insightful in its current form, however, as a reader i am not sure about the trade-off b/w accuarcy and cost when comparing training-free method vs the traditional route.

2. The reliability of the method on carefully tuning PCA parameters makes its non-trivial to use, particularly for non-ICL scenarios.

---

> ### Author Rebuttal · Authors · 2025-07-30
>
> We sincerely thank Reviewer S7er for the valuable comments. For the concerns and questions, we provide the following responses.
>
> **W1: Unclear Trade-off Between Accuracy and Cost**
>
> Thank you for the insightful comment.
>
> To clarify the trade-off between performance and cost, we conducted a comparative analysis of ICRL against four recent
> LLM-based methods for molecular property prediction and found that
> **ICRL achieves comparable or even superior performance to lightweight fine-tuning methods—while requiring no training and only ~2 seconds of CPU computation**.
>
> Specifically, we compare ICRL with instruction tuning via Q-LoRA (I-FT) [1], supervised full-parameter pretraining followed by
> supervised fine-tuning (S-PT + FT) [2], and unsupervised full-parameter pretraining followed
> by supervised fine-tuning (PT + FT) [3,4]. The comparison spans training paradigms, resource consumption,
> training duration, and performance on two benchmark datasets.
> **_We will incorporate this comparative analysis into the main text in the next revision to make the trade-off more transparent to readers._**
>
>
> **Setup:**
>
> -- To ensure fairness, we implement ICRL using the LLaMA-3.1-8B-Instruct model—comparable in inference cost to those used in prior work and feasible on a single GPU. Evaluation is based on RMSE. Notably, ICRL achieves even better performance with larger models; e.g., on ESOL, it reaches 0.839 RMSE using LLaMA-3.1-70B.
>
> -- Few-shot examples in ICRL are selected via stratified sampling from the training set, with 20 samples for ESOL and 50 for Lipophilicity, following the same protocol as in our main paper.
>
> -- Regarding computational cost, we extract information directly from the original papers whenever possible. For example, [1] reports using 4× A800-80G GPUs for I-FT, and [3] states that pretraining took 11 days. When GPU specifications or training times are not provided, we estimate them based on similar work. Performance metrics are collected from: [4] Table 3; [3] Table 8; [1] Table 2; and [2] Table 3. Standard deviations are reported only when available in the original texts.
>
> |     |     |     |     |     |     |     |
> | --- | --- | --- | --- | --- | --- | --- |
> |     | Type | Resource | Time | Performance (ESOL) | Performance (Lipophilicity) | Performance (Avg) |
> | MolecularGPT [1] | I-FT | 4 Tesla A800-80G GPUs | < 1 day | 1.471 | 1.157 | 1.314 |
> | Gimlet [2] | S-PT + FT | 2-4 GPUs | ~ 1 day | 1.132 | 1.345 | 1.239 |
> | SELFormer [3] | PT  | 2 NVIDIA A5000 GPUs | ~ 2 weeks | 1.357 | 3.192 | 2.275 |
> | | PT + FT | 2 NVIDIA A5000 GPUs |~ 2 weeks | 0.682 | 1.005 | 0.844 |
> | GPT-MolBERTa [4] | PT + FT | 2-4 GPUs | ~ 2 weeks | 0.477±0.01 | 0.758±0.01 | 0.612 |
> | **OT-PCA (ours)** | **Traing-free** | **CPUs** | **~ 2 seconds** | **1.140±0.01** | **1.349±0.01** | **1.245** |
> | **OT-PCA + ICL (ours)** | **Traing-free** | **CPUs** | **~ 2 seconds** | **1.094±0.01** | **1.277±0.01** | **1.186** |
>
> **Analysis:**
>
> -- Under comparable inference cost, **ICRL achieves performance on par with or even superior to that of lightweight fine-tuning and short-duration pretraining + fine-tuning approaches, all while remaining entirely training-free**. While lightweight tuning can improve downstream performance to some extent, its effect is limited when the base model has not been pre-trained on small-molecule corpora. In contrast, ICRL allows a general-purpose LLM to achieve competitive or better results with only **~2 seconds** of overhead, highlighting both the method's effectiveness and practicality.
>
> -- Improved understanding of small molecules does not necessarily translate to better performance on downstream tasks. In analogy to LLM pretraining, the pretraining in baseline methods is primarily intended to enhance molecular understanding. However, as shown in [3], the standalone pretrained model performs significantly worse than ICRL on downstream prediction, suggesting that pretraining alone mainly serves as a warm-up for subsequent supervised tuning.
>
> **Conclusion:**
>
> While a performance gap still exists between ICRL and full PT + FT pipelines, ICRL offers **comparable or even superior results** to supervised lightweight tuning—**without any training**—underscoring its practical value in low-resource or rapid-deployment scenarios.
>
>
> ---
>
> **W2: Method Requires Careful PCA Tuning**
>
> Thank you for the comment.
>
> We would like to clarify that the analysis presented in Figure 3(a) and 3(b) focuses specifically on _text-level injected representations_, aiming to show that more expressive representations do not always help the model better utilize FM features. In contrast, _embedding-level injection methods_ such as OT-PCA are much less sensitive to the PCA dimension—particularly when combined with ICL, as demonstrated in Appendix F.2.2 and Figure 6.
>
> In fact, across all modalities and datasets used in both the main paper and the rebuttal, we consistently set the PCA dimension to 20. This parameter **was not tuned during method implementation**; the figures simply illustrate a special **case analysis** rather than a requirement for manual adjustment.
>
> ---
>
> **Q1: Potential of Lightweight Learnable Projectors**
>
> Thank you for your insightful comment.
>
> As shown in W1, while lightweight fine-tuning can improve performance to some extent, it still underperforms compared to the training-free ICRL method—largely due to the inherent difficulty of molecular property prediction tasks for general-purpose LLMs.
>
> To further investigate this question, we explored two lightweight training strategies for the random projectors used in ICRL: one based on contrastive learning (inspired by CLIP), and another based on molecular captioning, following concurrent work. However, as shown in **Reviewer J6ma Q2**, both approaches perform worse than our OT-PCA method.
>
> We conducted additional experiments by directly training the projector on the downstream task, and found that this approach led to lower predictive performance and impaired generation quality, especially in few-shot settings. Specifically, due to the limited scale of the datasets and the lack of domain-specific knowledge in the LLM, this setup often degraded the model's in-context learning ability—sometimes even preventing it from producing complete outputs.
>
> These findings suggest that for these small molecular datasets, **lightweight training** of the projector—while feasible in principle—tends to result in **worse** performance compared to our **training-free** design.
>
> ---
>
> **Q2: Scalability of OT Alignment with Model Size**
>
> Thank you for the comment.
>
> We would like to clarify that the OT alignment process does **not** scale significantly with the size of the LLM or the dimensionality of the FM features. The table below summarizes the runtime of OT under different LLaMA model sizes and FM feature dimensions. Importantly, like traditional pretraining or fine-tuning, OT alignment is a **one-time computation**.
>
> During inference, applying OT requires only a single matrix operation per sample, which takes **under 1 millisecond** in our setup—adding negligible overhead to the overall generation process.
> We provide a more detailed discussion of ICRL's inference cost in Appendix E.7 and Table 12.
>
> |     |     |     |     |
> | --- | --- | --- | --- |
> |     | 3 B | 8 B | 70 B |
> | Unimol (512) | ~ 0.842 sec | ~ 0.843 sec | ~ 0.846 sec |
> | ESM2 (640) | ~ 0.844 sec | ~ 0.844 sec | ~ 0.849 sec |
>
> ---
>
> **Q3: Limitations of ICRL on Certain Modalities or Tasks**
>
> Thank you for your insightful comment.
>
> Indeed, our extensive experiments suggest that while ICRL can enhance LLM performance by leveraging FM representations, the degree of improvement is closely tied to the **"difficulty"** of the downstream task for the LLM—roughly indicated by its ICL performance.
>
> Specifically, although ICRL outperforms lightweight training approaches on molecular property prediction and molecular QA tasks, it performs less effectively on caption generation tasks **(see Reviewer aUho W2)**. In these cases, the base ICL performance is already poor, suggesting that general-purpose LLMs struggle to learn such complex mappings from a few examples alone. While ICRL still outperforms ICL in this setting, the gap between ICRL and task-specific expert models remains substantial.
>
> A similar limitation appears in protein property prediction tasks. Due to the high sequence similarity within protein datasets, it becomes difficult for the model to distinguish between samples, leading to weak ICL and ICRL performance (see Appendix E.3). This issue, however, can be mitigated by adopting stronger FMs, as discussed in **Reviewer J6ma W1**.
>
>
> ---
>
> [1] MolecularGPT: Open Large Language Model (LLM) for Few-Shot Molecular Property Prediction. Arxiv 2023.
>
> [2] GIMLET: A Unified Graph-Text Model for Instruction-Based Molecule Zero-Shot Learning. NeurIPS 2023.
>
> [3] SELFormer: Molecular Representation Learning via SELFIES Language Models. Machine Learning: Science and Technology 2023.
>
> [4] GPT-MolBERTa: GPT Molecular Features Language Model for molecular property prediction. Arxiv 2023.
>
> ---
>
> **_As reflected in our title, ICRL serves as a case study in our broader exploration of how LLMs can reason over non-text modalities in a training-free manner. We sincerely appreciate all the valuable feedback provided by the reviewers and will incorporate the insights and suggestions from this rebuttal into the revised version of the paper or pursue them in future work._**

---

> ### Author Response · Authors · 2025-08-05
> **To save reviewer's time, we put a summary of rebuttal**
>
> Dear Reviewer S7er,
>
> Thanks so much again for the time and effort in our work. Considering the limited time available and to save the reviewer's time, we summarized our responses below.
>
> **\[W1: Unclear Trade-off Between Accuracy and Cost]**
>
> **Response:**
> We conducted a comprehensive comparative analysis of ICRL against recent LLM-based molecular prediction methods. Results demonstrate that ICRL achieves performance **comparable to or even superior** than lightweight fine-tuning methods—with **no training required** and just **~2 seconds of CPU computation**. We will include this detailed analysis in the next revision for transparency.
>
> **\[W2: Method Requires Careful PCA Tuning]**
>
> **Response:**
> The PCA dimension parameter (set consistently at 20) was **not tuned** during implementation. Sensitivity highlighted in Figure 3 pertains specifically to text-level injections. Embedding-level injections, such as OT-PCA used in our method, exhibit robustness to PCA dimension, especially combined with ICL, as detailed in Appendix F.2.2.
>
> **\[Q1: Potential of Lightweight Learnable Projectors]**
>
> **Response:**
> Experiments with lightweight learnable projectors (contrastive learning and molecular captioning) consistently **underperformed compared to our training-free OT-PCA method**, particularly on small molecular datasets. Training projectors directly degraded performance, reinforcing the advantage of our training-free design.
>
> **\[Q2: Scalability of OT Alignment with Model Size]**
>
> **Response:**
> Optimal Transport alignment computation time does not scale significantly with LLM size or FM feature dimension. OT is a **one-time** computation, and inference-time overhead is **negligible (under 1 ms per sample)**. See detailed results in Appendix E.7 and Table 12.
>
> **\[Q3: Limitations of ICRL on Certain Modalities or Tasks]**
>
> **Response:**
> ICRL performance improvements correlate with task difficulty for general-purpose LLMs. While outperforming lightweight training approaches on molecular prediction and QA tasks, ICRL shows limited gains in tasks **inherently challenging for few-shot learning**, such as caption generation and protein prediction with high sequence similarity. Stronger foundational models can mitigate this limitation (discussed further in Reviewer J6ma W1).
>
> Since the discussion stage is already halfway through, may I know if our rebuttal addresses the concerns? If there are further concerns or questions, we are more than happy to address them. Thanks again for taking the time to review our work and provide insightful comments.
>
> Best regards,
>
> Authors

---

### Author Response · Authors · 2025-08-07
**Summary of Key Discussion Points**

Dear Reviewers and Area Chairs,

We express our sincere gratitude for the time, effort, and constructive feedback you have provided on our work. The insightful comments from all reviewers have been extremely helpful in shaping our understanding of the paper’s strengths and identifying areas for improvement. Below, we summarize the most important discussion points that emerged during the rebuttal phase:

---

**1. Trade-off Between Accuracy and Cost**

Several reviewers raised the need to clarify the **practical trade-off** between the performance of ICRL and that of training-based methods.

To address this, we conducted a comprehensive comparison between ICRL and a range of recent molecular property prediction pipelines, including instruction-tuned models (Q-LoRA), supervised and unsupervised pretraining followed by fine-tuning, as well as our own training-free methods (OT-PCA and OT-PCA + ICL).

This comparison, detailed in **Table in Reviewer S7er W1**, shows that
ICRL achieves performance **comparable to or better** than several training-based methods—
while requiring **only \~2 seconds of CPU computation**.

This underscores ICRL's **practicality and efficiency**, particularly in low-resource or rapid-deployment scenarios.



---

**2. Lightweight Trainable Projectors**

Multiple reviewers asked whether **lightweight trainable projectors** could improve performance over our training-free OT-PCA design.

To investigate this, we implemented several representative strategies (see **Reviewer J6ma Q2**), including contrastive learning, caption-based pretraining, their combination, and direct downstream fine-tuning.

However, all training-based approaches **consistently underperformed** compared to our **training-free** approach. In particular:


* On small molecular datasets, training often **degraded** the model's ICL behavior, likely due to the LLM's **lack of domain knowledge** and the **limited amount of supervision** available.
* Direct downstream fine-tuning yielded especially poor results—sometimes preventing the model from producing complete outputs—highlighting the risk of overfitting and poor generalization in few-shot settings.

These results suggest that, under our current setting—where a general-purpose LLM is applied to unfamiliar molecular tasks—**lightweight training strategies are insufficient**. The lack of prior exposure to domain-specific inputs likely limits their effectiveness.

In contrast, OT-PCA remains more **robust and stable**, making it a practical plug-and-play solution for low-resource and domain-shifted scenarios.


---

**3. Expanded Modalities and Task Diversity**

Several reviewers emphasized the need to validate ICRL across more **modalities** and **task types**.
To address this, we extended our evaluation to include diverse data modalities—**molecules**, **proteins**, **vision**, and **audio**—and task types such as **regression**, **classification**, **question answering**, and **caption generation**.

Across all settings, **ICRL consistently outperformed standard ICL** and performed **on par with fine-tuned baselines**, even when the base LLM had little prior exposure to the target domain (see **Reviewer aUho W2**).

In addition, our results suggest that ICRL's effectiveness is primarily influenced by
the characteristics of the representation rather than the input modality itself,
reinforcing the **modality-agnostic nature** of the framework.

Specifically, we observed that high similarity among injected representations limits performance gains—a pattern first identified in molecular tasks.
To test the generality of this finding, we conducted additional experiments in the protein modality, where switching to a more diverse FM encoder (e.g., from ESM to ProtBert) similarly led to substantial performance improvements (see **Reviewer J6ma Q1**).

These consistent trends across different domains underscore the **generality and robustness of the experimental conclusions** presented in the paper.

---

We thank you again for your engagement and thoughtful comments.
Please see the second message for our detailed revision plan.

Best regards,

Authors

---

> ### Author Response · Authors · 2025-08-07
> **Planned Revisions Based on Reviewer Feedback**
>
> Dear Reviewers and Area Chairs,
>
> Following up on our earlier summary of key discussion points, we provide below a detailed overview of our planned revisions in response to reviewer feedback.
>
>
> **Textual Revisions**:
>
> We will revise the manuscript to incorporate the following changes:
>
> * In **Section 3**, we will add a detailed description of the **stratified sampling strategy** used for in-context example selection, including the binning-based procedure.
>
> * In **Section 4**, we will:
>
>   * Clarify the use of **multiple regression metrics** (RMSE, Pearson's *r*, and Spearman's ρ), and state that experimental conclusions are consistent across these metrics.
>   * Include a discussion on **Tanimoto similarity-based selection**, and compare its effectiveness with stratified sampling.
>   * Add a **comparative analysis of training-free and lightweight fine-tuning methods**, supported by a cost-performance table across six methods for molecular property prediction.
>
> * In **Section 7**, we will:
>
>   * Expand the discussion on how **representation similarity and embedding variance** influence ICRL performance across modalities.
>   * Analyze **failure cases of lightly-trained projectors** (e.g., captioning- and contrastive-based training), and contrast them with the robustness of the training-free approach.
>   * Highlight **limitations of ICRL** on tasks with high inter-sample similarity (e.g., protein datasets), and discuss how stronger foundation models can mitigate these challenges.
>
> ---
>
> **Experimental Revisions**
>
> We will conduct and report the following experiments, with corresponding tables and implementation details added to the appendix, and the main findings discussed in the main text:
>
> * **In-context example selection:** Evaluate the use of **Tanimoto similarity** for selecting examples, and compare with stratified sampling in terms of ICL and ICRL performance.
>
> * **Cost-performance comparison:** Provide a **comparative table** across six baseline methods, reporting training cost, inference time, and regression accuracy on ESOL and Lipophilicity datasets.
>
> * **Protein modality evaluation:** Compare different FM encoders (e.g., ESM vs. ProtBert) to assess how **embedding similarity influences performance**, and to demonstrate that key conclusions—such as the relationship between representation diversity and model effectiveness—are **modality-agnostic**.
>
> * **Task diversity experiments:** Report results on **molecular QA** and **caption generation** tasks to evaluate ICRL beyond regression settings.
>
> * **Lightweight projector training strategies:** Evaluate three variants—**caption-only**, **contrastive-only**, and **combined** training—on both generation quality (BLEU/ROUGE) and regression performance (RMSE), and analyze failure modes relative to the training-free baseline.
>
> Again, we thank all reviewers and ACs for your thoughtful comments and support. We are excited to incorporate these revisions and continue refining our work. Should there be any additional questions or suggestions, we would be more than happy to discuss further.
>
> Best regards,
>
> Authors

---

### Note · Authors · 2025-08-12

We sincerely thank the Area Chair and all reviewers for their constructive feedback and engagement throughout the review process.

We are gratified that Reviewers **S7er**, **J6ma**, **aUho**, and **nfkW** acknowledged the originality, efficiency, and potential impact of ICRL, and that our detailed rebuttal and
new experiments effectively addressed their key concerns.

During the discussion, we provided substantial new analyses and experiments:

* A **comprehensive cost–performance comparison** showing that ICRL achieves performance comparable to or better than multiple fine-tuning pipelines, with **no training** and only \~2 seconds of CPU computation.
* Systematic evaluation of **lightweight** projector training strategies (caption-based, contrastive, and combined), all of which **underperformed** relative to our training-free OT-PCA design, highlighting its robustness in low-resource, domain-shifted scenarios.
* Expanded experiments across **four modalities** (molecules, proteins, vision, audio) and **multiple task types** (regression, classification, QA, captioning), confirming that both the effectiveness of ICRL and the conclusions of our study **generalize beyond regression tasks and the molecule modality.**

We also thank Reviewer AQ11 for the effort. While we appreciate the intent behind the feedback, we note that the review contained multiple factual inaccuracies,
overlooked substantial content already presented in the paper, and included behavior inconsistent with NeurIPS review guidelines.
We have provided point-by-point clarifications in our rebuttal and respectfully ask the AC to take these issues into consideration in the final assessment.

Overall, the reviewer–author dialogue has greatly improved the clarity, scope, and evaluation of our work. In the final version, we will incorporate all agreed-upon revisions, including expanded modality coverage, additional analyses, and clearer methodological descriptions.
We believe the proposed training-free framework—defined by its **novelty, efficiency, and generality**—offers a **compelling alternative to traditional training paradigms** and opens up a **new direction for multimodal reasoning with LLMs**.

Once again, we sincerely thank the Area Chair and all reviewers for their time, thoughtful feedback, and constructive engagement throughout the review process.

---

### Decision · Program_Chairs · 2025-09-17

**Decision:**

Accept (poster)

**Comment:**

This work introduces the training-free strategy of In-Context Representation Learning that lets a text-only LLM consume continuous embeddings from a non-text foundation model at inference time. The merged prompt is treated as an extended few-shot context. Results focus on molecular property regression but also show results for vision and audio. Uni-Mol is used as the foundation model and Llama-3.1-70B is used as the LLM.

Strengths:
* The paper is very insightful and novel.
* The main question is well laid out and motivated and easy to understand.
* The experimental setup is extensive and the effect well analyzed.

Weaknesses:
* The work would benefit from more baselines especially with various projectors.
* The modalities analyzed are limited. The focus is on molecules in SMILES format with light analysis on vision/audio generalization. It would be good to have stronger vision/audio evaluations with 2D/3D molecular formats.

The idea of adapting to other modalities without any training is novel and important as large open-weights models are generally text only and adapting them via fine-tuning to new modalities is prohibitively expensive for most users. Given the successful experimental results in this paper (on chemistry and other modalities) compared to other projector-based approaches I expect it will be impactful to the field by enabling the use and understanding of large models for more researchers working on non-text modalities. I recommend this paper be accepted.

In the discussion, there was debate around ICL results being better than the proposed approach. If the authors moved table 16 into the main part of the paper (where they show the approach is better combined with ICL than ICL alone), that would have resolved the discussion earlier. There was also discussion around various projectors and if they could improve the results (it seems they mostly don't but evaluation there could be more comprehensive). Reviewer AQ11 argued for rejection, however, there were factual inaccuracies in the review (claiming that only the chemistry modality was validated), the review was mostly superficial and the reviewer didn't engage in the discussion.

Finally, there was discussion around additional task types and modalities, the evaluation with other modalities could be more thorough and the authors ran additional experiments on new task types. I appreciate the authors putting in the additional time and effort to get additional results for the rebuttals, these seem to have resolved most of the concerns from the authors.